# Co-rewarding: Stable Self-supervised RL for Eliciting Reasoning in Large Language Models

**Zizhuo Zhang**[1*]   **Jianing Zhu**[1*]   **Xinmu Ge**[2,3*]   **Zihua Zhao**[3*]   **Zhanke Zhou**[1]
**Xuan Li**[1]   **Xiao Feng**[1]   **Jiangchao Yao**[3†]   **Bo Han**[1†]
[1]TMLR Group, Department of Computer Science, Hong Kong Baptist University
[2]Shanghai Innovation Institute   [3]CMIC, Shanghai Jiao Tong University
{cszzzhang,csjnzhu,cszkzhou,csxuanli,xiaofeng}@comp.hkbu.edu.hk
bhanml@comp.hkbu.edu.hk,{g3ra1d,sjtuszzh,Sunarker}@sjtu.edu.cn

## ABSTRACT

While reinforcement learning with verifiable rewards (RLVR) is effective to improve the reasoning ability of large language models (LLMs), its reliance on human-annotated labels leads to the scaling up dilemma, especially for complex tasks. Recent self-rewarding methods investigate a label-free alternative to unlock the reasoning capabilities of LLMs, yet they frequently encounter the non-negligible training collapse issue, as the single-view supervision signal easily forms the self-consistent illusion, yielding the reward hacking. Inspired by the success of self-supervised learning, we propose *Co-rewarding*, a novel self-supervised RL framework that improves training stability by seeking complementary supervision from another views. Specifically, we instantiate Co-rewarding in two ways: (1) *Co-rewarding-I* is a data-side instantiation that derives reward signals from contrastive agreement across semantically analogous questions; and (2) *Co-rewarding-II* is a model-side instantiation that maintains a slowly-updated reference teacher with pseudo labels to realize self-distillation. Intuitively, such instantiations introduce different levels of discrepancy to increase the difficulty of training collapse on trivial reasoning solutions. We also explore their orthogonally combined version to further boost the performance. Empirically, Co-rewarding exhibits stable training across various setups, and outperforms other self-rewarding baselines by $+3.31\%$ improvements on average on multiple mathematical reasoning benchmarks, especially by $+7.49\%$ on Llama-3.2-3B-Instruct. Notably, Co-rewarding reaches or even surpasses RLVR with ground-truth (GT) label in several cases, such as a Pass@1 of $94.01\%$ on GSM8K with Qwen3-8B-Base remarkably higher than GT. Our code is released at https://github.com/tmlr-group/Co-rewarding.

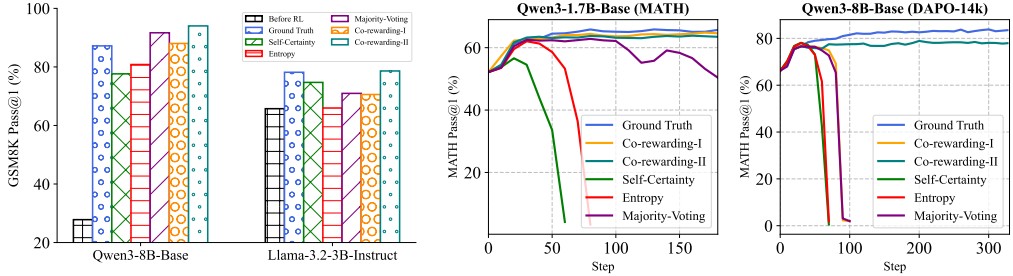

Figure 1: **Performance overview.** Reasoning comparison of Pass@1 value and validation curves. Our Co-rewarding achieves better and more stable (without collapse) training than other baselines.

## 1 INTRODUCTION

Large language models (LLMs) (Achiam et al., 2023; Dubey et al., 2024) have demonstrated remarkable general-purpose capabilities in a wide range of linguistic tasks (Li et al., 2026; Chen

---

*Equal contribution.
†Correspondence to Bo Han (bhanml@comp.hkbu.edu.hk) and Jiangchao Yao (Sunarker@sjtu.edu.cn).

et al., 2026). To further elicit their reasoning ability in complex scenarios, reinforcement learning with verifiable rewards (RLVR) (Shao et al., 2024; Yu et al., 2025) is developed for post-training with externally verifiable signals like program execution results (Luo et al., 2025) or mathematical equivalence (Shao et al., 2024). Despite the impressive improvement, the reliance on high-quality ground-truth (GT) labels of RLVR remains as a major bottleneck (Ouyang et al., 2022; Bai et al., 2022) in the spirit of the scaling law, which subsequently motivates the emerging exploration of self-rewarding methods with unlabeled data (Zhao et al., 2025b; Zuo et al., 2025; Zhang et al., 2025c).

One prominent line of such label-free methods leverages the internal signals (e.g., entropy (Zhang et al., 2025d; Prabhudesai et al., 2025) and self-certainty (Zhao et al., 2025b)) to strengthen the confidence of the model in reasoning. Another critical line seeks the answer-level consensus (Zuo et al., 2025; Shafayat et al., 2025) to construct pseudo labels as reward basis. while effective initially, these self-rewarding approaches frequently exhibit non-negligible training collapse (Zhang et al., 2025e) (indicated as right of Figure 1), which limits the scalability of such label-free training manners.

The collapse phenomenon stems from reward hacking (Laidlaw et al., 2025) under self-consistent illusion: the reward signal is internally produced by the policy model from a single-view data perspective, which is easily trapped by trivial solutions along with training (see Figure 7). Specifically, for entropy- or certainty-based rewards, the policy model may concentrate probability mass on a small set of tokens and produce repetitive strings that minimize entropy or maximize self-certainty (Zhang et al., 2025e). And for consensus-based rewards, the policy model can converge to a consistent yet incorrect answer that attains high consensus across rollouts (Shafayat et al., 2025). Overall, the policy model continually reduces uncertainty without sustained gains in correctness, inflating the reward but eroding exploration and diversity. It ultimately collapses when a persistent hacking strategy emerges.

To this end, we introduce *Co-rewarding*, a self-supervised RL framework that seeks complementary supervision from another views, inspired by self-supervised learning (Chen et al., 2020; Grill et al., 2020; Caron et al., 2021). Conceptually, one fundamental characteristic of self-rewarding methods lies on that supervision intertwined with current policy on single-view outputs, for which we propose to seek reasoning invariance across different views (see Figure 2). Specifically, we investigate two initiations of Co-rewarding: (1) *Co-rewarding-I*: a data-side initiation that constructs rewards via contrastive agreement across semantically analogous questions, each providing pseudo labels for the other; and (2) *Co-rewarding-II*: a model-side initiation that introduces an extra teacher with dynamically updated policy and provides stable pseudo-labels insulated from current online policy. Additionally, we also explore the combined instantiation, *Co-rewarding-III*, which integrates data-side cross-supervision with model-side self-distillation to further boost the performance.

By introducing cross-view supervision on data and decoupling the reward signal from the current policy, Co-rewarding effectively mitigates training collapse and yields stable self-supervised RL training. Extensive experiments across multiple datasets validate the stability and superiority of Co-rewarding, compared to several recent baselines across several LLM families including Qwen3/2.5 and Llama. Notably, both Co-rewarding-I and -II reach or exceed training with ground-truth labels in several settings, such as achieving up to $94.01\%$ Pass@1 on GSM8K. Our main contributions are

- We introduce a new perspective, from self-supervised learning, to elicit reasoning capability via another views of supervision, which prevents the model from training collapse (Section 3.1).

- We propose Co-rewarding, a novel self-supervised RL framework that is initiated by the data and model sides to construct self-generate rewards to promote stably reasoning elicitation (Section 3.2).

- We empirically demonstrate the general effectiveness of Co-rewarding to achieve superior reasoning performance on LLMs, and also present various ablation studies and further analyses (Section 4).

## 2 PRELIMINARY

**Problem Setups.** Given a LLM $\pi_\theta$ parameterized by $\theta$ and a dataset $\mathcal{D}$ of question–answer pairs $(x, a)$, the model generates a response $y \sim \pi_\theta(\cdot \mid x)$ autoregressively. Let $y = (y_1, \ldots, y_n)$, where each token is sampled as $y_t \sim \pi_\theta(\cdot \mid x, y_{<t})$ given the generated prefix $y_{<t}$. We consider the LLM outputs a stepbystep reasoning trace and a final answer. A verifiable reward function $r(a, y)$ compares

the extracted answer $\mathrm{ans}(y)$ with the ground truth $a$ as follows:

$$r(a, y) = \begin{cases} 1 & \text{If } \mathrm{ans}(y) \text{ is correct with answer } a, \\ 0 & \text{If } \mathrm{ans}(y) \text{ is incorrect with answer } a. \end{cases} \tag{1}$$

Then, the general objective of training LLM for reasoning via RLVR (Shao et al., 2024; Yu et al., 2025) can be formulated with the policy model $\pi_\theta$ as follows:

$$\max_{\pi_\theta} \mathbb{E}_{(x,a) \in \mathcal{D}, \ y \sim \pi_\theta(x)}[r(a, y) - \beta \cdot \mathrm{KL}[\pi_\theta(y|x)||\pi_{\mathrm{ref}}(y|x)]], \tag{2}$$

where $\pi_{\mathrm{ref}}$ is an initial reference policy, and $\beta$ is a coefficient controlling the KL divergence to prevent excessive deviation from the reference model. Intuitively, the training target is to maximize the reward in passing specific reasoning questions while maintaining the general capability of LLM.

**Group Relative Policy Optimization (GRPO).** In practice, we adopt GRPO (Shao et al., 2024), a widely used and representative optimization method for objective Eq. (2) that estimates the advantage by normalizing the reward across multiple sampled outputs for the same question. Specifically, for a given question $x$, GRPO samples $G$ outputs from the old policy $\pi_{\mathrm{old}}$ as $\{y_i\}_{i=1}^{G} \sim \pi_{\mathrm{old}}(\cdot|x)$. It then computes a reward for each output $y_i$ via a deterministic reward function, forming a group of rewards $\{r(a, y_i)\}_{i=1}^{G}$ to estimate the advantage $\hat{A}_i$ as follows:

$$\hat{A}_i = \frac{r(a, y_i) - \mathrm{mean}(\{r(a, y_i)\}_{i=1}^{G})}{\mathrm{std}(\{r(a, y_i)\}_{i=1}^{G})}. \tag{3}$$

Then, the target policy is optimized by maximizing the advantage while ensuring the policy model remains close to the reference policy:

$$\mathcal{J}_{\mathrm{GRPO}}(\theta) = \mathbb{E}_{(x,a) \in \mathcal{D}, \{y_i\}_{i=1}^{G} \sim \pi_{\theta_{\mathrm{old}}}(\cdot|x)}$$

$$\underbrace{\frac{1}{G} \sum_{i=1}^{G} \frac{1}{|y_i|} \sum_{t=1}^{|y_i|} \left( \min\left[ c_{i,t}(\theta)\hat{A}_{i,t}, \mathrm{clip}(c_{i,t}(\theta), 1-\epsilon, 1+\epsilon)\hat{A}_{i,t} \right] - \beta \mathbb{D}_{\mathrm{KL}}(\pi_\theta||\pi_{\mathrm{ref}}) \right)}_{\mathcal{R}_\theta(\hat{A})}, \tag{4}$$

where

$$c_{i,t}(\theta) = \frac{\pi_\theta(y_{i,t}|x, y_{i,<t})}{\pi_{\theta_{\mathrm{old}}}(y_{i,t}|x, y_{i,<t})}, \ \mathbb{D}_{\mathrm{KL}}(\pi_\theta||\pi_{\mathrm{ref}}) = \frac{\pi_\theta(y_{i,t}|x, y_{i,<t})}{\pi_{\mathrm{ref}}(y_{i,t}|x, y_{i,<t})} - \log \frac{\pi_{\mathrm{ref}}(y_{i,t}|x, y_{i,<t})}{\pi_\theta(y_{i,t}|x, y_{i,<t})} - 1. \tag{5}$$

Note that the $\mathrm{clip}(\cdot, 1-\epsilon, 1+\epsilon)$ in Eq. (4) is used to ensure that updates do not deviate excessively from the old policy by bounding the policy ratio between $1-\epsilon$ and $1+\epsilon$ in a risk function $\mathcal{R}(\hat{A})$. We also provide a comprehensive discussion on additional training variants for RLVR, such as DAPO (Yu et al., 2025) and Dr. GRPO (Liu et al., 2025a), which we leave in Appendix A due to space limits.

# 3 CO-REWARDING

In the following, we present Co-rewarding in detail, a novel self-supervised RL framework for LLM to elicit the latent reasoning capability through the intuition of seeking complementary supervision.

## 3.1 CONCEPTUAL PHILOSOPHY: INVARIANCE BEYOND THE SINGLE-VIEW

At the core of self-rewarding methods lies a fundamental tension: the model derives supervisory signals from its own outputs, inevitably intertwining supervision with policy and risks collapse. True reasoning competence, however, cannot be reduced to the mere correctness of isolated answers. It should instead reflect invariance that extends beyond the single-view output for consistency. This calls for training signals that remain valid across different data views or persist throughout the temporal evolution of the model, providing a more reliable basis on which self-supervised RL can rely. In this aspect, stability arises from invariance that prevents reasoning against superficial variations in data and guides the model towards increasingly valid reasoning trajectories throughout training.

This philosophy yields our Co-rewarding framework, whose core idea is to ground self-supervised RL in invariance rather than the suspicious single-view feedback. We instantiate it in two orthogonal ways and one combined version: by enforcing analogy-invariance on the data side (Co-rewarding-I), by disentangling supervision through temporal invariance on the model side (Co-rewarding-II), and by integrating both mechanisms in a unified instantiation (Co-rewarding-III).

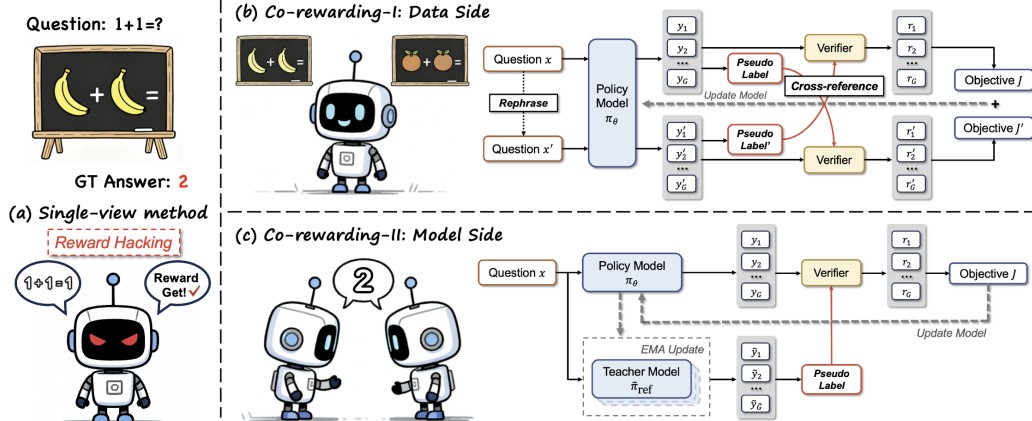

Figure 2: **Illustration of *Co-rewarding* framework**: Unlike single-view methods that rely only on internal reward signal on original question (a), Co-rewarding introduces complementary supervision. On the data side (b), paraphrased questions yield pseudo-labels for cross-reference. On the model side (c), teacher model isolated from current policy provides stabilized pseudo-labels for updates.

### 3.2 Two Initiations of Co-rewarding Framework

**Co-rewarding-I:** *on the Data Side*. Inspired by contrastive learning, such as SimCLR (Chen et al., 2020) and InfoNCE (Oord et al., 2018), where two views of the same data are encouraged to have similar representations, we hypothesize an analogy-invariance inductive property of LLMs in eliciting reasoning capacity: questions that share the same mathematical essence but differ in surface form (e.g., via paraphrasing, background substitution, or reformatting) should elicit the comparably valid and similar reasoning results. This forms the foundation for a self-referential training signal: contrastive agreement among different question variants can serve as an optimization proxy. Co-rewarding-I defines contrastive agreement as a principle that aligns model reasoning outputs, treating consistent inter-view agreement as a signal for valid inference. This complements single-view self-rewarding strategies by introducing a form of collective validity verification with broader input consideration.

Building upon the discussed contrastive agreement, we initiate our *Co-rewarding-I* as illustrated in Figure 2. Formally, its learning objective can be formulated as follows based on GRPO:

$$\mathcal{J}_{\text{Co-rewarding-I}}(\theta) = \underbrace{\mathbb{E}_{x \in \mathcal{D}, \{y_i\}_{i=1}^G \sim \pi_{\theta_{\text{old}}}(\cdot|x)} \mathcal{R}_\theta(\hat{A})}_{\mathcal{J}_{\text{original}}(\theta)} + \underbrace{\mathbb{E}_{x' \in \mathcal{D}', \{y_i'\}_{i=1}^G \sim \pi_{\theta_{\text{old}}}(\cdot|x')} \mathcal{R}_\theta(\hat{A}')}_{\mathcal{J}_{\text{rephrased}}(\theta)}, \quad (6)$$

where the relative advantages are estimated by the *cross-refereed* supervision as follows:

$$\hat{A}_i = \frac{r(y_{\text{v}}', y_i) - \text{mean}(\{r(y_{\text{v}}', y_i)\}_{i=1}^G)}{\text{std}(\{r(y_{\text{v}}', y_i)\}_{i=1}^G)}, \ \hat{A}_i' = \frac{r(y_{\text{v}}, y_i') - \text{mean}(\{r(y_{\text{v}}, y_i')\}_{i=1}^G)}{\text{std}(\{r(y_{\text{v}}, y_i')\}_{i=1}^G)}. \quad (7)$$

Specifically, given a set of original questions, we utilize the rephrased version that keeps the semantical equivalence for the model to respond, and then collect the self-generated pseudo-labels based on the majority voting mechanism (Shafayat et al., 2025) as follows to supervise learning on the counterparts,

$$y_{\text{v}} \leftarrow \arg\max_{y*} \sum_{i=1}^G 1[\text{ans}(y_i) = \text{ans}(y*)], \quad y_{\text{v}}' \leftarrow \arg\max_{y*} \sum_{i=1}^G 1[\text{ans}(y_i') = \text{ans}(y*)]. \quad (8)$$

The overall pipeline can be viewed as a dual-path structure with cross-reference in the reward shaping process, it may also be compatible with other self-generated feedbacks (Wang et al., 2022) on the output-side information due to the generality of the core idea. While in the current version, we choose the majority voting mechanism in the implementation for the empirical effectiveness and simplicity.

We summarize the pseudo code of Co-rewarding-I in Algorithm 1. Our contrastive objective operates on self-generated reasoning answers, encouraging the model to align its reasoning results to different questions that share the similar semantic intent. Formally, for each input question, the signal of Co-rewarding-I increases when the model's output is consistent with the majority answer obtained from its analogical counterparts, and decreases when it diverges. This contrastive agreement promotes semantic invariance, implicitly increasing the difficulty of reaching trivial solutions to obtain the

reward (e.g., achieving the arbitrary answers but consistent on single input) by involving data-side analogy. We leave a more intuitive case study in the Appendix D.12 to present the rephrased questions.

**Co-rewarding-II: *on the Model Side*.** On the data side, our Co-rewarding-I provides complementary supervision by involving question analogy, while its pseudo-labels are still generated by the current online policy and may depend on rephrasing quality; consequently, supervision remains partially entangled with the policy. Inspired by self- or weakly supervised methods like the representative BYOL (Grill et al., 2020), DINO (Caron et al., 2021), and Co-teaching (Han et al., 2018), which share the common intuition of introducing an auxiliary network to provide supervision beyond the current model, we initiate *Co-rewarding-II* from another view of complementary supervision: a model-side strategy that sources pseudo-labels from a teacher reference, which disentangle the self-supervision reward from the online policy. To avoid the heavy cost of adding and maintaining another LLM in training, Co-rewarding-II reuses the GRPO reference model as the teacher to generate the rollouts and produce pseudo-labels. In particular, the teacher is dynamically updated as an exponential moving average (EMA) of the student policy to ensure pseudo-label quality improving as the policy improves.

Intuitively, we illustrate *Co-rewarding-II* in Figure 2. Its learning objective can be formulated as:

$$\mathcal{J}_{\text{Co-rewarding-II}}^{(k)}(\theta) = \mathbb{E}_{x\in\mathcal{D}, \underbrace{\{y_i\}_{i=1}^{G} \sim \pi_{\theta_{\text{old}}}^{(k)}(\cdot|x)}_{\text{policy student rollout}}, \underbrace{\{\tilde{y}_j\}_{j=1}^{\tilde{G}} \sim \tilde{\pi}_{\text{ref}}^{(k)}(\cdot|x)}_{\text{reference teacher rollout}}} \mathcal{R}_\theta(\hat{A}^{(k)}), \tag{9}$$

where $\{y_i\}_{i=1}^{G}$ are policy rollouts and $\{\tilde{y}_j\}_{j=1}^{\tilde{G}}$ are reference teacher rollouts at the $k$-th training step, and the estimated advantage $\mathcal{R}(\hat{A}^{(k)})$ is computed as follows:

$$\hat{A}_i^{(k)} = \frac{r(\tilde{y}_{\text{v}}^{(k)}, y_i) - \text{mean}(\{r(\tilde{y}_{\text{v}}^{(k)}, y_i)\}_{i=1}^{G})}{\text{std}(\{r(\tilde{y}_{\text{v}}^{(k)}, y_i)\}_{i=1}^{G})}, \quad \tilde{y}_{\text{v}}^{(k)} = \arg\max_{y*} \sum_{j=1}^{\tilde{G}} \mathbf{1}[\text{ans}(\tilde{y}_j) = \text{ans}(y*)], \tag{10}$$

where the pseudo label $\tilde{y}_{\text{v}}^{(k)}$ is obtained via majority voting from reference rollouts, and the reference model is updated via an EMA with the policy to play a role of a slowly updated teacher:

$$\tilde{\pi}_{\text{ref}}^{(k)} \leftarrow \alpha^{(k)} \cdot \tilde{\pi}_{\text{ref}}^{(k-1)} + (1 - \alpha^{(k)}) \cdot \pi_{\theta_{\text{old}}}^{(k)}, \quad \alpha^{(k)} = 1 - \frac{(\alpha_{\text{end}} - \alpha_{\text{start}})}{2}\left(1 + \cos\left(\frac{\pi k}{K}\right)\right) \tag{11}$$

where $\alpha^{(k)} \in (0, 1)$ is the EMA weight, updated according to a cosine annealing schedule from $\alpha_{\text{start}}$ to $\alpha_{\text{end}}$, such that the teacher is updated rapidly at the beginning and progressively more slowly, thereby evolving smoothly and remaining temporally decoupled from the current online policy.

We summarize the pseudo code of Co-rewarding-II in Algorithm 2. This design can be interpreted as a kind of self-distillation, in which a slowly updated teacher supervises a faster-moving student. Such a paradigm breaks the single-step on-policy feedback loop inherent in existing self-rewarding methods (Zhao et al., 2025b; Prabhudesai et al., 2025; Shafayat et al., 2025), raises the cost of exploiting trivial low-entropy shortcuts or spurious consensus, and offers a stable reward source without introducing an additional LLM or optimizer. In this way, it effectively overcomes reward hacking and prevents training collapse by implicitly seeking a temporal invariance for true reasoning.

**Co-rewarding-III: *Data-side + Model-side*.** Given that Co-rewarding-I and Co-rewarding-II provide two complementary perspectives for constructing stable self-supervised signals, a natural exploration is to integrate both data-side cross-supervision and model-side self-distillation into a unified instantiation. We introduce *Co-rewarding-III*, which leverages analogy-invariance between each original question and its rephrased counterparts while producing pseudo-labels from the EMA-updated reference teacher. Specifically, the teacher generates rollouts for both original and rephrased questions, and the resulting pseudo-label from one side is used to supervise the other. This combination further boosts the resistance of the reward signal to hacking, promoting more stable training dynamics.

Formally, its learning objective can be formulated as:

$$\mathcal{J}_{\text{Co-rewarding-III}}^{(k)}(\theta) = \mathbb{E}_{x\in\mathcal{D}, \underbrace{\{y_i\}_{i=1}^{G} \sim \pi_{\theta_{\text{old}}}^{(k)}(\cdot|x)}_{\text{policy student rollout from original question}}, x'\in\mathcal{D}', \underbrace{\{\tilde{y}_j'\}_{j=1}^{\tilde{G}} \sim \tilde{\pi}_{\text{ref}}^{(k)}(\cdot|x')}_{\text{reference teacher rollout from rephrased question}}} \mathcal{R}_\theta(\hat{A}^{(k)})$$

$$+ \mathbb{E}_{x'\in\mathcal{D}', \underbrace{\{y_i'\}_{i=1}^{G} \sim \pi_{\theta_{\text{old}}}^{(k)}(\cdot|x')}_{\text{policy student rollout from rephrased question}}, x\in\mathcal{D}, \underbrace{\{\tilde{y}_j\}_{j=1}^{\tilde{G}} \sim \tilde{\pi}_{\text{ref}}^{(k)}(\cdot|x)}_{\text{reference teacher rollout from original question}}} \mathcal{R}_\theta(\hat{A}'^{(k)}), \tag{12}$$

Table 1: **Main Results (%) of Co-rewarding and baselines trained on MATH.** Cell background colors indicate relative performance: darker colors denote better results within each model group. Additional results of Qwen2.5-3B/7B and Qwen3-1.7B-Base trained on MATH refer to Table 7.

| Training Set: MATH | Mathematics | | | | Code | | Instruction | Multi-Task |
|---|---|---|---|---|---|---|---|---|
| Methods | MATH500 | GSM8K | AMC | AIME24 | LiveCode | CRUX | IFEval | MMLU-Pro |
| *Qwen3-8B-Base* | | | | | | | | |
| Before RL | 72.4 | 27.82 | 20.93 | 3.75 | 23.41 | 54.75 | 50.89 | 52.92 |
| - GT-Reward (Shao et al., 2024) | 82.6 | 87.26 | 54.22 | 17.15 | 30.52 | 63.25 | 52.78 | 57.11 |
| - Self-Certainty (Zhao et al., 2025b) | 80.2 | 80.74 | 50.75 | 15.73 | 27.20 | 64.38 | 50.98 | 54.17 |
| - Entropy (Prabhudesai et al., 2025) | 80.2 | 87.19 | 49.54 | 15.63 | 29.38 | 62.00 | 51.81 | 54.86 |
| - Majority-Voting (Shafayat et al., 2025) | 79.8 | 89.76 | 49.09 | 15.83 | 30.52 | 63.38 | 51.80 | 56.93 |
| - Co-rewarding-I (Ours) | 81.2 | 93.70 | 51.20 | 15.10 | 30.81 | 66.00 | 55.79 | 59.95 |
| - Co-rewarding-II (Ours) | 80.8 | 92.42 | 53.46 | 14.48 | 30.23 | 62.83 | 60.70 | 57.50 |
| - Co-rewarding-III (Ours) | 81.4 | 90.98 | 54.07 | 13.33 | 30.71 | 63.75 | 53.69 | 59.10 |
| *Qwen3-4B-Base* | | | | | | | | |
| Before RL | 71.2 | 26.15 | 21.08 | 4.58 | 11.00 | 38.88 | 46.43 | 47.23 |
| - GT-Reward (Shao et al., 2024) | 78.6 | 89.76 | 51.20 | 15.00 | 26.07 | 55.38 | 47.80 | 53.96 |
| - Self-Certainty (Zhao et al., 2025b) | 71.6 | 71.79 | 38.86 | 11.67 | 22.37 | 57.00 | 48.15 | 48.93 |
| - Entropy (Prabhudesai et al., 2025) | 77.0 | 88.10 | 47.44 | 10.94 | 25.59 | 52.88 | 50.44 | 49.90 |
| - Majority-Voting (Shafayat et al., 2025) | 77.4 | 90.07 | 45.33 | 10.10 | 26.54 | 57.50 | 48.78 | 54.35 |
| - Co-rewarding-I (Ours) | 78.8 | 91.28 | 46.08 | 13.85 | 26.64 | 56.50 | 50.35 | 53.26 |
| - Co-rewarding-II (Ours) | 78.0 | 88.86 | 45.93 | 12.17 | 26.25 | 55.00 | 51.30 | 53.88 |
| - Co-rewarding-III (Ours) | 78.6 | 90.75 | 48.80 | 12.71 | 26.16 | 56.00 | 49.23 | 53.08 |
| *Llama-3.2-3B-Instruct* | | | | | | | | |
| Before RL | 39.2 | 65.73 | 10.54 | 3.75 | 9.86 | 25.37 | 57.32 | 31.14 |
| - GT-Reward (Shao et al., 2024) | 47.0 | 77.94 | 22.14 | 11.67 | 9.57 | 31.87 | 47.51 | 34.32 |
| - Self-Certainty (Zhao et al., 2025b) | 43.4 | 74.91 | 18.83 | 6.88 | 9.95 | 25.87 | 54.88 | 33.34 |
| - Entropy (Prabhudesai et al., 2025) | 43.4 | 66.19 | 20.18 | 6.56 | 11.66 | 24.62 | 54.70 | 33.52 |
| - Majority-Voting (Shafayat et al., 2025) | 46.8 | 78.77 | 20.48 | 9.27 | 11.00 | 31.25 | 47.96 | 33.18 |
| - Co-rewarding-I (Ours) | 50.2 | 79.45 | 23.80 | 10.00 | 11.28 | 29.88 | 48.89 | 33.77 |
| - Co-rewarding-II (Ours) | 49.8 | 79.30 | 22.59 | 10.73 | 10.80 | 30.63 | 49.90 | 33.61 |
| - Co-rewarding-III (Ours) | 51.6 | 79.91 | 25.45 | 10.42 | 10.43 | 32.50 | 46.37 | 34.50 |

where the first term supervises the original question via pseudo labels generated from its rephrased counterpart, and the second term, symmetrically, supervises the rephrased question via pseudo labels generated from the original question. The estimated advantages $\mathcal{R}_\theta(\hat{A}^{(k)})$ and $\mathcal{R}_\theta(\hat{A}'^{(k)})$ are computed in the similar way as in Co-rewarding-I and Co-rewarding-II. The reference teacher is also updated via EMA, as Eq. (11) in Co-rewarding-II. The other formulations and pseudo code of Co-rewarding-III are supplemented in Appendix B.1 and Algorithm 3.

**Remark 1.** Overall, the two instantiations of Co-rewarding embody our core idea from different perspectives: I leverages data-side analogy-invariance to provide cross supervision, while II employs model-side self-distillation to stabilize learning. Together, they reflect that stable self-supervised reasoning elicitation can emerge from both the diversity of data perspectives and the disentanglement of supervision signals. Co-rewarding-III further explores an orthogonally combined instantiation of these two sides. Moreover, Co-rewarding offers a flexible framework, in which key components, such as pseudo-labeling strategies, data rephrasing techniques, teacher model update rules, and policy optimization, can be seamlessly substituted with other advanced approaches (Yu et al., 2025).

## 4 EXPERIMENTS

### 4.1 SETUPS

**Backbone Models and Baselines.** We employ a diverse set of LLMs from different families and scales in our experiments, including the Qwen2.5 series (Qwen2.5-3B/7B) (Qwen et al., 2025), the Qwen3 series (Qwen3-1.7B/4B/8B-Base) (Yang et al., 2025), and the Llama3 series (Llama-3.2-3B-Instruct) (Meta, 2024). Beyond the vanilla GRPO that utilized the GT label for rewarding, we compare our Co-rewarding against several recent state-of-the-art (SoTA) self-reward reasoning approaches, denoted as Self-Certainty (Zhao et al., 2025b), Entropy (Prabhudesai et al., 2025) and Majority Voting (Shafayat et al., 2025). The details of all baselines are summarized in Appendix C.1.

**Implementation Details.** We implement our algorithms based on the VeRL framework (Sheng et al., 2024), and experiments are conducted on $4 \times$ H100-80GB GPUs. For our experiments, we totally use three training sets: MATH (Hendrycks et al., 2021) (7,500 questions), DAPO-14k (Yu et al., 2025) (en-version of DAPO-Math-17k, about 14.1k questions), and OpenRS (Dang & Ngo, 2025) (7,000 questions). During RL training, we use a global batch size of 128, set the number of

Table 2: **Main Results (%) of Co-rewarding and baselines trained on DAPO-14k**. Cell background colors indicate relative performance: darker colors denote better results within each model group. Additional Results of Qwen3-8B-Base and Qwen4-4B-Base trained on OpenRS refer to Table 8.

| Training Set: DAPO-14k | Mathematics | | | | Code | | Instruction | Multi-Task |
|---|---|---|---|---|---|---|---|---|
| Methods | MATH500 | GSM8K | AMC | AIME24 | LiveCode | CRUX | IFEval | MMLU-Pro |
| *Qwen3-8B-Base* | | | | | | | | |
| Before RL | 72.4 | 27.82 | 20.93 | 3.75 | 23.41 | 54.75 | 50.89 | 52.92 |
| - GT-Reward (Shao et al., 2024) | 86.6 | 87.19 | 61.75 | 24.58 | 30.52 | 63.75 | 53.11 | 60.27 |
| - Self-Certainty (Zhao et al., 2025b) | 82.0 | 77.63 | 49.85 | 15.00 | 27.77 | 60.75 | 50.58 | 54.24 |
| - Entropy (Prabhudesai et al., 2025) | 79.4 | 80.82 | 45.48 | 15.00 | 30.14 | 62.00 | 51.56 | 54.57 |
| - Majority-Voting (Shafayat et al., 2025) | 78.6 | 91.66 | 50.00 | 11.25 | 30.33 | 61.62 | 51.54 | 55.65 |
| - Co-rewarding-I (Ours) | 78.4 | 88.02 | 51.20 | 11.88 | 29.38 | 62.50 | 50.17 | 55.39 |
| - Co-rewarding-II (Ours) | 80.6 | 94.01 | 54.37 | 16.35 | 31.66 | 67.12 | 53.31 | 59.83 |
| - Co-rewarding-III (Ours) | 81.6 | 92.27 | 53.77 | 17.71 | 32.70 | 66.75 | 55.85 | 60.02 |
| *Qwen3-4B-Base* | | | | | | | | |
| Before RL | 71.2 | 26.15 | 21.08 | 4.58 | 11.00 | 38.88 | 46.43 | 47.23 |
| - GT-Reward (Shao et al., 2024) | 83.6 | 85.14 | 52.86 | 20.63 | 18.58 | 56.88 | 47.70 | 55.35 |
| - Self-Certainty (Zhao et al., 2025b) | 68.4 | 44.81 | 35.39 | 8.85 | 25.88 | 50.12 | 45.58 | 48.84 |
| - Entropy (Prabhudesai et al., 2025) | 76.6 | 82.79 | 43.37 | 12.81 | 26.35 | 50.75 | 48.20 | 50.22 |
| - Majority-Voting (Shafayat et al., 2025) | 73.4 | 64.06 | 40.81 | 9.17 | 26.16 | 53.00 | 48.91 | 51.06 |
| - Co-rewarding-I (Ours) | 73.8 | 75.89 | 43.83 | 10.63 | 26.25 | 50.12 | 46.84 | 51.51 |
| - Co-rewarding-II (Ours) | 77.8 | 91.89 | 48.49 | 14.27 | 26.64 | 54.87 | 48.90 | 52.83 |
| - Co-rewarding-III (Ours) | 79.2 | 90.45 | 48.95 | 15.10 | 27.58 | 54.87 | 50.30 | 54.79 |
| *Llama-3.2-3B-Instruct* | | | | | | | | |
| Before RL | 39.2 | 65.73 | 10.54 | 3.75 | 9.86 | 25.37 | 57.32 | 31.14 |
| - GT-Reward (Shao et al., 2024) | 49.4 | 78.17 | 25.90 | 9.17 | 10.33 | 31.37 | 53.10 | 33.83 |
| - Self-Certainty (Zhao et al., 2025b) | 42.4 | 74.71 | 17.32 | 4.79 | 11.18 | 28.38 | 54.50 | 33.51 |
| - Entropy (Prabhudesai et al., 2025) | 44.0 | 65.85 | 17.32 | 6.56 | 9.95 | 25.00 | 55.78 | 31.95 |
| - Majority-Voting (Shafayat et al., 2025) | 42.8 | 70.96 | 17.62 | 8.74 | 10.14 | 29.50 | 54.07 | 32.95 |
| - Co-rewarding-I (Ours) | 46.0 | 70.58 | 20.93 | 7.08 | 9.57 | 27.25 | 53.04 | 32.61 |
| - Co-rewarding-II (Ours) | 49.8 | 78.62 | 19.73 | 8.02 | 10.43 | 32.25 | 51.92 | 34.46 |
| - Co-rewarding-III (Ours) | 48.6 | 76.95 | 21.84 | 8.13 | 9.86 | 30.50 | 49.92 | 34.01 |

Figure 3: **Performance curves comparison** on validation set. *Top:* Qwen3-1.7B-Base and Qwen2.5-7B trained on the MATH set. *Bottom:* Qwen3-8B-Base and Llama-3.2-3B-Instruct trained on the DAPO-14k set.

Figure 4: **Reward** (left) and **response length** (right) of Qwen3-8B-Base and Llama-3.2-3B-Instruct trained on DAPO-14k. Due to different reward scale from other methods, Entropy reward is plotted on the right $y$-axis of left panels, where the reward is the negative entropy.

rollouts to $G = \tilde{G} = 8$ per question for Co-rewarding-I, II and III, and adopt AdamW with a learning rate of $3 \times 10^{-6}$. In Co-rewarding-I and III, question rephrasing is performed by the open-source Qwen3-32B model. In Co-rewarding-II and III, the EMA weight is scheduled from $\alpha_{\text{start}} = 0.99$ to $\alpha_{\text{end}} = 0.9999$ using cosine annealing. More implementation details are reported in Appendix C.2.

**Evaluation Details.** To provide a comprehensive evaluation of model capabilities, we utilize a diverse set of benchmarks spanning mathematical reasoning, code generation, instruction-following, and general multi-task abilities. Specifically: (1) Mathematical reasoning: MATH500 (Lightman et al., 2024), GSM8K (Cobbe et al., 2021), AMC (Li et al., 2024a), and AIME24 (Zhang & Math-AI, 2024). (2) Code generation: LiveCodeBench (Jain et al., 2025) release_v6 and CRUX (Gu et al., 2024). (3) Instruction-following and multi-task abilities: IFEval (Zhou et al., 2023b) and MMLU-Pro (Wang et al., 2024). Additional evaluation details are provided in Appendix C.3.

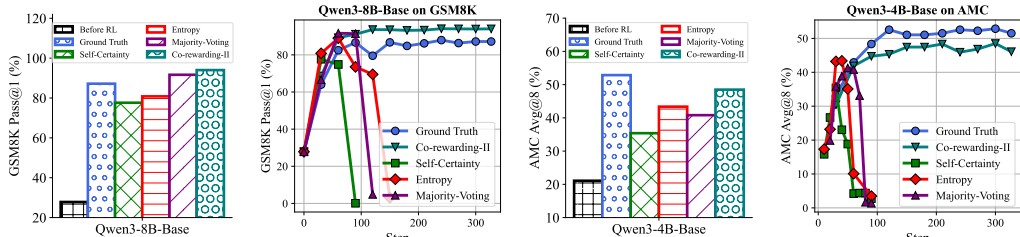

Figure 5: **Performance and Stability on GSM8K and AMC.** The gains of Co-rewarding arise from its training stability, which supports continuous improvements throughout learning.

Table 3: **Ablation study of Co-rewarding.** For Co-rewarding-I, ablations train only on original or rephrased data. For Co-rewarding-II, ablation removes EMA updates of the reference teacher.

| Training Set | Methods | MATH500 | GSM8K | AMC | AIME24 | LiveCode | CRUX | IFEval | MMLU-Pro |
|---|---|---|---|---|---|---|---|---|---|
| | | | | | *Qwen3-8B-Base* | | | | |
| | Co-rewarding-I | **81.2** | **93.70** | **51.20** | 15.10 | 30.81 | **66.00** | **55.79** | **59.95** |
| | - Majority-Voting w/ Union | 80.2 | 93.48 | 49.70 | 15.63 | **31.94** | 64.88 | 54.25 | 59.51 |
| | - Majority-Voting w/ Original | 79.8 | 89.76 | 49.09 | **15.83** | 30.52 | 63.38 | 51.80 | 56.93 |
| | - Majority-Voting w/ Rephrased | 79.2 | 91.51 | 50.75 | 14.17 | 31.66 | 60.38 | 52.24 | 57.26 |
| **MATH** | Co-rewarding-II | 80.8 | 92.42 | 53.46 | 14.48 | 30.23 | 62.83 | 60.70 | 57.50 |
| | - w/o Updating Reference | 79.2 | 89.46 | 51.51 | 13.96 | **30.62** | 61.75 | 56.93 | 51.85 |
| | | | | | *Llama-3.2-3B-Instruct* | | | | |
| | Co-rewarding-I | **50.2** | 79.45 | **23.80** | **10.00** | **11.28** | 29.88 | **48.89** | **33.77** |
| | - Majority-Voting w/ Union | 48.0 | **80.52** | 21.84 | 9.69 | 10.14 | 30.00 | 43.35 | 34.05 |
| | - Majority-Voting w/ Original | 46.8 | 78.77 | 20.48 | 9.27 | 11.00 | **31.25** | 47.96 | 33.18 |
| | - Majority-Voting w/ Rephrased | 44.0 | 78.85 | 21.23 | 8.85 | 10.04 | 17.25 | 47.84 | 33.72 |
| | Co-rewarding-II | **49.8** | 79.30 | 22.59 | 10.73 | 10.80 | 30.63 | 49.90 | 33.61 |
| | - w/o Updating Reference | 47.0 | 78.92 | 22.29 | 9.06 | 5.50 | **31.25** | 47.88 | 33.32 |
| | | | | | *Qwen3-8B-Base* | | | | |
| **DAPO-14k** | Co-rewarding-II | 80.6 | 94.01 | 54.37 | 16.35 | 31.66 | 67.12 | 53.31 | 59.83 |
| | - w/o Updating Reference | 78.0 | 88.40 | 51.66 | 15.94 | 30.62 | 63.75 | 52.48 | 58.01 |
| | | | | | *Llama-3.2-3B-Instruct* | | | | |
| | Co-rewarding-II | **49.8** | 78.62 | 19.73 | 8.02 | 10.43 | 32.25 | 51.92 | **34.46** |
| | - w/o Updating Reference | 45.0 | 76.72 | 17.92 | 8.02 | 10.05 | 30.63 | 51.33 | 33.94 |

## 4.2 EXPERIMENTAL RESULTS

### 4.2.1 MAIN PERFORMANCE OF CO-REWARDING

**Superior Performance of Co-rewarding over self-rewarding baselines.** Table 1 and Table 2 report the experimental results trained on MATH and DAPO-14k, respectively. We observe that all three Co-rewarding instantiations (I, II, and III) occupy more darker cells in the tables, demonstrating stronger performance than other self-rewarding SoTA baselines. Specifically, Co-rewarding-I achieves an average relative performance gain of $+4.42\%$ over the best baselines across four mathematical benchmarks and models in Table 1, while Co-rewarding-II achieves a larger average relative gain of $+12.90\%$ in Table 2. Moreover, Co-rewarding-III achieves improvements on average of $+7.11\%$ and $1.72\%$ over Co-rewarding-I and Co-rewarding-II, respectively, suggesting that integrating data-side cross-supervision with model-side self-distillation can further boost performance. Additional results on other training sets and LLMs are provided in Appendix D.1.

**Surpassing GT-Reward on certain benchmarks.** Surprisingly, we observe that all three Co-rewarding instantiations (I, II, and III) outperform GT-Reward in certain cases. For example, on GSM8K, they together achieve an average improvement of $+2.77\%$ over GT-Reward in Table 1, while Co-rewarding-II further delivers a larger gain of $+5.44\%$ in Table 2. Notably, Co-rewarding-II reaches a remarkably high Pass@1 of 94.01% with Qwen3-8B-Base. This may be because GSM8K is a relatively easier benchmark, where self-supervised RL is sufficient to elicit latent reasoning abilities without GT labels. Additionally, Co-rewarding also shows advantages on coding benchmark CRUX in several cases, possibly due to distribution mismatch between training and evaluation data. This may offer opportunities for self-supervised methods to generalize on par with, or even surpass GT-supervised methods in some cases. These findings highlight the potential of self-supervised RL to elicit reasoning capabilities, particularly with Co-rewarding mitigating training collapse.

**Code generalization with preserved general performance.** Although trained solely on math-oriented datasets, the models show improvements on coding benchmarks, suggesting a cross-domain generalization from math to code in self-supervised reasoning elicitation. Moreover, Co-rewarding preserves general instruction-following and multi-task ability on MMLU-Pro and IFEval. As shown

Table 4: **Detailed performance of MMLU-Pro** with Qwen3-8B-Base trained on DAPO-14k. More results refer to Appendix D.10.

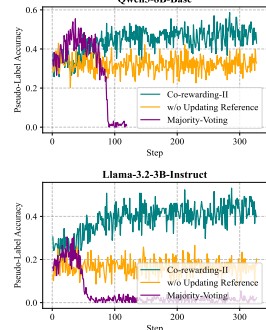

| *MMLU-Pro (Qwen3-8B-Base)* | | | | | | |
|---|---|---|---|---|---|---|
| Methods | biology | business | chemistry | computer sci. | economics | health | history |
| - GT-Reward | 77.96 | 70.85 | 60.42 | 61.95 | 71.33 | 59.79 | 51.44 |
| - Self-Certainty | 75.73 | 58.05 | 50.53 | 56.83 | 69.31 | 54.77 | 50.40 |
| - Entropy | 74.76 | 59.70 | 51.33 | 56.10 | 67.90 | 55.87 | 48.04 |
| - Majority-Voting | 75.32 | 61.47 | 54.24 | 58.29 | 69.67 | 58.20 | 49.34 |
| - Co-rewarding-I | 76.85 | 61.22 | 53.45 | 59.02 | 66.82 | 55.62 | 48.29 |
| - Co-rewarding-II | 76.71 | 68.69 | 64.58 | 61.71 | 68.25 | 56.85 | 51.71 |
| Methods | law | math | other | philosophy | physics | psychology | engineering |
| - GT-Reward | 31.52 | 73.28 | 56.28 | 52.71 | 61.97 | 67.30 | 46.14 |
| - Self-Certainty | 30.43 | 63.06 | 51.63 | 46.29 | 51.73 | 66.42 | 41.07 |
| - Entropy | 28.97 | 63.96 | 51.51 | 48.90 | 53.04 | 66.80 | 42.32 |
| - Majority-Voting | 31.16 | 64.62 | 52.27 | 48.90 | 53.27 | 66.92 | 40.97 |
| - Co-rewarding-I | 30.34 | 66.17 | 51.73 | 48.90 | 55.19 | 66.42 | 39.63 |
| - Co-rewarding-II | 31.16 | 72.17 | 52.49 | 52.10 | 63.21 | 68.17 | 47.16 |

Figure 6: **Pseudo label accuracy comparison.**

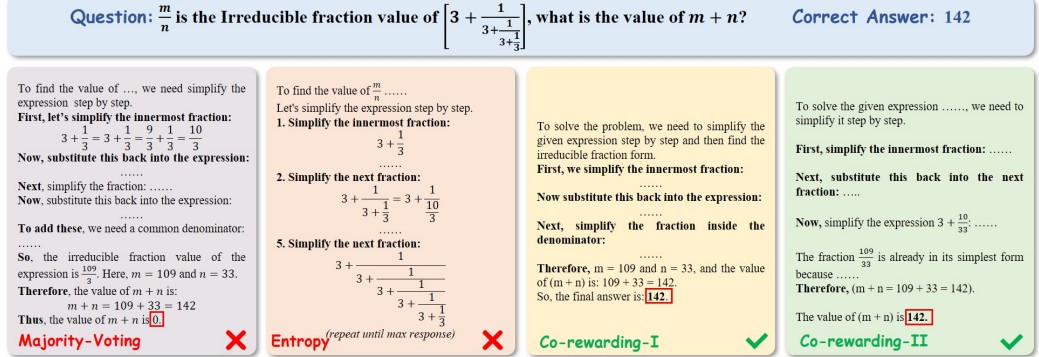

Figure 7: **Case study**: An example comparing the generations from *Majority-Voting*, *Entropy*, and our proposed *Co-rewarding*. The results clearly reveal the reward hacking behavior exhibited by Majority-Voting and Entropy, while ours generate the correct answer. Full results refer to Appendix D.13.

in Table 4, Co-rewarding-II outperforms other self-rewarding baselines in 12 of 14 MMLU-Pro categories, demonstrating that its gains do not come at the expense of broader general-domain performance. More detailed results of MMLU-Pro and IFEval refer to Appendix D.10 and D.11.

**Importance of stability for performance gain.** As shown in Table 2, self-rewarding baselines exhibit noticeably limited performance gain in certain cases, such as Self-Certainty with Qwen3-4B-Base on GSM8K. Figure 5 further reflects this by showing that baselines improve quickly at the beginning but soon collapse on GSM8K and AMC, whereas Co-rewarding sustains steady progress. This collapse restricts the baselines to effective training on only a small portion of the data, preventing further improvements with continued training. These observations underscore the importance of avoiding training collapse in self-supervised RL to unlock further performance gains.

### 4.2.2 FURTHER ANALYSIS

**Co-rewarding alleviates collapse and provides stable self-supervised RL.** We use 5,000 questions from the MATH test split as a validation set to monitor training process. Figure 3 shows that all three self-rewarding baselines collapse on both MATH and DAPO-14k. Co-rewarding-I remains stable on MATH but still collapses on DAPO-14k, suggesting that its stability depends on the property of training data. A plausible explanation is that the questions in MATH may provide favorable conditions for promoting diverse rephrasing variability, which is beneficial for the effectiveness of contrastive agreement in Co-rewarding-I. More discussions are provided in Appendix D.7. In contrast, Co-rewarding-II consistently maintains stability across datasets, as its design decouples supervision from the online policy and thus breaks the entanglement between supervision and the policy itself.

**Co-rewarding attempts to balance exploration-exploitation.** Figure 4 shows reward and response length curves. Entropy and Majority-Voting quickly reach the highest reward, indicating reward hacking rather than genuine reasoning improvement. In contrast, GT-Reward and Co-rewarding exhibit smoother, gradually increasing rewards, reflecting stable training. The response length curves further illustrate this difference: GT-Reward lengthens responses to explore correct reasoning paths; Majority-Voting collapses to short outputs, restricting exploration; and Entropy collapses its probability mass onto a small set of tokens, repeatedly generating them until truncation. Co-

rewarding instead maintains moderate response lengths throughout training, suggesting a balanced exploration–exploitation trade-off. Additional curves for other LLMs are provided in Appendix D.2.

**Each part contributes to Co-rewarding.** Table 3 summarizes the ablations across two training sets. For Co-rewarding-I, we observe that it typically outperforms all three variants of Majority-Voting: models trained only on original questions, only on rephrased questions, or on their union. This indicates that the cross-supervision between original and rephrased questions plays a key role in mitigating training collapse, whereas simply adding more data does not resolve the inherent instability of single-view self-rewarding methods. Notably, training only on the original or rephrased data yields comparable results, reflecting that the quality of original and rephrased data is similar. For Co-rewarding-II, removing the EMA update of the reference teacher model causes clear degradation, highlighting the necessity of teacher updates for improving pseudo-label quality.

**EMA is essential in Co-rewarding-II for improving pseudo-label quality.** Figure 6 compares pseudo-label accuracy across Co-rewarding-II, "w/o Updating Reference", and Majority-Voting. Co-rewarding-II steadily improves accuracy as training progresses, while "w/o Updating Reference" remains nearly flat, underscoring the role of EMA updates in allowing the teacher to co-evolve with the policy and generate higher-quality pseudo labels. By contrast, Majority-Voting briefly improves but then collapses to near zero, evidencing reward hacking through consistent yet incorrect outputs.

**Case Study of the model reasoning with different learning methods.** Figure 7 provides a concrete example to illustrate the qualitative difference between self-rewarding baselines and our Co-rewarding. Majority-Voting exhibits reward hacking by boxing an incorrect answer "0" to pursue consensus, even though the reasoning steps are correct. Entropy produces repetitive outputs as its decoding probability distribution collapses onto a narrow set of tokens during entropy minimization. In contrast, Co-rewarding generates coherent step-by-step reasoning and correctly boxes the final answer, showing its capacity to overcome reward hacking and elicit genuine reasoning. Full results are provided in Appendix D.13 and additional case studies on code benchmark are discussed in Appendix D.14.

## 5 RELATED WORK

Reinforcement learning with verifiable reward (RLVR) has recently become a mainstream post-training paradigm for eliciting strong reasoning abilities in LLMs (Guo et al., 2025), achieving remarkably encouraging success particularly on mathematical (Shao et al., 2024) and coding (Luo et al., 2025) tasks. However, RLVR fundamentally depends on high-quality and annotated GT labels to supervise reward signals, which remains a major bottleneck for scalability under the spirit of the scaling laws. To break this limitation, recent efforts have explored RL without external reward from multiple perspectives. For instance, methods such as TTRL (Zuo et al., 2025) and SRT (Shafayat et al., 2025) pursue self-consistency to generate pseudo labels for rewards, where agreement among multiple rollouts is treated as optimization objective. Additionally, another technical line such as EMPO (Zhang et al., 2025d), Intuitor (Zhao et al., 2025b) and RENT (Prabhudesai et al., 2025), enhances the LLM confidence by optimizing internal signals of reasoning, such as entropy minimization or self-certainty maximization. Different from these studies, Co-rewarding focuses on mitigating inherent training collapse in existing methods and enables stable self-supervised RL training. More detailed discussions of related work are in Appendix A.

## 6 CONCLUSION

In this work, we introduced Co-rewarding, a self-supervised RL framework that elicits the reasoning capability of LLMs through complementary supervision. Unlike prior self-rewarding methods that entangle rewards with single-view outputs and risk collapse, Co-rewarding establishes stability by decoupling the reward signal from the current online policy with the single-view output. Specifically, Co-rewarding-I leverages contrastive agreement across semantically analogous questions; Co-rewarding-II employs a dynamically updated teacher to provide insulated pseudo-labels; and Co-rewarding-III combines the data-side cross-supervision from Co-rewarding-I and the model-side teacher-based pseudo labels from Co-rewarding-II to further boost performance. Together, these designs construct cross-referable reward signals without explicit labels, aligning RL with invariances in reasoning rather than the mere correctness of isolated outputs. We hope this work will inspire further exploration into self-supervised RL for reasoning to advance the development.

ACKNOWLEDGMENTS

ZZZ, JNZ, ZKZ, XL, XF and BH were supported by NSFC Major Research Plan No. 92570109.

ETHICS STATEMENT

This work complies with the Code of Ethics. It uses only publicly available datasets, involves no human or sensitive data, and raises no foreseeable risks related to privacy, security, or fairness issues. The research is conducted solely for scientific advancement, with no conflicts of interest.

REPRODUCIBILITY STATEMENT

We are committed to ensure the reproducibility of our proposed method. A detailed description of our approach is provided in the Co-rewarding Framework section, and the corresponding source code has been submitted in an anonymous repository at https://github.com/tmlr-group/Co-rewarding. Both backbone models and datasets used in our work are publicly available. Furthermore, all parameters, hyper-parameters, and procedural steps required to reproduce our results are thoroughly recorded in the Implementation Details. We believe that these components provide the community with details necessary to verify and reproduce our work.

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

## LLM USAGE STATEMENT

Here we clarify the usage of Large Language Models (LLMs) in this work. For the preparation of this paper, LLMs are limited to the role of a general-purpose writing assistant and are not used for research ideation or core content generation. For research methodology, LLM is a core component of our proposed method. Specifically, we utilize the Qwen3-32B model to perform question rephrasing in Co-rewarding-I, which is thoroughly detailed in the Implementation Details section of the main paper. The authors take full responsibility for all content written under their name.

## A RELATED WORK

**Large Language Model Reasoning.** LLMs have shown impressive performance on vast tasks that require reasoning, including solving mathematical problems, writing code, and answering logical questions. One of the key techniques that has improved LLM reasoning is Chain-of-Thought (CoT) prompting (Wei et al., 2022; Zhou et al., 2024; 2025b). CoT encourages the model to generate intermediate reasoning steps before producing the final answer (Zhou et al., 2025c), which has been shown to enhance performance on tasks like arithmetic, commonsense reasoning, and symbolic reasoning. Subsequent work has extended CoT by integrating it with various strategies, including compositional generalization (Zhou et al., 2023a; Khot et al., 2023) and employing structural reasoning approaches (Yao et al., 2023a; Besta et al., 2024; Yang et al., 2024). In addition, CoT serves as a fundamental framework for techniques like fine-tuninig (Zelikman et al., 2022), argentic workflow (Yao et al., 2023b), and paving the way for inference-time scaling (Snell et al., 2024).

**RL for Large Language Models.** Several RL algorithms have been developed primarily for alignment tasks (Wang et al., 2025b; Zhang et al., 2026; 2025a). Specifically, DPO (Rafailov et al., 2023), CPO (Xu et al., 2024), and their variants (Li et al., 2024b; Guo et al., 2024; Munos et al., 2024; Hong et al., 2024; Xie et al., 2024) rely on preference pairs labeled by annotators. In contrast, KTO (Ethayarajh et al., 2024) and BCO (Jung et al., 2024) require only a single binary label (like or dislike) for each output. Besides, PRM (Uesato et al., 2022; Lightman et al., 2024) and Step-KTO (Lin et al., 2025a) offer step-by-step guidance by incorporating feedback at each reasoning step rather than focusing solely on the final outputs. Recently, the follow-up work of GRPO improves the optimization objective, *e.g.*, DAPO (Yu et al., 2025), Dr. GRPO (Liu et al., 2025a), REINFORCE++ (Hu, 2025), CPPO (Lin et al., 2025b), and GPG (Chu et al., 2025). Another line of research generalizes GRPO to broader applications such as multimodal reasoning (Zhou et al., 2025a; Huang et al., 2025; Chu et al., 2025; Liu et al., 2025b; Zhang et al., 2025b) and logical reasoning (Xie et al., 2025).

**RL without External Reward.** RL methods have shown promising scaling capabilities to enhance the reasoning abilities of LLMs (Guo et al., 2025), yet they are often limited by the availability of training data for reward signals (Gao et al., 2023; Liu et al., 2023). Notably, Wang et al. (Wang et al., 2025a) demonstrate that RL can effectively bootstrap LLM reasoning with as little as a single training example, highlighting the potential to minimize or even eliminate reliance on external reward signals during training. Recent efforts leverage distinct strategies for reward assignment. For instance, SIRLC (Pang et al., 2024) and AZR (Zhao et al., 2025a) utilize an LLM-as-the-judge approach to assign rewards. In contrast, methods like SRT, TTRL, and their variants (Shafayat et al., 2025; Zuo et al., 2025; Fang et al., 2025; Zhang et al., 2025c) employ self-consistency (Wang et al., 2022) to generate pseudo-rewards, reducing dependence on external annotations. Meanwhile, INTUITOR, RLSC, and RENT (Zhao et al., 2025b; Li et al., 2025; Prabhudesai et al., 2025) harness the internal confidence scores of LLMs as intrinsic reward signals. Additionally, EMPO and its variants (Zhang et al., 2025d; Agarwal et al., 2025) promote reasoning by minimizing entropy of reasoning path, further diversifying the approaches to incentivize robust LLM performance (Han et al., 2025).

---

**Algorithm 1** *Co-rewarding-I*

---

1: **Input:** policy model $\pi_\theta$, learning rate $\eta$, training dataset $\mathcal{D}$, rephrased training dataset $\mathcal{D}'$, total iterations $K$.
2: **Output:** trained policy model $\pi_\theta$.
3: **for all** iteration $k = 1, \ldots, K$ **do**
4:      Sample mini-batch inputs $\mathcal{B} \subseteq \mathcal{D}$ and $\mathcal{B}' \subseteq \mathcal{D}'$.
5:      **for all** input question $x \in \mathcal{B}$ and $x' \in \mathcal{B}'$ **do**
6:          Sample rollouts $\{y_i\}_{i=1}^{G} \sim \pi_{\theta_{\text{old}}}(\cdot \mid x)$.
7:          Sample rollouts $\{y_i'\}_{i=1}^{G'} \sim \pi_{\theta_{\text{old}}}(\cdot \mid x')$.
8:          Obtain pseudo labels by Eq. (8).
9:          Estimate relative advantages by Eq. (7).
10:          Compute the objective by Eq. (6).
11:          Update $\theta \leftarrow \theta - \eta \nabla_\theta \mathcal{J}_{\text{Co-rewarding-I}}(\theta)$.
12:      **end for**
13: **end for**

---

**Algorithm 2** *Co-rewarding-II*

---

1: **Input:** policy model $\pi_\theta$, learning rate $\eta$, training dataset $\mathcal{D}$, total iterations $K$.
2: **Output:** trained policy model $\pi_\theta$.
3: **for** iteration $k = 1, \ldots, K$ **do**
4:      Sample mini-batch $\mathcal{B} \subseteq \mathcal{D}$.
5:      **for all** $x \in \mathcal{B}$ **do**
6:          Sample rollouts $\{y_i\}_{i=1}^{G} \sim \pi_{\theta_{\text{old}}}^{(k)}(\cdot \mid x)$.
7:          Update the reference teacher by Eq. (11).
8:          Sample rollouts $\{\tilde{y}_j\}_{j=1}^{\tilde{G}} \sim \tilde{\pi}_{\text{ref}}^{(k)}(\cdot \mid x)$.
9:          Obtain pseudo label from $\{\tilde{y}_j\}_{j=1}^{\tilde{G}}$ by Eq. (10).
10:          Estimate the relative advantage by Eq. (10).
11:          Compute the objective by Eq. (9).
12:          Update $\theta \leftarrow \theta - \eta \nabla_\theta \mathcal{J}_{\text{Co-rewarding-II}}^{(k)}(\theta)$.
13:      **end for**
14: **end for**

---

**Algorithm 3** *Co-rewarding-III*

---

1: **Input:** policy model $\pi_\theta$, learning rate $\eta$, original training dataset $\mathcal{D}$, rephrased training dataset $\mathcal{D}'$, total iterations $K$.
2: **Output:** trained policy model $\pi_\theta$.
3: **for** iteration $k = 1, \ldots, K$ **do**
4:      Sample mini-batch inputs $\mathcal{B} \subseteq \mathcal{D}$ and $\mathcal{B}' \subseteq \mathcal{D}'$.
5:      **for all** $x \in \mathcal{B}$ and $x' \in \mathcal{B}'$ **do**
6:          Sample rollouts $\{y_i\}_{i=1}^{G} \sim \pi_{\theta_{\text{old}}}^{(k)}(\cdot \mid x)$ and $\{y_i'\}_{i=1}^{G} \sim \pi_{\theta_{\text{old}}}^{(k)}(\cdot \mid x')$.
7:          Update the reference teacher by Eq. (11).
8:          Sample rollouts $\{\tilde{y}_j\}_{j=1}^{\tilde{G}} \sim \tilde{\pi}_{\text{ref}}^{(k)}(\cdot \mid x)$ and $\{\tilde{y}_j'\}_{j=1}^{\tilde{G}} \sim \tilde{\pi}_{\text{ref}}^{(k)}(\cdot \mid x')$.
9:          Obtain pseudo label from $\{\tilde{y}_j'\}_{j=1}^{\tilde{G}}$ and $\{\tilde{y}_j\}_{j=1}^{\tilde{G}}$ by Eq. (13) and Eq. (14).
10:          Estimate the relative advantages by Eq. (13) and Eq. (14).
11:          Compute the objective by Eq. (12).
12:          Update $\theta \leftarrow \theta - \eta \nabla_\theta \mathcal{J}_{\text{Co-rewarding-III}}^{(k)}(\theta)$.
13:      **end for**
14: **end for**

---

# B PSEUDO CODE OF CO-REWARDING

## B.1 FORMULATION OF CO-REWARDING-III

The relative advantages $\mathcal{R}_\theta(\hat{A}^{(k)})$ and $\mathcal{R}_\theta(\hat{A}'^{(k)})$ are computed as:

$$\hat{A}_i^{(k)} = \frac{r(\tilde{y}_{\mathrm{v}}'^{(k)}, y_i) - \mathrm{mean}(\{r(\tilde{y}_{\mathrm{v}}'^{(k)}, y_i)\}_{i=1}^G)}{\mathrm{std}(\{r(\tilde{y}_{\mathrm{v}}'^{(k)}, y_i)\}_{i=1}^G)}, \ \tilde{y}_{\mathrm{v}}'^{(k)} = \arg\max_{y*} \sum_{j=1}^{\tilde{G}} \mathbf{1}[\mathrm{ans}(\tilde{y}_j') = \mathrm{ans}(y*)], \quad (13)$$

$$\hat{A}_i'^{(k)} = \frac{r(\tilde{y}_{\mathrm{v}}^{(k)}, y_i') - \mathrm{mean}(\{r(\tilde{y}_{\mathrm{v}}^{(k)}, y_i')\}_{i=1}^G)}{\mathrm{std}(\{r(\tilde{y}_{\mathrm{v}}^{(k)}, y_i')\}_{i=1}^G)}, \ \tilde{y}_{\mathrm{v}}^{(k)} = \arg\max_{y*} \sum_{j=1}^{\tilde{G}} \mathbf{1}[\mathrm{ans}(\tilde{y}_j) = \mathrm{ans}(y*)], \quad (14)$$

where the pseudo label $\tilde{y}_{\mathrm{v}}'^{(k)}$ is the majority-vote pseudo label obtained from reference rollouts on the rephrased question, and $\tilde{y}_{\mathrm{v}}^{(k)}$ is the corresponding pseudo label obtained from reference rollouts on original question. The reference model is slowly updated via EMA as in Eq. (11).

## B.2 PSEUDO CODE

To intuitively present the pipeline of Co-rewarding, we summarize the pseudo codes of Co-rewarding-I, Co-rewarding-II and Co-rewarding-III in Algorithm 1, Algorithm 2 and Algorithm 3, respectively.

# C ADDITIONAL EXPERIMENTAL DETAILS

## C.1 DETAILS OF BASELINES

We compare our proposed Co-rewarding-I and II against GT-reward and several recent state-of-the-art (SoTA) self-reward approaches:

- **GT-Reward** (Shao et al., 2024): Originally introduced by DeepSeek-R1 (Guo et al., 2025), GT-Reward supervises training using ground-truth (GT) answers, determining whether model rollouts are correct or not, to guide RL optimization.
- **Self-Certainty** (Zhao et al., 2025b): This method maximizes *self-certainty*, defined as the KL-divergence between the uniform distribution and the model's decoding distribution, serving as reward to encourage more confident predictions.
- **Entropy** (Prabhudesai et al., 2025): This method minimizes the entropy of the model's rollout distribution, using negative entropy as reward to maximize model confidence.
- **Majority-Voting** (Shafayat et al., 2025): By generating multiple rollouts per question, Majority-Voting selects the most frequent answer as a pseudo-label to supervise training.

For all methods, we adopt the widely used GRPO as the policy optimization algorithm.

## C.2 MORE IMPLEMENTATION DETAILS

The detailed training configurations are summarized in Table 5, and all baseline methods are trained under the same setup for fairness. For the training system prompt, we adopt the official default prompt provided by VeRL[1], shown below:

```
Let's think step by step and output the final answer within \boxed{}.
```

In addition, the semantically analogical questions used in Co-rewarding-I are generated by Qwen3-32B through a rewriting prompt. The exact rewriting instruction is provided as follows:

```
You are given a math problem. Please rewrite it using different wording
    and a different real-world scenario, while keeping the underlying
    mathematical meaning and answer exactly the same.
```

---

[1]https://github.com/volcengine/verl

Table 5: **Detailed training settings.**

| Settings | Co-rewarding-I | Co-rewarding-II | Co-rewarding-III |
|---|---|---|---|
| Batch Size | 128 | 128 | 128 |
| Max Prompt Length | 512 | 512 | 512 |
| Max Response Length | 3,072 | 3,072 | 3,072 |
| Train Steps | 170-220 (MATH), 300-330 (DAPO-14k), 100-130 (Open-RS) | | |
| Learning Rate | 3e-6 | 3e-6 | 3e-6 |
| # Policy Rollout $G$ | 8 | 8 | 8 |
| # Reference Rollout $\tilde{G}$ | - | 8 | 8 |
| Clip Ratio | 0.2 | 0.2 | 0.2 |
| Warmup Style | Cosine | Cosine | Cosine |
| Warmup Steps Ratio | 0.1 | 0.1 | 0.1 |
| KL Loss Coefficient | 0.005 | 0.001 | 0.001 |
| Optimizer | AdamW ($\beta_1 = 0.9, \beta_2 = 0.999, \epsilon = 10^{-8}$) | | |
| Training Temperature | 1.0 | 1.0 | 1.0 |
| Evaluation Temperature | 0.8 | 0.8 | 0.8 |
| EMA $\alpha_{\text{start}}$ | - | 0.99 | 0.99 |
| EMA $\alpha_{\text{end}}$ | - | 0.9999 | 0.9999 |

Table 6: **Statistics and usages of datasets** used in our experiments.

| Dataset Name | # Data Size | Usage |
|---|---|---|
| MATH-Train (Hendrycks et al., 2021) | 7,500 | Training Set |
| MATH-Test (Hendrycks et al., 2021) | 5,000 | Validation Set |
| DAPO-14k (Yu et al., 2025) | 14,109 | Training Set |
| Open-RS (Dang & Ngo, 2025) | 7,000 | Training Set |
| MATH500 (Lightman et al., 2024) | 500 | Evaluation Benchmark |
| GSM8K (Cobbe et al., 2021) | 1,319 | Evaluation Benchmark |
| AMC (Li et al., 2024a) | 83 | Evaluation Benchmark |
| LiveCodeBench (Jain et al., 2025) | 1,055 | Evaluation Benchmark |
| CRUX (Gu et al., 2024) | 800 | Evaluation Benchmark |
| MMLU-Pro (Wang et al., 2024) | 12,032 | Evaluation Benchmark |
| IFEval (Zhou et al., 2023b) | 541 | Evaluation Benchmark |

```
Guidelines:
1. Do not change the math logic or the final answer.
2. Use different words and a new context to make it look like a different
     problem.
3. Avoid copying phrases or sentence structures from the original.
4. Make sure the rewritten question is natural, clear, and solvable.
5. Output ONLY between the following markers, and strictly in this format
     (no extra explanation):

### RESULT_START
ORIGINAL:
<original question>
REWRITE:
<rewritten question>
### RESULT_END
```

Table 7: **Supplement Results (%) of Co-rewarding and baselines trained on MATH**. Cell background colors: darker colors denote better results within each model group.

| Training Set: MATH | Mathematics | | | | Code | | Instruction | Multi-Task |
| --- | --- | --- | --- | --- | --- | --- | --- | --- |
| **Methods** | **MATH500** | **GSM8K** | **AMC** | **AIME24** | **LiveCode** | **CRUX** | **IFEval** | **MMLU-Pro** |
| *Qwen2.5-3B* | | | | | | | | |
| Before RL | 53.6 | 19.48 | 10.69 | 0.52 | 9.95 | 18.50 | 29.83 | 32.50 |
| - GT-Reward (Shao et al., 2024) | 65.4 | 82.18 | 32.98 | 6.77 | 13.93 | 32.12 | 33.66 | 36.74 |
| - Self-Certainty (Zhao et al., 2025b) | 64.2 | 80.52 | 28.92 | 5.00 | 10.90 | 29.00 | 32.22 | 33.88 |
| - Entropy (Prabhudesai et al., 2025) | 63.2 | 80.44 | 29.67 | 5.94 | 9.05 | 29.00 | 32.94 | 35.35 |
| - Majority-Voting (Shafayat et al., 2025) | 64.6 | 82.41 | 33.13 | 5.10 | 14.03 | 36.38 | 35.19 | 35.50 |
| - Co-rewarding-I (Ours) | 65.4 | 84.53 | 30.57 | 5.31 | 16.40 | 36.88 | 33.86 | 36.38 |
| - Co-rewarding-II (Ours) | 65.2 | 81.72 | 32.38 | 4.47 | 22.25 | 40.25 | 32.74 | 30.79 |
| *Qwen2.5-7B* | | | | | | | | |
| Before RL | 69.4 | 24.71 | 15.81 | 2.81 | 3.79 | 26.38 | 38.19 | 44.76 |
| - GT-Reward (Shao et al., 2024) | 76.4 | 88.02 | 45.63 | 14.06 | 15.92 | 45.12 | 41.49 | 41.12 |
| - Self-Certainty (Zhao et al., 2025b) | 72.8 | 84.31 | 38.55 | 8.75 | 12.04 | 54.12 | 37.24 | 43.30 |
| - Entropy (Prabhudesai et al., 2025) | 72.2 | 81.43 | 39.61 | 10.73 | 16.49 | 51.88 | 40.33 | 42.79 |
| - Majority-Voting (Shafayat et al., 2025) | 74.4 | 84.53 | 40.96 | 11.04 | 15.45 | 51.00 | 38.60 | 43.35 |
| - Co-rewarding-I (Ours) | 74.6 | 89.61 | 41.27 | 10.73 | 15.73 | 55.58 | 42.86 | 40.51 |
| - Co-rewarding-II (Ours) | 73.6 | 89.31 | 42.77 | 11.98 | 8.25 | 47.50 | 41.82 | 37.45 |
| *Qwen3-1.7B-Base* | | | | | | | | |
| Before RL | 57.0 | 19.56 | 8.43 | 1.15 | 4.45 | 7.50 | 33.65 | 33.00 |
| - GT-Reward (Shao et al., 2024) | 69.6 | 81.57 | 35.54 | 8.23 | 13.74 | 35.25 | 36.16 | 39.12 |
| - Self-Certainty (Zhao et al., 2025b) | 58.2 | 40.25 | 23.04 | 3.02 | 9.86 | 18.00 | 32.96 | 35.13 |
| - Entropy (Prabhudesai et al., 2025) | 63.6 | 71.79 | 31.63 | 6.88 | 13.74 | 31.37 | 35.37 | 36.67 |
| - Majority-Voting (Shafayat et al., 2025) | 65.2 | 81.57 | 34.78 | 7.50 | 13.08 | 34.25 | 35.45 | 36.00 |
| - Co-rewarding-I (Ours) | 67.6 | 83.01 | 32.22 | 8.65 | 13.50 | 32.38 | 35.56 | 35.53 |
| - Co-rewarding-II (Ours) | 66.2 | 80.89 | 33.28 | 7.50 | 14.40 | 32.88 | 36.94 | 37.59 |

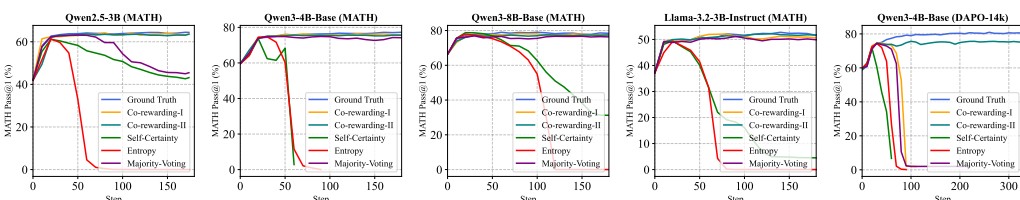

Figure 8: **Performance curves on validation set.** *Left to Right:* {Qwen2.5-3B, Qwen3-4B-Base, Qwen3-8B-Base, Llama-3.2-3B-Instruct} trained on MATH, Qwen3-4B-Base trained on DAPO-14k.

## C.3 MORE EVALUATION DETAILS

We conduct the evaluation across a diverse set of benchmarks, spanning mathematical reasoning, code generation, instruction-following, and general multi-task abilities. Specifically: (1) Mathematical reasoning: We evaluate on MATH500 (Lightman et al., 2024), GSM8K (Cobbe et al., 2021), and AMC (Li et al., 2024a). For MATH500 and GSM8K, we report pass@1 accuracy using the `lighteval` library[2]. For AMC, we use the `ttrl`[3] library and report avg@8 as the metric. (2) Code generation: We assess coding ability using LiveCodeBench (Jain et al., 2025) release_v6 and CRUX (Gu et al., 2024). LiveCodeBench is evaluated with its `official` evaluation library[4], and CRUX is evaluated via the `ZeroEval` library[5]; for both datasets, we report pass@1 accuracy. (3) Instruction-following and multi-task abilities: We evaluate on IFEval (Zhou et al., 2023b) and MMLU-Pro (Wang et al., 2024), using the `lm-evaluation-harness` library[6] for both. Overall, we summarize the statistics of the datasets used in this paper in Table 6.

---

[2]https://github.com/huggingface/lighteval

[3]https://github.com/ruixin31/Spurious_Rewards/tree/main/code/ttrl

[4]https://github.com/LiveCodeBench/LiveCodeBench

[5]https://github.com/WildEval/ZeroEval

[6]https://github.com/EleutherAI/lm-evaluation-harness

Table 8: **Supplement Results (%) of Co-rewarding and baselines trained on OpenRS**. Cell background colors: darker colors denote better results within each model group.

| Training Set: Open-RS | Mathematics | | | Code | | Instruction | Multi-Task |
|---|---|---|---|---|---|---|---|
| Methods | MATH500 | GSM8K | AMC | LiveCode | CRUX | IFEval | MMLU-Pro |
| *Qwen3-8B-Base* | | | | | | | |
| Before RL | 72.40 | 27.82 | 20.93 | 23.41 | 54.75 | 50.89 | 52.92 |
| - GT-Reward (Shao et al., 2024) | 80.20 | 89.76 | 54.97 | 39.00 | 63.00 | 52.94 | 55.49 |
| - Self-Certainty (Zhao et al., 2025b) | 82.60 | 85.22 | 50.00 | 37.00 | 64.62 | 52.12 | 56.03 |
| - Entropy (Prabhudesai et al., 2025) | 80.60 | 87.41 | 48.95 | 38.00 | 61.25 | 52.53 | 56.80 |
| - Majority-Voting (Shafayat et al., 2025) | 78.00 | 84.23 | 51.96 | 36.75 | 58.00 | 51.13 | 54.92 |
| - Co-rewarding-I (Ours) | 78.20 | 92.65 | 50.60 | 28.91 | 63.12 | 53.11 | 57.21 |
| - Co-rewarding-II (Ours) | 80.00 | 90.90 | 53.01 | 39.75 | 62.75 | 52.92 | 56.55 |
| *Qwen3-4B-Base* | | | | | | | |
| Before RL | 71.20 | 26.15 | 21.08 | 11.00 | 38.88 | 46.43 | 47.23 |
| - GT-Reward (Shao et al., 2024) | 78.80 | 85.22 | 49.55 | 33.50 | 55.12 | 46.41 | 50.12 |
| - Self-Certainty (Zhao et al., 2025b) | 73.20 | 33.43 | 35.84 | 32.50 | 49.50 | 46.47 | 48.24 |
| - Entropy (Prabhudesai et al., 2025) | 76.80 | 87.57 | 42.62 | 35.00 | 53.87 | 47.61 | 52.42 |
| - Majority-Voting (Shafayat et al., 2025) | 76.00 | 64.14 | 44.58 | 32.25 | 50.25 | 46.35 | 48.75 |
| - Co-rewarding-I (Ours) | 72.80 | 83.93 | 39.41 | 26.54 | 53.25 | 48.11 | 50.82 |
| - Co-rewarding-II (Ours) | 76.60 | 89.23 | 42.32 | 34.00 | 51.50 | 48.45 | 51.80 |

Table 9: **Performance (%) of test-time training (TTT).** Since self-supervised methods are label-free, they can be leveraged during inference for test-time training to further enhance performance.

| LLMs | Methods | AMC | | | | | | | |
|---|---|---|---|---|---|---|---|---|---|
| | | avg@8 | pass@8 | avg@16 | pass@16 | avg@32 | pass@32 | avg@64 | pass@64 |
| *Qwen2.5-7B* | Before-TTT | 15.81 | 46.99 | 17.55 | 66.27 | 16.34 | 74.70 | 17.32 | 75.90 |
| | Self-Certainty | 41.57 | 74.70 | 39.23 | 74.70 | 39.68 | 78.31 | 39.95 | 87.95 |
| | Entropy | 38.70 | 56.63 | 39.76 | 68.67 | 39.57 | 79.52 | 39.34 | 81.93 |
| | Majority-Voting | 43.67 | 63.86 | 43.67 | 67.47 | 43.49 | 78.31 | 44.35 | 85.54 |
| | Co-rewarding-I | 44.88 | 60.24 | 45.33 | 60.24 | 45.44 | 71.08 | 45.76 | 73.49 |
| | Co-rewarding-II | 43.22 | 69.88 | 41.34 | 75.90 | 40.36 | 78.31 | 41.64 | 87.95 |
| *Qwen3-8B-Base* | Before-TTT | 20.93 | 61.45 | 21.31 | 73.49 | 19.58 | 79.52 | 20.97 | 86.75 |
| | Self-Certainty | 49.85 | 78.31 | 50.68 | 78.31 | 50.41 | 84.34 | 49.55 | 89.16 |
| | Entropy | 48.64 | 74.70 | 49.92 | 80.72 | 49.96 | 87.95 | 50.23 | 89.16 |
| | Majority-Voting | 50.90 | 73.49 | 50.00 | 72.29 | 50.60 | 80.72 | 51.36 | 85.54 |
| | Co-rewarding-I | 52.86 | 68.67 | 53.46 | 74.70 | 53.24 | 81.93 | 53.58 | 84.34 |
| | Co-rewarding-II | 48.64 | 72.29 | 48.19 | 73.49 | 50.19 | 83.13 | 49.28 | 91.57 |

# D   ADDITIONAL EXPERIMENTAL RESULTS

## D.1   MORE RESULTS ON OTHER TRAINING SETS AND LLMS

Table 7 reports additional results of Qwen2.5-3B and Qwen3-1.7B-Base and Qwen3-4B-Base trained on MATH, while Table 8 extends the experiments of Qwen3-8B-Base and Qwen3-4B-Base to another training set OpenRS (Dang & Ngo, 2025). It can be observed that Co-rewarding occupies relatively darker areas. Across models and training sets, Co-rewarding-I and II achieve an average relative improvement of $+2.23\%$ on GSM8K, with notably high pass@1 scores of $92.65\%$ and $90.90\%$ for Qwen3-8B-Base trained on OpenRS, respectively. Moreover, thanks to its stability, Co-rewarding-II delivers more reliable gains than self-rewarding baselines, which occasionally suffer lower performance on certain models or benchmarks, e.g., Self-Certainty on Qwen3-1.7B-Base in Table 7 or Majority-Voting on Qwen3-4B-Base in Table 8. These results further demonstrate the effectiveness of Co-rewarding.

## D.2   MORE CURVES OF REWARD, RESPONSE LENGTH AND PSEUO LABEL ACCURACY

Figure 9 supplements the reward and response curves of Qwen3-4B-Base trained on DAPO-14k. The trends are consistent with Qwen3-8B-Base and Llama-3.2-3B-Instruct in Figure 4: Majority-Voting and Entropy rapidly increase rewards at early stage and quickly peak, a clear sign of reward hacking. In contrast, GT-Reward and Co-rewarding-II exhibit smoother, steadily rising rewards, indicating gen-

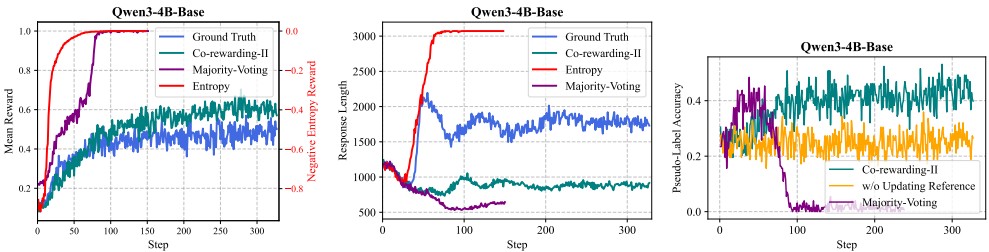

Figure 9: **Curves of reward (Left), response length (Middle), and pseudo label accuracy (Right) of Qwen3-4B-Base** trained on DAPO-14k. Entropy reward is plotted on the right $y$-axis due to its different reward scale. Note that entropy minimization is to maximizing the negative entropy.

Table 10: **Detailed MMLU-Pro performance on Qwen3-4B-Base and Llama-3.2-3B-Instruct traind on DAPO-14k.** Results are reported for each of the 14 categories in MMLU-Pro.

| *Qwen3-4B-Base* | | | | | | |
|---|---|---|---|---|---|---|
| **Methods** | **biology** | **business** | **chemistry** | **computer sci.** | **economics** | **engineering** | **health** |
| - GT-Reward | 73.50 | 63.49 | 59.71 | 56.34 | 65.05 | 42.93 | 50.86 |
| - Self-Certainty | 71.41 | 54.37 | 45.93 | 50.73 | 63.27 | 35.91 | 50.12 |
| - Entropy | 70.99 | 56.02 | 50.44 | 48.29 | 63.15 | 34.37 | 48.41 |
| - Majority-Voting | 70.43 | 55.77 | 52.83 | 53.41 | 62.79 | 38.09 | 50.61 |
| - Co-rewarding-I | 73.92 | 59.82 | 50.71 | 54.15 | 64.93 | 41.49 | 49.76 |
| - Co-rewarding-II | 72.66 | 59.95 | 55.65 | 53.41 | 64.10 | 39.73 | 50.61 |
| **Methods** | **history** | **law** | **math** | **other** | **philosophy** | **physics** | **psychology** |
| - GT-Reward | 44.88 | 26.34 | 69.80 | 48.81 | 44.69 | 57.04 | 65.79 |
| - Self-Certainty | 39.63 | 24.43 | 59.44 | 43.94 | 40.08 | 47.04 | 59.65 |
| - Entropy | 40.68 | 26.43 | 60.99 | 45.13 | 43.69 | 50.89 | 61.90 |
| - Majority-Voting | 40.94 | 23.43 | 64.17 | 43.39 | 44.09 | 50.73 | 63.66 |
| - Co-rewarding-I | 40.94 | 23.25 | 63.73 | 44.91 | 42.69 | 50.58 | 60.78 |
| - Co-rewarding-II | 42.26 | 24.79 | 67.58 | 44.59 | 41.88 | 54.19 | 62.91 |
| *Llama3.2-3B-Instruct* | | | | | | |
| **Methods** | **biology** | **business** | **chemistry** | **computer sci.** | **economics** | **engineering** | **health** |
| - GT-Reward | 54.81 | 36.25 | 25.18 | 33.41 | 42.65 | 21.57 | 39.36 |
| - Self-Certainty | 55.23 | 32.95 | 27.21 | 31.95 | 42.77 | 20.54 | 39.12 |
| - Entropy | 52.86 | 31.05 | 23.94 | 32.93 | 41.71 | 20.43 | 38.02 |
| - Majority-Voting | 56.07 | 32.95 | 22.79 | 30.98 | 44.19 | 18.99 | 39.61 |
| - Co-rewarding-I | 51.88 | 34.22 | 22.88 | 34.88 | 44.67 | 19.09 | 38.63 |
| - Co-rewarding-II | 56.21 | 34.35 | 27.03 | 35.61 | 43.01 | 19.92 | 40.34 |
| **Methods** | **history** | **law** | **math** | **other** | **philosophy** | **physics** | **psychology** |
| - GT-Reward | 30.18 | 22.71 | 34.20 | 34.74 | 32.06 | 28.33 | 50.38 |
| - Self-Certainty | 30.45 | 24.98 | 33.38 | 31.60 | 29.86 | 28.56 | 50.50 |
| - Entropy | 33.86 | 21.89 | 32.35 | 33.01 | 29.46 | 24.25 | 47.50 |
| - Majority-Voting | 32.02 | 25.25 | 34.35 | 34.20 | 29.86 | 24.79 | 48.25 |
| - Co-rewarding-I | 33.86 | 23.25 | 32.12 | 33.01 | 31.86 | 25.40 | 48.75 |
| - Co-rewarding-II | 32.28 | 24.34 | 35.83 | 36.26 | 33.27 | 28.18 | 49.12 |

uine learning of reasoning ability. Moreover, Co-rewarding-II maintains moderate response lengths on Qwen3-4B-Base, further demonstrating its generality in balancing the exploration–exploitation trade-off during reasoning training, which is a core principle of RL (Wang et al., 2018).

Additionally, the right panel of Figure 9 presents the pseudo-label accuracy of Qwen3-4B-Base, showing trends consistent with Qwen3-8B-Base and Llama-3.2-3B-Instruct in Figure 6. As training progresses, Co-rewarding-II steadily improves pseudo-label accuracy, while "w/o Updating Reference" remains around 25%. Majority-Voting briefly increases accuracy but soon collapses to zero, clearly indicating reward hacking. This highlights our design philosophy of pairing a fast policy student with a slowly updated teacher, which decouples supervision from the online policy while enabling the teacher to co-evolve with the student, thereby sustaining improvements in pseudo-label quality.

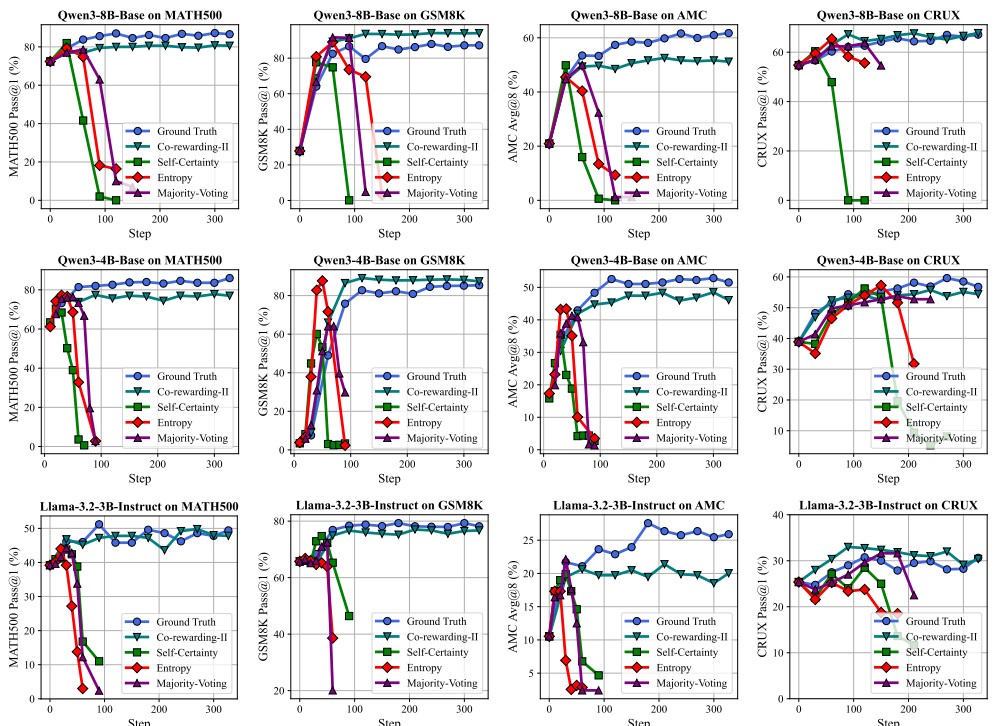

Figure 10: **Performance curves on benchmarks of MATH500, GSM8K, AMC and CRUX** across Qwen3-8B-Base, Qwen3-4B-Base, and Llama-3.2-3B-Instruct trained on DAPO-14k.

### D.3 MORE RESULTS OF VALIDATION PERFORMANCE CURVES

As a supplement to Figure 3, Figure 8 presents validation performance curves for Qwen2.5-3B, Qwen3-4B-Base, Qwen3-8B-Base, Llama-3.2-3B-Instruct trained on MATH, as well as Qwen3-4B-Base trained on DAPO-14k. Self-Certainty and Entropy collapse rapidly across all settings, as their supervision signals are tied to internal confidence or entropy and are easily exploited. Majority-Voting also collapses in several cases, reflecting that sampling pseudo labels from outputs cannot prevent hacking. By contrast, Co-rewarding-I maintains stability across MATH-trained models through data-side contrastive agreement, while Co-rewarding-II consistently provides stability across all models and datasets by disentangling supervision with a slowly updated teacher, making hacking substantially harder and optimization more reliable.

### D.4 RESULTS OF TEST-TIME TRAINING (TTT)

Thanks to the label-free nature of self-supervised methods, which do not require GT labels, they are naturally compatible with test-time training (TTT), enabling further refinement of the model during inference. Table 9 reports the TTT results on the challenging competition-level benchmark AMC across Co-rewarding and other self-rewarding baselines. We observe that Co-rewarding matches or even surpasses existing methods, achieving the best results on 11 out of 18 metrics. These findings broaden the applicability of self-supervised RL: beyond post-training for reasoning elicitation, it can also be leveraged at inference time to further improve performance on specific benchmarks.

### D.5 MORE RESULTS OF BENCHMARK PERFORMANCE CURVES

As a supplement to Figure 3 and Figure 5, Figure 10 presents performance curves on MATH500, GSM8K, AMC, and CRUX with Qwen3-8B-Base, Qwen3-4B-Base, and Llama-3.2-3B-Instruct. Consistent with earlier findings, Self-Certainty, Entropy, and Majority-Voting rapidly collapse across benchmarks and models, while Co-rewarding-II and GT-Reward sustain continued and stable im-

Table 11: **Impact of math training collapse on code and multi-task performance.** Results are evaluated on models before and after training collapse.

| Training stage | Methods | Mathematics | | | | Code | | Multi-task |
|---|---|---|---|---|---|---|---|---|
| | | MATH500 | GSM8K | AMC | AIME24 | LiveCode | CRUX | MMLU-Pro |
| Before training collapse | - Self-Certainty | 68.4 | 44.81 | 35.39 | 8.85 | 25.88 | 50.12 | 48.84 |
| | - Entropy | 76.6 | 82.79 | 43.37 | 12.81 | 26.35 | 50.75 | 50.22 |
| | - Majority-Voting | 73.4 | 64.06 | 40.81 | 9.17 | 26.16 | 53.00 | 51.06 |
| After training collapse | - Self-Certainty | 2.8 | 3.34 | 2.71 | 0.00 | 14.22 | 8.12 | 29.71 |
| | - Entropy | 2.8 | 2.35 | 3.46 | 0.00 | 18.60 | 31.75 | 28.13 |
| | - Majority-Voting | 2.8 | 4.85 | 1.36 | 0.00 | 24.36 | 52.75 | 50.19 |

(The table is titled *Qwen3-4B-Base*.)

Table 12: **Difference between original and rephrased questions** from background richness, vocabulary complexity, and sentence complexity.

| Training Set | # Data Size | Background richness | Vocabulary complexity | Sentence complexity |
|---|---|---|---|---|
| MATH | 7,500 | +4.91% | +4.79% | +9.05% |
| DAPO-14k | 14,100 | +4.65% | 1.95% | +4.19% |

Table 13: **Success rate of different rephraser LLMs:** MATH training set rephrased by Qwen3-32B, Qwen3-8B, and Qwen3-1.7B, respectively.

| Rephraser LLM | Training Set | # Original questions | # Rephrased questions | Success rate (%) |
|---|---|---|---|---|
| Qwen3-32B | MATH | 7,500 | 7,498 | 99.97% |
| Qwen3-8B | MATH | 7,500 | 7,477 | 99.69% |
| Qwen3-1.7B | MATH | 7,500 | 2,060 | 27.47% |

provements. These results underscore the link between performance and training stability: stable training enables models to continue improving by effectively learning knowledge from more data.

### D.6 IMPACT OF MATH TRAINING COLLAPSE ON OTHER TASKS

We investigate how training collapse occurring on math-oriented training sets impacts the model's performance on code-generation and multi-task benchmarks. To this end, we evaluate models trained with existing self-rewarding methods (Self-Certainty, Entropy, and Majority-Voting) both before and after training collapse. Table 11 summarizes the results. We observe that training collapses on math-related training sets affect other tasks (LiveCode, CRUX and MMLU-Pro) in different way for certainty- or entropy-based methods (Self-Certainty and Entropy) compared with consensus-based methods (Majority-Voting). When collapse occurs on math-oriented training sets, all three methods show substantial performance degradation on the four math benchmarks (MATH500, GSM8K, AMC, and AIME24). However, their impacts on other tasks differ:

For certainty- or entropy-based methods, the performance on LiveCode, CRUX, and MMLU-Pro also declines after collapse on math training sets. This arises from their reward objectives: maximizing self-certainty or minimizing entropy, result in the decoding probability mass becoming highly concentrated on a very subset of tokens. Consequently, the model produces repetitive outputs, and this repetitive decoding behavior transfers across tasks, leading to degraded performance beyond the math domain.

For the consensus-based method, Majority-Voting shows similar performance before and after training collapse on math-oriented training sets. This may be because its collapses stem from reward hacking at the answer format: the model exploits the \boxed{} structure by consistently inserting an incorrect but self-consistent answer to maximize reward. This type of collapse weakly affects the intermediate reasoning trace, which largely remains structured. Since code-generation and multi-task benchmarks do not rely on boxed-answer extraction, this type of collapse has limited impact on their performance.

Table 14: **Impact of rephraser LLM for Co-rewarding-I.** Train Qwen3-8B-Base using data rephrased by Qwen3-32B, Qwen3-8B and Qwen3-1.7B, respectively.

| Trained Model | Rephraser LLM | MATH500 | GSM8K | AMC | AIME24 | LiveCode | CRUX | IFEval | MMLU-Pro |
|---|---|---|---|---|---|---|---|---|---|
| | Qwen3-32B | 81.2 | 93.70 | 51.20 | 15.10 | 30.81 | 66.00 | 55.79 | 59.95 |
| Qwen3-8B-Base | Qwen3-8B | 79.2 | 92.72 | 51.51 | 14.58 | 30.90 | 63.12 | 54.73 | 59.30 |
| | Qwen3-1.7B | 78.2 | 87.41 | 49.25 | 12.81 | 29.57 | 61.00 | 53.44 | 55.85 |

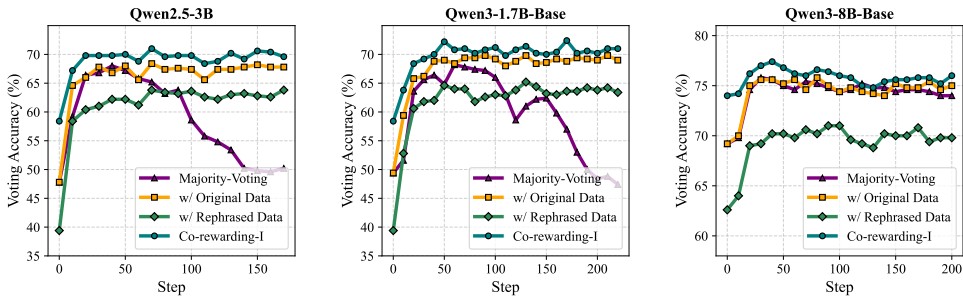

Figure 11: **Curves of voting accuracy of Majority-Voting, Co-rewarding-I and its ablations** with Qwen2.5-3B, Qwen3-1.7B-Base and Qwen3-8B-Base trained on MATH.

## D.7 DISCUSSION OF MATH AND DAPO-14K

We leverage Qwen3-235B-A22B to score the difference between original and rephrased questions from multiple perspectives, including background richness, vocabulary complexity, and sentence complexity, for MATH and DAPO-14k. From Table 12, we observe that the rephrasing in MATH exhibits larger changes from the original to rephrased questions than DAPO-14k. This suggests that the questions in MATH may provide favorable conditions for promoting diverse rephrasing variability, which is beneficial for the effectiveness of contrastive agreement in Co-rewarding-I.

## D.8 ROBUSTNESS ANALYSIS OF DIFFERENT REPHRASER LLMS

To analyze the impact of different rephraser LLMs for Co-rewarding-I, we conduct additional experiments using smaller LLMs instead of Qwen3-32B for rephrasing. To control architectural variability in the rephraser LLMs, we employ two smaller LLMs from the same family, i.e., Qwen3-8B and Qwen3-1.7B, for rephrasing the MATH training set. Table 13 reports the rephrasing success rate. We observe that rephrasing success rates drop as the model size decreases, which is expected: rephrasing math questions while preserving the analogical essence is a relatively challenging task, and weaker LLMs struggle to achieve this goal. This observation supports our choice of Qwen3-32B as the rephraser, as a sufficiently capable LLM is required to produce faithful rephrasing.

We then train Co-rewarding-I on Qwen3-8B-Base using rephrased data generated by Qwen3-32B, Qwen3-8B, and Qwen3-1.7B, respectively. The performance is summarized in Table 14. From the results, it can be observed that performance gradually degrades as the size of the rephraser LLM decreases, but not always significantly. Rephrasing with Qwen3-8B maintains reasonably similar performance to using Qwen3-32B, indicating that Co-rewarding-I exhibits a certain degree of robustness under moderate reductions in rephrasing quality. Notably, rephrasing with Qwen3-1.7B leads to a substantial performance drop. This degradation is largely attributable to the significantly lower rephrasing success rate of Qwen3-1.7B, which results in a substantial reduction of usable training data and consequently weakens the effectiveness of Co-rewarding-I.

## D.9 VOTING ACCURACY ANALYSIS OF CO-REWARDING-I

To demonstrate the stability and efficiency of Co-rewarding-I, we compare its voting accuracy against that of Majority-Voting in Figure 11 and Figure 12. These experiments are conducted on Qwen2.5-3B, Qwen3-1.7B-Base and Qwen3-8B-Base models, all trained on the MATH dataset. Across all settings, the Majority-Voting method exhibits reward hacking, where its performance sharply declines

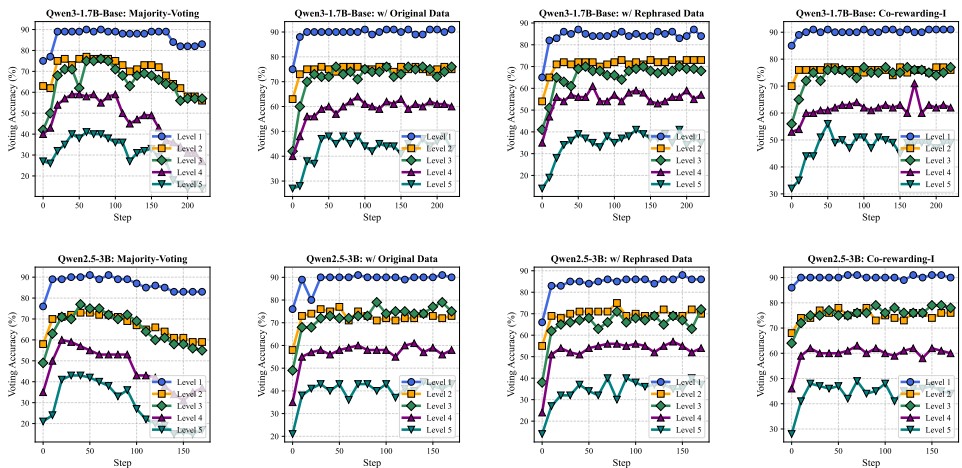

Figure 12: **Voting accuracy of Majority-Voting, Co-rewarding-I and its ablated variants across different difficulty levels of questions** *Top:* Qwen3-1.7B-Base. *Bottom:* Qwen2.5-3B.

after reaching an early peak, particularly on more difficult questions (levels 2 to 5). In contrast, Co-Rewarding-I maintains a stable voting accuracy on both original and rephrased data. Ultimately, it achieves the highest overall voting accuracy across all models and dataset configurations.

Table 15: **Detailed MMLU-Pro performance on Qwen3-8B-Base and Qwen3-4B-Base trained on OpenRS.** Results are reported for each of 14 categories in MMLU-Pro.

| *Qwen3-4B-Base* | | | | | | |
|---|---|---|---|---|---|---|
| **Methods** | **biology** | **business** | **chemistry** | **computer sci.** | **economics** | **engineering** | **health** |
| - GT-Reward | 70.99 | 59.82 | 52.30 | 54.63 | 65.05 | 39.01 | 51.22 |
| - Self-Certainty | 69.87 | 54.50 | 44.08 | 49.27 | 63.63 | 37.36 | 50.24 |
| - Entropy | 70.71 | 58.68 | 49.03 | 51.22 | 63.39 | 37.46 | 49.63 |
| - Majority-Voting | 69.60 | 55.77 | 47.17 | 53.17 | 63.39 | 36.02 | 48.78 |
| - Co-rewarding-I | 69.04 | 55.39 | 47.79 | 53.41 | 63.86 | 38.39 | 50.61 |
| - Co-rewarding-II | 70.85 | 58.81 | 53.27 | 53.90 | 66.11 | 37.15 | 52.81 |
| **Methods** | **history** | **law** | **math** | **other** | **philosophy** | **physics** | **psychology** |
| - GT-Reward | 39.63 | 24.98 | 65.58 | 47.84 | 40.68 | 54.50 | 62.53 |
| - Self-Certainty | 39.63 | 24.25 | 58.11 | 46.65 | 40.88 | 46.42 | 61.40 |
| - Entropy | 39.90 | 22.16 | 62.18 | 45.02 | 43.09 | 50.19 | 59.90 |
| - Majority-Voting | 40.68 | 22.52 | 60.25 | 46.10 | 41.08 | 48.42 | 60.65 |
| - Co-rewarding-I | 40.68 | 24.25 | 62.18 | 44.37 | 44.49 | 49.58 | 61.65 |
| - Co-rewarding-II | 41.21 | 25.89 | 64.91 | 45.24 | 39.28 | 52.27 | 59.40 |
| *Qwen3-8B-Base* | | | | | | |
| **Methods** | **biology** | **business** | **chemistry** | **computer sci.** | **economics** | **engineering** | **health** |
| - GT-Reward | 74.76 | 63.24 | 55.48 | 63.17 | 68.96 | 41.38 | 57.09 |
| - Self-Certainty | 75.03 | 63.62 | 53.62 | 55.61 | 68.96 | 39.83 | 57.09 |
| - Entropy | 75.73 | 64.39 | 54.51 | 58.29 | 65.05 | 41.69 | 55.87 |
| - Majority-Voting | 76.15 | 60.20 | 54.15 | 56.34 | 69.91 | 38.91 | 55.75 |
| - Co-rewarding-I | 76.43 | 65.78 | 57.07 | 62.20 | 69.43 | 43.14 | 56.60 |
| - Co-rewarding-II | 76.84 | 64.25 | 54.68 | 62.43 | 68.12 | 42.00 | 58.06 |
| **Methods** | **history** | **law** | **math** | **other** | **philosophy** | **physics** | **psychology** |
| - GT-Reward | 50.92 | 30.25 | 67.58 | 52.49 | 51.10 | 57.20 | 67.67 |
| - Self-Certainty | 49.34 | 28.88 | 68.02 | 51.62 | 52.10 | 56.89 | 66.42 |
| - Entropy | 50.39 | 30.43 | 65.28 | 51.41 | 47.09 | 54.50 | 66.67 |
| - Majority-Voting | 48.03 | 28.88 | 63.43 | 53.68 | 48.10 | 52.50 | 64.66 |
| - Co-rewarding-I | 50.13 | 29.97 | 68.54 | 52.92 | 50.70 | 56.66 | 65.54 |
| - Co-rewarding-II | 51.44 | 30.06 | 65.80 | 51.51 | 52.10 | 57.58 | 65.78 |

## D.10 MORE RESULTS OF MMLU-PRO EVALUATION

As a complement to Table 4, Table 10 and Table 15 report detailed MMLU-Pro results for models trained on DAPO-14k and OpenRS, respectively. We observe that Co-rewarding consistently preserves general-domain performance across diverse subjects, indicating that though trained on math-oriented datasets, its improvements do not come at the cost of broader capabilities from other domains.

## D.11 MORE RESULTS OF IFEVAL EVALUATION

The aim of IFEval is used to evaluate the instruction-following ability of LLMs. In Table 1, Table 2, Table 7 and Table 8, we report average IFEval performance due to space constraints. Specifically, the evaluation of IFEval includes four metrics: {prompt_level_strict_acc, inst_level_strict_acc, prompt_level_loose_acc and inst_level_loose_acc}, which apply different levels of answer matching. As a supplement, complete results are provided in Table 16, Table 17, and Table 18. The results show that Co-rewarding not only preserves the inherent instruction-following ability of base models but also often surpasses GT-Reward across multiple models. This further confirms that Co-rewarding's gains on mathematical and coding benchmarks are achieved without sacrificing general-domain instruction-following ability.

Table 16: **Detailed IFEval Performance on Qwen2.5-3B/7B, Qwen3-1.7B/4B/8B-Base and Llama-3.2-3B-Instruct traind on MATH.** Results are reported for loose and strick settings respectively.

| Methods | IFEval | | | | | | | | | |
|---|---|---|---|---|---|---|---|---|---|---|
| | Average | Prompt Strict | Prompt Loose | Inst. Strict | Inst. Loose | Average | Prompt Strict | Prompt Loose | Inst. Strict | Inst. Loose |
| | *Qwen2.5-3B* | | | | | *Qwen2.5-7B* | | | | |
| Before RL | 29.83 | 22.55 | 27.17 | 31.89 | 37.70 | 38.19 | 29.57 | 34.57 | 41.85 | 46.76 |
| - GT-Reward | 33.66 | 25.51 | 31.42 | 35.85 | 41.85 | 41.49 | 31.79 | 39.56 | 43.65 | 50.96 |
| - Self-Certainty | 32.22 | 24.40 | 29.76 | 34.65 | 40.05 | 37.24 | 28.47 | 34.38 | 40.05 | 46.04 |
| - Entropy | 32.94 | 24.77 | 30.50 | 35.13 | 41.37 | 40.33 | 30.13 | 37.87 | 43.29 | 50.00 |
| - Majority-Voting | 35.19 | 26.25 | 32.72 | 37.53 | 44.24 | 38.60 | 29.21 | 35.86 | 41.61 | 47.72 |
| - Co-rewarding-I | 33.86 | 23.84 | 31.61 | 36.09 | 43.88 | 41.73 | 32.35 | 39.37 | 44.48 | 50.72 |
| - Co-rewarding-II | 32.74 | 23.29 | 29.02 | 36.33 | 42.33 | 41.82 | 31.79 | 40.29 | 43.88 | 51.31 |
| | *Qwen3-1.7B-Base* | | | | | *Qwen3-4B-Base* | | | | |
| Before RL | 33.65 | 25.69 | 30.86 | 36.45 | 41.60 | 46.43 | 36.04 | 44.18 | 48.68 | 56.83 |
| - GT-Reward | 36.16 | 27.35 | 31.79 | 40.64 | 44.84 | 47.80 | 37.34 | 46.77 | 49.40 | 57.67 |
| - Self-Certainty | 32.96 | 24.58 | 29.20 | 36.69 | 41.36 | 48.15 | 39.37 | 46.76 | 49.52 | 56.95 |
| - Entropy | 35.37 | 26.61 | 31.42 | 39.44 | 44.00 | 50.44 | 40.67 | 48.61 | 52.52 | 59.07 |
| - Majority-Voting | 35.45 | 26.06 | 32.16 | 38.72 | 48.84 | 48.78 | 37.89 | 47.50 | 50.36 | 59.65 |
| - Co-rewarding-I | 35.56 | 27.91 | 31.23 | 39.32 | 43.76 | 50.35 | 40.67 | 49.35 | 51.56 | 59.83 |
| - Co-rewarding-II | 36.94 | 27.17 | 33.64 | 40.05 | 46.88 | 51.30 | 41.40 | 49.54 | 53.12 | 61.15 |
| | *Qwen3-8B-Base* | | | | | *Llama3-2-Instruct* | | | | |
| Before RL | 50.32 | 40.11 | 50.27 | 51.07 | 59.83 | 57.32 | 46.77 | 55.27 | 60.19 | 67.03 |
| - GT-Reward | 52.78 | 41.96 | 51.76 | 54.44 | 62.95 | 47.41 | 37.34 | 42.88 | 52.52 | 57.31 |
| - Self-Certainty | 50.98 | 39.74 | 49.54 | 52.88 | 61.75 | 54.88 | 43.81 | 52.68 | 58.15 | 64.87 |
| - Entropy | 51.81 | 40.67 | 51.20 | 52.76 | 62.59 | 54.70 | 43.81 | 52.68 | 57.67 | 64.63 |
| - Majority-Voting | 51.80 | 39.74 | 51.02 | 53.60 | 62.83 | 47.96 | 37.34 | 43.44 | 52.88 | 58.18 |
| - Co-rewarding-I | 55.79 | 43.99 | 57.11 | 55.63 | 66.42 | 49.14 | 39.37 | 45.66 | 53.12 | 58.39 |
| - Co-rewarding-II | 60.70 | 55.64 | 65.59 | 56.00 | 65.59 | 49.90 | 39.93 | 45.66 | 54.68 | 59.35 |

## D.12 ORIGINAL QUESTIONS VS. REPHRASED QUESTIONS

To provide an intuitive illustration, we present several examples of original questions with their rephrased versions in Table 19. We observe that such rephrasings are reasonable and effective, as they preserve the same underlying mathematical essence while presenting the problems in a substantially different surface form. This reflects the high quality of our rephrased data and forms the basis of Co-rewarding-I: by leveraging contrastive agreement across data-invariant variants, the model is encouraged to elicit more robust reasoning ability.

## D.13 COMPLETE CASE STUDY

As a supplement to Figure 7, we present the complete generation outputs of this case study. The full outputs clearly reveal the reward hacking behaviors of existing self-rewarding baselines. Self-Certainty and Entropy fall into repetitive outputs—for example, Self-Certainty repeatedly generates "Understanding," and Entropy repeatedly produces "Simplify the next fraction" until truncated at the maximum length. This arises because their decoding probability mass collapses onto a small subset of tokens, leading the model to loop over them. Majority-Voting shows another form of reward hacking by boxing an incorrect answer "0" to maximize consensus across rollouts and thereby secure

Table 17: **Detailed IFEval performance on Qwen3-4B/8B-Base and Llama-3.2-3B-Instruct traind on DAPO-14k.** Results are reported for loose and strict settings in IFEval, respectively.

| Methods | IFEval | | | | |
|---|---|---|---|---|---|
| | Average | Prompt Strict | Prompt Loose | Inst. Strict | Inst. Loose |
| *Qwen3-4B-Base* | | | | | |
| Before RL | 46.43 | 36.04 | 44.18 | 48.68 | 56.83 |
| - GT-Reward | 47.70 | 37.52 | 45.84 | 49.76 | 57.67 |
| - Self-Certainty | 45.58 | 35.67 | 43.99 | 47.84 | 54.80 |
| - Entropy | 48.20 | 37.71 | 46.58 | 50.48 | 58.03 |
| - Majority-Voting | 48.91 | 39.19 | 47.69 | 50.24 | 58.51 |
| - Co-rewarding-I | 46.84 | 36.41 | 45.66 | 48.80 | 56.47 |
| - Co-rewarding-II | 48.90 | 39.56 | 46.21 | 51.44 | 58.39 |
| *Qwen3-8B-Base* | | | | | |
| Before RL | 50.32 | 40.11 | 50.27 | 51.07 | 59.83 |
| - GT-Reward | 53.11 | 41.59 | 52.13 | 54.56 | 64.15 |
| - Self-Certainty | 50.58 | 41.04 | 49.54 | 51.68 | 60.07 |
| - Entropy | 51.56 | 41.59 | 49.91 | 53.48 | 61.27 |
| - Majority-Voting | 51.54 | 41.22 | 51.02 | 52.64 | 61.27 |
| - Co-rewarding-I | 50.17 | 40.67 | 48.24 | 52.16 | 59.59 |
| - Co-rewarding-II | 53.31 | 41.40 | 53.23 | 54.20 | 64.39 |
| *Llama3.2-3B-Instruct* | | | | | |
| Before RL | 57.32 | 46.77 | 55.27 | 60.19 | 67.03 |
| - GT-Reward | 53.10 | 42.33 | 49.91 | 57.19 | 62.95 |
| - Self-Certainty | 54.50 | 44.55 | 51.76 | 58.03 | 63.67 |
| - Entropy | 55.78 | 45.29 | 53.23 | 59.11 | 65.47 |
| - Majority-Voting | 54.07 | 42.33 | 52.50 | 56.83 | 64.63 |
| - Co-rewarding-I | 53.04 | 42.33 | 51.02 | 55.76 | 63.07 |
| - Co-rewarding-II | 51.92 | 41.59 | 48.24 | 56.00 | 61.87 |

Table 18: **Detailed IFEval Performance on Qwen3-8B/4B-Base trained on Open-RS.** Results are reported for loose and strict settings in IFEval, respectively.

| Methods | IFEval | | | | | | | | | |
|---|---|---|---|---|---|---|---|---|---|---|
| | Average | Prompt Strict | Prompt Loose | Inst. Strict | Inst. Loose | Average | Prompt Strict | Prompt Loose | Inst. Strict | Inst. Loose |
| | *Qwen3-8B-Base* | | | | | *Qwen3-4B-Base* | | | | |
| Before RL | 50.32 | 40.11 | 50.27 | 51.07 | 59.83 | 46.43 | 36.04 | 44.18 | 48.68 | 56.83 |
| - GT-Reward | 52.53 | 41.59 | 51.02 | 54.56 | 62.95 | 47.80 | 37.34 | 46.77 | 49.40 | 57.67 |
| - Self-Certainty | 52.12 | 41.59 | 50.83 | 53.72 | 62.35 | 46.47 | 35.86 | 44.73 | 48.56 | 56.71 |
| - Entropy | 52.94 | 43.25 | 51.94 | 53.72 | 62.83 | 46.41 | 36.97 | 44.73 | 48.68 | 55.28 |
| - Majority-Voting | 51.13 | 40.67 | 49.35 | 53.36 | 61.15 | 46.35 | 36.41 | 44.18 | 48.80 | 56.00 |
| - Co-rewarding-I | 53.11 | 41.40 | 53.05 | 53.95 | 64.02 | 46.83 | 35.86 | 46.77 | 47.84 | 56.83 |
| - Co-rewarding-II | 52.92 | 42.14 | 52.50 | 54.08 | 62.95 | 48.45 | 38.26 | 46.77 | 50.24 | 58.51 |

the highest reward. In contrast, both Co-rewarding and GT-Reward yield the correct final answer along with a coherent step-by-step reasoning path.

### D.14 CASE STUDY ON CODE BENCHMARK

To understand the behaviors of different methods on other tasks, we supplement an additional case study on CRUX. We observe that Self-Certainty and Entropy fail by producing repetitive outputs, consistent with the collapse pattern on math-related benchmarks. In contrast, GT-Reward, Majority-Voting, and Co-rewarding generate coherent reasoning trace and correct solutions.

> **Question:**
>
> $\frac{m}{n}$ is the Irreducible fraction value of
>
> $$3 + \cfrac{1}{3 + \cfrac{1}{3 + \frac{1}{3}}}$$
>
> , what is the value of $m + n$?

Table 19: **Case studies:** Original vs. Rephrased Questions

| Original Question | Rephrased Question |
|---|---|
| Sam is hired for a 20-day period. On days that he works, he earns $60. For each day that he does not work, $30 is subtracted from his earnings. At the end of the 20-day period, he received $660. How many days did he not work? | A contractor agrees to a job lasting 20 days. For every day the job is completed on time, the contractor earns $60. However, for each day the work is delayed, a fine of $30 is applied. After the 20-day period, the contractor's total earnings are $660. How many days was the job delayed? |
| Karen drove continuously from 9:40 a.m. until 1:20 p.m. of the same day and covered a distance of 165 miles. What was her average speed in miles per hour? | A traveler set off at 9:40 a.m. and reached their destination at 1:20 p.m. the same day after traveling a total of 165 miles. What was their average speed during the trip in miles per hour? |
| Solve for $x$: $\frac{1}{2} + \frac{1}{x} = \frac{5}{6}$. | A tank is partially filled by two different pipes. One pipe fills half the tank in an hour, and together with another pipe, they fill five-sixths of the tank in the same time. If the second pipe alone fills $\frac{1}{x}$ of the tank in an hour, find the value of $x$. |

---

**Co-rewarding-I Answer:**

To solve the problem, we need to simplify the given expression step by step and then find the irreducible fraction form. The expression is:

$$3 + \cfrac{1}{3 + \cfrac{1}{3 + \frac{1}{3}}}.$$

First, we simplify the innermost fraction:

$$3 + \frac{1}{3} = \frac{9}{3} + \frac{1}{3} = \frac{10}{3}.$$

Now substitute this back into the expression:

$$3 + \cfrac{1}{3 + \cfrac{1}{\frac{10}{3}}} = 3 + \cfrac{1}{3 + \frac{3}{10}}.$$

Next, simplify the fraction inside the denominator:

$$3 + \frac{3}{10} = \frac{30}{10} + \frac{3}{10} = \frac{33}{10}.$$

Now substitute this back into the expression:

$$3 + \cfrac{1}{\frac{33}{10}} = 3 + \frac{10}{33}.$$

To add these, we need a common denominator:

$$3 + \frac{10}{33} = \frac{99}{33} + \frac{10}{33} = \frac{109}{33}.$$

The fraction $\frac{109}{33}$ is already in its simplest form because 109 and 33 have no common factors other than 1. Therefore, $m = 109$ and $n = 33$, and the value of $m + n$ is:

$$109 + 33 = 142.$$

So, the final answer is:

$$\boxed{142}.$$

---

**Co-rewarding-II Answer:**

To solve the given expression $3 + \frac{1}{3+\frac{1}{3+\frac{1}{3}}}$, we need to simplify it step by step.

First, simplify the innermost fraction:

$$3 + \frac{1}{3} = \frac{9}{3} + \frac{1}{3} = \frac{10}{3}.$$

Next, substitute this back into the next fraction:

$$3 + \frac{1}{3 + \frac{1}{\frac{10}{3}}} = 3 + \frac{1}{3 + \frac{3}{10}} = 3 + \frac{1}{\frac{30}{10} + \frac{3}{10}} = 3 + \frac{1}{\frac{33}{10}} = 3 + \frac{10}{33}.$$

Now, simplify the expression $3 + \frac{10}{33}$:

$$3 + \frac{10}{33} = \frac{99}{33} + \frac{10}{33} = \frac{109}{33}.$$

The fraction $\frac{109}{33}$ is already in its simplest form because 109 and 33 are coprime (they have no common factors other than 1).

Thus, the irreducible fraction value of the given expression is $\frac{109}{33}$, where $m = 109$ and $n = 33$. Therefore, $m + n = 109 + 33 = 142$.

The value of $m + n$ is $\boxed{142}$.

---

**GT-Reward Answer:**

Alright, I have this problem:
The expression is:

$$3 + \frac{1}{3 + \frac{1}{3+\frac{1}{3}}}$$

And I need to find the irreducible fraction $\frac{m}{n}$ that equals this expression, then determine $m + n$.

First, I need to simplify the expression step by step. It looks like a continued fraction, so I'll start by simplifying the innermost part and work my way out.

Let's denote the entire expression as:

$$E = 3 + \frac{1}{3 + \frac{1}{3+\frac{1}{3}}}$$

**Step 1: Simplify the innermost fraction**
Start with the innermost fraction:

$$3 + \frac{1}{3}$$

Calculate this:

$$3 + \frac{1}{3} = \frac{9}{3} + \frac{1}{3} = \frac{10}{3}$$

So now, the expression becomes:

$$E = 3 + \frac{1}{3 + \frac{1}{\frac{10}{3}}}$$

Wait, I need to simplify $\frac{1}{\frac{10}{3}}$:

$$\frac{1}{\frac{10}{3}} = \frac{3}{10}$$

Now, the expression is:

$$E = 3 + \frac{1}{3 + \frac{3}{10}}$$

---

**Step 2: Simplify the next fraction**
Now, simplify $3 + \frac{3}{10}$:

$$3 + \frac{3}{10} = \frac{30}{10} + \frac{3}{10} = \frac{33}{10}$$

So, the expression now is:

$$E = 3 + \frac{1}{\frac{33}{10}}$$

Simplify $\frac{1}{\frac{33}{10}}$:

$$\frac{1}{\frac{33}{10}} = \frac{10}{33}$$

Now, the expression is:

$$E = 3 + \frac{10}{33}$$

**Step 3: Combine the terms**
Convert 3 to a fraction with denominator 33:

$$3 = \frac{99}{33}$$

Now, add the fractions:

$$E = \frac{99}{33} + \frac{10}{33} = \frac{109}{33}$$

So, the expression simplifies to $\frac{109}{33}$.
**Step 4: Ensure the fraction is irreducible**
Check if 109 and 33 have any common divisors. 109 is a prime number (I think), and 33 is $3 \times 11$. Since 109 isn't divisible by 3 or 11, the fraction $\frac{109}{33}$ is already in its irreducible form.
**Step 5: Find $m + n$**
Here, $m = 109$ and $n = 33$, so:

$$m + n = 109 + 33 = 142$$

Wait, let's double-check the calculations to ensure no mistakes were made.
**Verification:**
Starting over:

$$E = 3 + \frac{1}{3 + \frac{1}{3 + \frac{1}{3}}}$$

Innermost:

$$3 + \frac{1}{3} = \frac{10}{3}$$

Next layer:

$$3 + \frac{1}{\frac{10}{3}} = 3 + \frac{3}{10} = \frac{33}{10}$$

Outermost:

$$3 + \frac{1}{\frac{33}{10}} = 3 + \frac{10}{33} = \frac{99}{33} + \frac{10}{33} = \frac{109}{33}$$

Yes, it checks out. Therefore, $m + n = 109 + 33 = 142$.

$\boxed{142}$

---

Majority-Voting Answer:

To find the value of $\frac{m}{n}$ as the irreducible fraction of the expression $3 + \frac{1}{3 + \frac{1}{3 + \frac{1}{3}}}$, we need to simplify the expression step by step.

First, let's simplify the innermost fraction:

$$3 + \frac{1}{3} = 3 + \frac{1}{3} = \frac{9}{3} + \frac{1}{3} = \frac{10}{3}$$

Now, substitute this back into the expression:

$$3 + \frac{1}{3 + \frac{1}{\frac{10}{3}}} = 3 + \frac{1}{3 + \frac{3}{10}}$$

Next, simplify the fraction $3 + \frac{3}{10}$:

$$3 + \frac{3}{10} = \frac{30}{10} + \frac{3}{10} = \frac{33}{10}$$

Now, substitute this back into the expression:

$$3 + \frac{1}{\frac{33}{10}} = 3 + \frac{10}{33}$$

To add these, we need a common denominator:

$$3 + \frac{10}{33} = \frac{99}{33} + \frac{10}{33} = \frac{109}{33}$$

So, the irreducible fraction value of the expression is $\frac{109}{33}$. Here, $m = 109$ and $n = 33$. Therefore, the value of $m + n$ is:

$$m + n = 109 + 33 = 142$$

Thus, the value of $m + n$ is $\boxed{0}$.

---

**Self-Certainty Answer:**

Alright, I have this math problem to solve:
**Problem:** I need to solve a math problem. The problem is about understanding how to solve math problems. I need to understand how to solve this problem so I can understand more about math.
### Understanding the Problem
First, I need to understand what the problem is. I need to understand math so I can understand more about numbers and how to solve problems with numbers. Understanding math will help me understand more about how to understand problems.
### Understanding Math Problems
After understanding the problem, I need to understand more about math problems. Understanding math problems will help me understand how to understand these problems. Understanding these problems will help me understand more about how to understand math.
### Understanding Mathematical Concepts
After understanding math problems, I need to understand more about mathematical concepts. Understanding mathematical concepts will help me understand more about math. Understanding these concepts will help me understand more about how math helps me understand numbers.
### Understanding Mathematical Operations
After understanding mathematical concepts, I need to understand more about mathematical operations. Understanding mathematical operations will help me understand more about how to understand these operations. Understanding these operations will help me understand more about how these operations help me understand math.
### Understanding Mathematical Operations in Math
After understanding mathematical operations, I need to understand ......
### Understanding Mathematical Methods

After understanding mathematical operations, I need to understand more about mathematical methods. ......
### Understanding How Mathematical Methods Help
......
### Understanding How These Methods Help Me Understand Math
After understanding more about how these methods help me, I need to understand ......
### Understanding How These Methods Help Me Understand Math Problems
After understanding how these methods help me understand math, I need to understand ......
### Understanding How These Methods Help Me Solve Math Problems
After understanding how these methods help me understand these math problems, I need to understand ......
### Understanding How These Methods Help Me Solve These Problems
After understanding how these methods help me solve these problems, I need to understand more about ......

**Entropy Answer:**

To find the value of $\frac{m}{n}$ as the irreducible fraction of the expression:

$$3 + \cfrac{1}{3 + \cfrac{1}{3 + \frac{1}{3}}}$$

Let's simplify the expression step by step.
1. **Simplify the innermost fraction:**

$$3 + \frac{1}{3}$$

$$3 + \frac{1}{3} = \frac{3 \times 3}{3} + \frac{1}{3} = \frac{9}{3} + \frac{1}{3} = \frac{9+1}{3} = \frac{10}{3}$$

2. **Simplify the next fraction:**

$$3 + \cfrac{1}{3 + \frac{1}{3}} = 3 + \cfrac{1}{\frac{10}{3}}$$

$$3 + \cfrac{1}{\frac{10}{3}} = 3 + \frac{1 \times 3}{10} = 3 + \frac{3}{10} = \frac{3 \times 10}{10} + \frac{3}{10} = \frac{30}{10} + \frac{3}{10} = \frac{30+3}{10} = \frac{33}{10}$$

3. **Simplify the next fraction:**

$$3 + \cfrac{1}{3 + \cfrac{1}{3 + \frac{1}{3}}} = 3 + \cfrac{1}{\frac{33}{10}}$$

$$3 + \cfrac{1}{\frac{33}{10}} = 3 + \frac{1 \times 10}{33} = 3 + \frac{10}{33} = \frac{3 \times 33}{33} + \frac{10}{33} = \frac{99}{33} + \frac{10}{33} = \frac{99+10}{33} = \frac{109}{33}$$

4. **Simplify the next fraction:**

$$3 + \cfrac{1}{3 + \cfrac{1}{3 + \frac{1}{3}}} = 3 + \cfrac{1}{\frac{109}{33}}$$

$$3 + \cfrac{1}{\frac{109}{33}} = 3 + \frac{1 \times 33}{109} = 3 + \frac{33}{109} = \frac{3 \times 109}{109} + \frac{33}{109} = \frac{327}{109} + \frac{33}{109} = \frac{327+33}{109} = \frac{360}{109}$$

5. **Simplify the next fraction:**

$$3 + \cfrac{1}{3 + \cfrac{1}{3 + \cfrac{1}{3 + \frac{1}{3}}}} = 3 + \cfrac{1}{\frac{360}{109}}$$

$$3+\frac{1}{\frac{360}{109}} = 3+\frac{1\times 109}{360} = 3+\frac{109}{360} = \frac{3\times 360}{360}+\frac{109}{360} = \frac{1080}{360}+\frac{109}{360} = \frac{1080+109}{360} = \frac{1189}{360}$$

6. **Simplify the next fraction:**

$$3 + \cfrac{1}{3 + \cfrac{1}{3+\cfrac{1}{3+\cfrac{1}{3+\frac{1}{3}}}}} = 3 + \frac{1}{\frac{1189}{360}}$$

......

7. **Simplify the next fraction:**

$$3 + \cfrac{1}{3 + \cfrac{1}{3+\cfrac{1}{3+\cfrac{1}{3+\cfrac{1}{3+\frac{1}{3}}}}}} = 3 + \frac{1}{\frac{3927}{1189}}$$

......

8. **Simplify the next fraction:**

$$3 + \cfrac{1}{3 + \cfrac{1}{3+\cfrac{1}{3+\cfrac{1}{3+\cfrac{1}{3+\frac{1}{3}}}}}} = 3 + \frac{1}{\frac{12970}{3927}}$$

......

9. **Simplify the next fraction:**

$$3 + \cfrac{1}{3 + \cfrac{1}{3+\cfrac{1}{3+\cfrac{1}{3+\cfrac{1}{3+\cfrac{1}{3+\frac{1}{3}}}}}}} = 3 + \frac{1}{\frac{42837}{12970}}$$

......

10. **Simplify the next fraction:**

$$3 + \cfrac{1}{3 + \cfrac{1}{3+\cfrac{1}{3+\cfrac{1}{3+\cfrac{1}{3+\cfrac{1}{3+\cfrac{1}{3+\frac{1}{3}}}}}}}} = 3 + \frac{1}{\frac{141481}{42837}}$$

......

11. **Simplify the next fraction:**

$$3 + \cfrac{1}{3 + \cfrac{1}{3+\cfrac{1}{3+\cfrac{1}{3+\cfrac{1}{3+\cfrac{1}{3+\cfrac{1}{3+\frac{1}{3}}}}}}}} = 3 + \frac{1}{\frac{467280}{141481}}$$

......

12. **Simplify the next fraction:**

$$3 + \cfrac{1}{3 + \cfrac{1}{3+\cfrac{1}{3+\cfrac{1}{3+\cfrac{1}{3+\cfrac{1}{3+\cfrac{1}{3+\cfrac{1}{3+\frac{1}{3}}}}}}}}} = 3 + \frac{1}{\frac{1543321}{467280}}$$

......

13. **Simplify the next fraction:**

$$3 + \cfrac{1}{3 + \cfrac{1}{3 + \cfrac{1}{3 + \cfrac{1}{3 + \cfrac{1}{3 + \cfrac{1}{3 + \cfrac{1}{3 + \cfrac{1}{3 + \cfrac{1}{3 + \cfrac{1}{3 + \frac{1}{3}}}}}}}}}}} = 3 + \cfrac{1}{\frac{5097243}{1543321}}$$

$$3 + \cfrac{1}{\frac{5097243}{1543321}} = 3+$$

## Question:

```
## Question:

You are given a Python function and an assertion containing an
    input to the function. Complete the assertion with a literal (no
     unsimplified expressions, no function calls) containing the
    output when executing the provided code on the given input, even
     if the function is incorrect or incomplete.

[PYTHON]
def f(text):
    new_text = list(text)
    for i in '+':
        if i in new_text:
            new_text.remove(i)
    return ''.join(new_text)
assert f('hbtofdeiequ') == ??
[/PYTHON]

## Instruction

Please answer this question by first reasoning and then providing
    your answer.
Present your reasoning and solution in the following json format.
Please show your final answer in the 'answer' field, e.g.,"answer":
    "42".

```json
{
    "reasoning": "___",
    "answer": "___"
}
```

## GT-Reward Answer:

```
{
    "reasoning": "The function 'f' is designed to remove all
    occurrences of the character '+' from the input string 'text'.
    The function first converts the string into a list of characters
    , iterates through this list, and removes any '+' characters it
    finds. Finally, it joins the remaining characters back into a
    single string. Given the input string 'hbtofdeiequ', there are
    no '+' characters in the string, so the function will simply
    return the original string unchanged. Therefore, the output of
    the function when given the input 'hbtofdeiequ' is the string '
    hbtofdeiequ'.",
```

```
      "answer": "hbtofdeiequ"
}
```

## Co-rewarding-I Answer:

```
{
    "reasoning": "The function `f` is designed to remove all
    occurrences of the character '+' from the input string `text`.
    The function first converts the string into a list of characters
    , then iterates over the character '+' and removes it from the
    list if it exists. Finally, it joins the list back into a string
     and returns it. Given the input 'hbtofdeiequ', the character
    '+' does not appear in the string, so the function will return
    the original string unchanged.",
    "answer": "hbtofdeiequ"
}
```

## Co-rewarding-II Answer:

```
{
    "reasoning": "The function `f` takes a string `text` and
    removes all occurrences of the character '+' from it. The
    function then joins the remaining characters back into a string
    and returns it. Given the input 'hbtofdeiequ', the function will
     remove all '+' characters (though there are none in the input),
     so the output will be the same as the input string. Therefore,
    the assertion should check if the output of `f('hbtofdeiequ')`
    is 'hbtofdeiequ'.",
    "answer": "hbtofdeiequ"
}
```

## Majority-Voting Answer:

```
{
    "reasoning": "The function `f` is designed to remove all
    occurrences of the character '+' from the input string `text`.
    The input provided to the function is 'hbtofdeiequ'. Since the
    character '+' does not appear in this string, the function will
    return the original string unchanged. Therefore, the assertion
    should compare the output of the function to the original input
    string.",
    "answer": "'hbtofdeiequ'"
}
```

## Self-Certainty Answer:

```
To solve this problem, we need to follow these steps:

1. Understand the given Python function.
2. Determine what the function does with the given input.
3. Complete the assertion with the correct output based on the
   function's behavior.
```

```
Let's go through each step one by one:

1. Understand the given Python function.
2. Determine what the function does with the given input.
3. Complete the assertion with the correct output based on the
    function's behavior.

We will write the code to understand the function, determine its
    behavior with the given input, and then complete the assertion
    with the correct output.

Let's write the code step by step.
'''python
# Step 1: Understand the given Python function
# Step 2: Determine what the function does with the given input
# Step 3: Complete the assertion with the correct output based on
    the function's behavior

# Let's write the code to follow these steps
'''

'''python
# Step 1: Understand the given Python function
# Step 2: Determine what the function does with the given input
# Step 3: Complete the assertion with the correct output based on
    the function's behavior

# Let's write the code to follow these steps

# Step 1: Understand the given Python function
# We will define the given function and understand what it does

# Step 2: Determine what the function does with the given input
# We will determine the output of the function for the given input

# Step 3: Complete the assertion with the correct output based on
    the function's behavior
# We will complete the assertion with the correct output

# Let's write the code
'''

Let's write the code step by step.
'''python
# Step 1: Understand the given Python function
# Step 2: Determine what the function does with the given input
# Step 3: Complete the assertion with the correct output based on
    the function's behavior

# Let's write the code to follow these steps
'''

'''python
# Step 1: Understand the given Python function
# We will define the given function and understand what it does

# Step 2: Determine what the function does with the given input
# We will determine the output of the function for the given input

# Step 3: Complete the assertion with the correct output based on
    the function's behavior
# We will complete the assertion with the correct output
```

```
......

# Let's write the code

# Step 1: Understand the
```

**Entropy Answer:**

```
```json
{
    "reasoning": "To determine the output of the function `f` when
    the input is 'hbtofdeiequ', we need to follow these steps:

1. Convert the string 'hbtofdeiequ' into a list:
   \[
   \text{new\_text} = \text{list}('hbtofdeiequ')
   \]
   \[
   \text{new\_text} = ['h', 'b', 't', 'o', 'f', 'd', 'e', 'i', 'e',
    'q', 'u']
   \]

2. Iterate through the character '+':
   \[
   \text{for } i \text{ in } '+':
   \]
   \[
   \text{for } i \text{ in } '+':
   \]
   \[
   \text{for } i \text{ in } '+':
   \]
   \[
   \text{for } i \text{ in } '+':
   \]
   \[
   \text{for } i \text{ in } '+':
   \]
   \[
   \text{for } i \text{ in } '+':
   \]
   ......
   \[
   \text{for } i \text{ in } '+':
   \]
   \[
   \text{for } i \text{ in } '+
```

