# OpenReview forum: "Co-rewarding: Stable Self-supervised RL for Eliciting Reasoning in Large Language Models"
_ICLR.cc/2026/Conference — ICLR 2026 Poster_

### Official Review · Reviewer_pcnh · 2025-10-30

**Soundness:** 3
**Presentation:** 3
**Contribution:** 3
**Rating:** 6
**Confidence:** 4

**Summary:**

The paper introduces Co-rewarding, a self-supervised RL framework for eliciting LLM reasoning that aims to avoid the training collapse often seen in label-free/self-reward approaches. The core idea is to decouple the reward signal from the current policy's single-view outputs by enforcing invariance views across data-side and model-side.

Co-rewarding-I (data-side) obtains contrastive agreement between semantically analogous (rephrased) questions and the original, using majority-vote pseudo-labels across rollouts to shape a GRPO-style objective $J_{\text{Co-rewarding-I}}$ with cross-refereed advantages (Eqs. (6)–(8)). Co-rewarding-II (model-side) reuses the GRPO reference model as a slowly updated teacher via EMA (cosine-scheduled $\alpha(k)$) to generate rollouts and majority-vote pseudo-labels, thus temporally decoupling supervision from the online policy (Eqs. (9)–(11)).

Across Qwen and Llama backbones, Co-rewarding improves stability and accuracy over recent self-reward baselines (Self-Certainty, Entropy, Majority Voting).

**Strengths:**

1. The instability of single-view self-reward (reward hacking -> collapse) is articulated and empirically supported, with Co-rewarding showing steadier progress and avoided collapse.
2. The idea is interesting, differs from (i) internal score rewards (entropy/certainty) and (ii) single-view self-consistency.
3. The method is simple to implement and seems to be quite practical, it reuses GRPO reference as teacher, and no extra LLM is introduced (beyond rephrasing for co-rewarding-I).
4. Good experimental results, achieves 94.01% GSM8K Pass@1 on DAPO-14k with Qwen3-8B-Base.

**Weaknesses:**

1. The paper shows co-rewarding-I collapses on DAPO-14k but II does not; the causes are hypothesized (rephrasing efficacy) rather than directly measured.
2. The use of Qwen3-32B to generate rephrases may raise a concern of equivalence/label leakage beyond examples; although instructions are provided and case studies look reasonable.
3. On code/IFEval/MMLU-Pro, improvements are mixed/modest, and there is little task-level analysis (e.g., error modes, solution length vs. correctness).
4. The mechanisms behind beating GT-reward is not dissected (e.g., improved exploration vs. label noise in GT-reward?).

**Questions:**

1. For I, can you quantify features (lexical diversity, named-entity variation, length) that predict stability across MATH vs DAPO-14k, validating the "rich background descriptions" hypothesis?
2. For AMC/CRUX/IFEval/MMLU-Pro, where gains are small, what failure modes are most common? Would task-aware rephrasing (e.g., code refactorings or spec restatements) strengthen I/II?

---

> ### Author Response · Authors · 2025-11-22
> **Response to reviewer pcnh (Part 1/3)**
>
> We sincerely thank the reviewer for the time and effort spent reviewing our paper and for acknowledging the the novelty of our method, extensive experiments, and strong performance. Below, we address the raised concerns and questions in detail:
>
> > **W1:** The paper shows co-rewarding-I collapses on DAPO-14k but II does not; the causes are hypothesized (rephrasing efficacy) rather than directly measured.
> **Q1:** For I, can you quantify features (lexical diversity, named-entity variation, length) that predict stability across MATH vs DAPO-14k, validating the "rich background descriptions" hypothesis?
>
> **A:** Thanks for your insightful comments. We agree with the reviewer that providing quantitative evidence can help us understand the different behaviors of Co-rewarding-I on MATH and DAPO-14k. Following the reviewer's suggestion, **we leverage Qwen3-235B-A22B to score the difference between original and rephrased questions from multiple perspectives, including background richness, vocabulary complexity, and sentence complexity, for MATH and DAPO-14k**. The results are reported as follows:
>
> | Training Set | Number of Questions | Background Richness | Vocabulary Complexity | Sentence Complexity |
> |--------------|----------------------|-----------------------|------------------------|----------------------|
> | MATH         | 7,500               | +4.91%               | +4.79%                | +9.05%              |
> | DAPO-14k     | 14,100              | +4.65%               | +1.95%                | +4.19%              |
>
> From the results, we observe that the rephrasing in MATH exhibits larger changes from the original to rephrased questions compared to DAPO-14k. This suggests that the questions in MATH may provide favorable conditions for promoting diverse rephrasing variability, which is beneficial for the effectiveness of contrastive agreement in Co-rewarding-I.
>
> > **W2:** The use of Qwen3-32B to generate rephrases may raise a concern of equivalence/label leakage beyond examples; although instructions are provided and case studies look reasonable.
>
> **A:** Thanks for your constructive comments. We would like to clarify that **our rephrasing process does not expose any ground-truth answers to the LLM rephraser**. As shown in the rephrasing prompt in Appendix C.2, the LLM rephraser is only provided with the question text, without any information of the label, solution or reasoning trace. The prompt template instructs a surface-level reformulation while preserving the underlying mathematical meaning, rather than inferring or predicting the answer.
>
> To promote rephrased data quality, we adopt Qwen3-32B, a recent and highly capable open-source LLM, as a rephraser. Based on our manual inspection of numerous rephrased-original pairs, **we do not observe signs of label leakage: the rephraser LLM faithfully follows our instructions and does not introduce any information that could hint at the ground-truth answers into rephrased questions**. Empirically, the comparable performance between "Majority-Voting w/ Original" and "Majority-Voting w/ Rephrased" in Table 3 further suggests that the rephrased questions have similar quality to their original counterparts, providing additional evidence that our rephrasing process is reasonable.

---

> ### Author Response · Authors · 2025-11-22
> **Response to reviewer pcnh (Part 2/3)**
>
> > **W3:** On code/IFEval/MMLU-Pro, improvements are mixed/modest, and there is little task-level analysis (e.g., error modes, solution length vs. correctness).
>
> **A:** Thanks for your valuable comments. Our current analysis mainly focuses on math reasoning scenarios, and we agree with the reviewer that integrating discussions for other tasks will improve the clarity of the manuscript. Following the reviewer's suggestion, **we further investigate how training collapse occurring on math-oriented training sets impacts the model's performance on code-generation and multi-task benchmarks**. To this end, we evaluate models trained with existing self-rewarding methods (Self-Certainty, Entropy, and Majority-Voting) both before and after training collapse. The results are summarized as follows:
> |Training Set: DAPO-14k|Method| MATH500 | GSM8K  | AMC    | AIME24 | LiveCode | CRUX   | MMLU-Pro |
> |-|-|---------|--------|--------|--------|----------|--------|----------|
> |**Before collapse**| Self-Certainty  | 68.4    | 44.81  | 35.39  | 8.85   | 25.88    | 50.12  | 48.84    |
> || Entropy         | 76.6    | 82.79  | 43.37  | 12.81  | 26.35    | 50.75  | 50.22    |
> |                         | Majority-Voting | 73.4    | 64.06  | 40.81  | 9.17   | 26.16    | 53.00  | 51.06    |
> | **After collapse**      | Self-Certainty  | 2.8     | 3.34   | 2.71   | 0.00   | 14.22    | 8.12   | 29.71    |
> |                         | Entropy         | 2.8     | 2.35   | 3.46   | 0.00   | 18.60    | 31.75  | 28.13    |
> |                         | Majority-Voting | 2.8     | 4.85   | 1.36   | 0.00   | 24.36    | 52.75  | 50.19|
>
> From the above results, we observe that **training collapses on math-related training sets affect other tasks (LiveCode, CRUX and MMLU-Pro) in different way for certainty- or entropy-based methods (Self-Certainty and Entropy) compared with consensus-based methods (Majority-Voting)**. When collapse occurs on math-oriented training sets, all three methods show substantial performance degradation on the four math benchmarks (MATH500, GSM8K, AMC, and AIME24). However, their impacts on other tasks differ:
>
> **For certainty- or entropy-based methods, the performance on LiveCode, CRUX, and MMLU-Pro also declines after collapse on math-related training sets.** This arises from their reward objectives: maximizing self-certainty or minimizing entropy, result in the decoding probability mass becoming highly concentrated on a very subset of tokens. Consequently, the model produces repetitive outputs, and this repetitive decoding behavior transfers across tasks, leading to degraded performance beyond the math domain.
>
> **For the consensus-based method, Majority-Voting shows similar performance before and after training collapse on math-oriented training sets.** This may be because its collapses stem from reward hacking at the answer format:  the model learns to exploit the "\boxed{}" structure by consistently inserting an incorrect but self-consistent answer to maximize reward. This type of collapse weakly affects the intermediate reasoning trace, which largely remains structured. Since code-generation and multi-task benchmarks do not rely on boxed-answer extraction, this type of collapse has limited impact on their performance.
>
> To further illustrate the behavioral differences across methods, following the reviewer's suggestion, **we supplement additional case studies on CRUX**. From these examples, we observe that Self-Certainty and Entropy fail by producing repetitive outputs, consistent with the collapse pattern identified earlier. In contrast, GT-Reward, Majority-Voting, and Co-rewarding generate coherent reasoning trace and correct solutions. These results and case studies will be included in the revised manuscript.

---

> ### Author Response · Authors · 2025-11-22
> **Response to reviewer pcnh (Part 3/3)**
>
> > **W4:** The mechanisms behind beating GT-reward is not dissected (e.g., improved exploration vs. label noise in GT-reward?).
>
> **A:** Thanks for your constructive comments. We agree with the reviewer that introducing a discussion about GT-Reward can improve the clarity of our manuscript. We would like to **further discuss the observation that our Co-rewarding seems to outperform GT-Reward GRPO in a few cases**. From Table 1 and Table 2, we can observe that GT-Reward with a reliable supervision signal still achieves stronger performance on most benchmarks. Surprisingly, in a few cases, such as on GSM8K, Co-rewarding reaches or exceeds GT-Reward. **We view these cases as encouraging evidence to reflect our performance superiority over other self-rewarding baselines.** This may be because GSM8K is a relatively easier benchmark, where self-supervised RL is sufficient to elicit the latent reasoning ability of the pretrained model without relying on GT labels. The stability of Co-rewarding effectively mitigates training collapse, allowing the model to train for longer and utilize more data, which other self-rewarding baselines often fail to do.
>
> We also observe that on code generation benchmarks, self-supervised methods sometimes surpass GT-Reward. **This may be attributed to the distribution difference between the training dataset and the evaluation benchmarks.** This distribution mismatch occasionally allows a self-supervised method to generalize as well as, or even better than, label-based supervised method in certain cases.
>
> In addition, **we note that annotated datasets such as MATH are not always perfectly clean**. GT-Reward may be affected by label noise in these datasets. For example, after manually inspecting the MATH training set, we find several cases where the official annotations are ill-formed or non-standard:
>
> | Question                                 | Annotated label | Standard annotation |
> |-------------------------------------------|------------------|----------------------|
> | Evaluate $\log_3\frac{1}{\sqrt3}$.        | -\frac12         | -\frac{1}{2}         |
> | Evaluate $\log_8 2$.                      | \frac13          | \frac{1}{3}          |
>
> These noisy labels may cause GT-Reward to penalize correct responses or reinforce incorrect ones, potentially affecting performance. We will include the above discussion in the revised manuscript.
>
> > **Q2:** For AMC/CRUX/IFEval/MMLU-Pro, where gains are small, what failure modes are most common? Would task-aware rephrasing (e.g., code refactorings or spec restatements) strengthen I/II?
>
> **A:** Thanks for your insightful question. We would like to clarify that rephrasing is only performed in Co-rewarding-I, not in Co-rewarding-II. Our current research primarily focuses on math-oriented reasoning, and all training sets (MATH, DAPO-14k, OpenRS) are math-related datasets. **Accordingly, the evaluation benchmarks include several different diagnostic purposes rather than all aiming to measure direct gains**:
>
> For **math-related benchmarks**, such as MATH500, GSM8K, and AMC, **the goal is to directly evaluate the improvement in mathematical reasoning elicited by different methods**. In particular, although the reviewer mentions small gains, we would like to clarify that, on AMC, Co-rewarding-I and Co-rewarding-II achieve average improvement of +6.07% and +7.50% over the best self-rewarding baselines, respectively. This suggests that our method yields meaningful benefits in math problem-solving.
>
> For **code-generation benchmarks**, such as LiveCode and CRUX, we include them to **examine whether math-induced reasoning improvements generalize to out-of-distribution tasks** rather than expecting large gains. The results indicate that math-oriented training can be helpful in improving code reasoning ability.
>
> For **general-domain benchmarks** like IFEval and MMLU-Pro, our primary goal is to **verify that math-focused training does not harm the model's instruction-following and broad multi-task ability**. Co-rewarding maintains performance comparable to the base model on both benchmarks, indicating that our training process does not harm general-domain capability.
>
> Regarding task-award rephrasing, e.g., code refactoring or specification restatement, we think that this is a promising avenue for extending our Co-rewarding self-supervised RL framework to broader reasoning tasks. We also note that different domains may require distinct rephrasing strategies, which may face different challenges. The reviewer's suggestion represents a promising direction for future research. We will include the above discussion in the revised manuscript.

---

> ### Author Response · Authors · 2025-11-23
> **Kind reminder: updated manuscript**
>
> Thanks again for the reviewer's valuable time and efforts in reviewing our manuscript.
>
> We would like to let the reviewer know that **the manuscript has been updated**, and all modified content is marked with **purple** highlights for clarity. For the modified tables and figures, their captions have also been highlighted in **purple**.
>
> This is just a kind reminder. If you have any concerns or further suggestions, we would be glad to engage in further discussion.
>
> Authors of 11193

---

> ### Author Response · Authors · 2025-11-26
> **Looking forward to your reply**
>
> Dear Reviewer pcnh,
>
> We sincerely thank you for your efforts in reviewing our work and for your great support! Please let us know if you need any further information or if there are additional points you would like to discuss with us. We would be glad to engage further discussion and answer any questions of your interest.
>
> Thank you very much!
>
> Best regards,
>
> Authors of #11193

---

### Official Review · Reviewer_odXG · 2025-10-30

**Soundness:** 3
**Presentation:** 2
**Contribution:** 3
**Rating:** 6
**Confidence:** 4

**Summary:**

The paper defines two methods for unsupervised RL finetuning of LLMs for reasoning tasks. The first uses a similarity-based objective across semantic augmentations to train problems; the second uses majority voting across rollouts from a reference policy that periodically tracks the RL policy with an EMA update. The results show that these unsupervised reward signals stabilise training relative to other ground-truth-free baselines and can in some cases outperform using ground truth reward.

**Strengths:**

The paper sets out to define a stable and performant approach to RL finetuning for reasoning without use of ground truth labels and the results demonstrate the effectiveness of the proposed method. Three models are used across in-domain and out-of-domain benchmarks and two separate training datasets are considered. Combined with suitable ablations and analysis of training dynamics (length, reward) this makes for a comprehensive results section.

**Weaknesses:**

The training datasets used are ones for which ground truth labels are available. It seems important to validate the method in a setting that is better motivated by self-supervised methods (i.e., those without the availability of verifiable rewards during training).

The written communication of the paper could be improved. There are grammatical errors throughout.

The related work currently features in the appendix and should be in the main paper text.

The paper primarily reports pass@1 results, which leaves the reliability of the results uncertain.

**Questions:**

Why not experiment with a combination of Co-rewarding-I and Co-rewarding-II?

Why do you think Co-rewarding sometimes outperforms using GT reward? This seems unintuitive and is not currently adequately discussed, only stated.

---

> ### Author Response · Authors · 2025-11-22
> **Response to reviewer odXG (Part 1/4)**
>
> We sincerely thank the reviewer for the time and effort spent reviewing our paper and for acknowledging the clarity of our problem formulation, the novelty of our method, and the extensive experiments. Below, we address the raised concerns and questions in detail:
>
> > **W1:** The training datasets used are ones for which ground truth labels are available. It seems important to validate the method in a setting that is better motivated by self-supervised methods (i.e., those without the availability of verifiable rewards during training).
>
> **A:** Thanks for your valuable comments. We would like to clarify that, although the datasets used in our experiments contain ground-truth (GT) answers, **neither our method nor any self-rewarding baselines access or use these labels during training; only the questions themselves are used for training**. We use these datasets because they are the standard reasoning training sets in prior work [1,2,3] and allow a comparison between our self-supervised approach and the vanilla GT-supervised GRPO. **This setup is also widely adopted in self-supervised learning fields**, such as SimCLR [4] and DINO [5], where models are trained without GT labels even though the datasets include them, enabling comparisons with supervised methods. Therefore, **although the datasets used in our work contain GT answers, our experimental setup does not access or leak any GT information during training** and fully satisfies the self-supervised setting. Furthermore, we fully agree with the reviewer that extending self-supervised RL approaches to more other tasks is a promising direction in future research.
>
> [1] Yu Q, Zhang Z, Zhu R, et al. Dapo: An open-source llm reinforcement learning system at scale[J]. arXiv preprint arXiv:2503.14476, 2025.
> [2] Dang Q A, Ngo C. Reinforcement Learning for Reasoning in Small LLMs: What Works and What Doesn't[J]. arXiv preprint arXiv:2503.16219, 2025.
> [3] Zhao X, Kang Z, Feng A, et al. Learning to reason without external rewards[J]. arXiv preprint arXiv:2505.19590, 2025.
> [4] Chen T, Kornblith S, Norouzi M, et al. A simple framework for contrastive learning of visual representations[C]//International conference on machine learning. PmLR, 2020: 1597-1607.
> [5] Caron M, Touvron H, Misra I, et al. Emerging properties in self-supervised vision transformers[C]//Proceedings of the IEEE/CVF international conference on computer vision. 2021: 9650-9660.
>
> > **W2:** The written communication of the paper could be improved. There are grammatical errors throughout.
>
> **A:** Thanks for your kind reminder. We appreciate the reviewer's feedback on writing quality, and we will carefully proofread and check the full paper to correct the grammatical issues and further improve the overall writing quality of the manuscript.
>
> > **W3:** The related work currently features in the appendix and should be in the main paper text.
>
> **A:** Thanks for your valuable comments. We agree that discussing the related work in the main text can improve clarity and contextualization. Currently, the main body focuses on reporting several important experimental analyses that help clarify our method, which leads to space constraints. Following the reviewer's suggestion, **we have added a Related Work section in the main paper to improve readability**.

---

> ### Author Response · Authors · 2025-11-22
> **Response to reviewer odXG (Part 2/4)**
>
> > **W4:** The paper primarily reports pass@1 results, which leaves the reliability of the results uncertain.
>
> **A:** Thanks for your insightful comments. We would like to clarify that our choice of evaluation metrics is consistent with the commonly used settings in prior studies [1, 6, 7]. For example, Pass@1 is used for MATH500 and GSM8K to assess the problem-solving capability of models, while Avg@8 is used for AMC to evaluate the stability and reliability of models in generating solutions. **As a supplement, we evaluate the Avg@4 for MATH500 and Avg@32 for AIME24 across diverse models and baselines.** The results are reported as follows:
> |Training Set: MATH|Method|MATH500 Pass@1|MATH500 Avg@4|GSM8K Pass@1|AMC Avg@8|AIME24 Avg@32|
> |-|-|-|-|-|-|-|
> |**Qwen3-8B-Base**|Before RL|72.4|70.8|27.82|20.93|3.75|
> ||Self-Certainty|80.2|80.4|80.74|50.75|15.73|
> ||Entropy|80.2|79.5|87.19|49.54|15.63|
> ||Majority-Voting|79.8|80.4|89.76|49.09|**15.83**|
> ||Co-rewarding-I|**81.2**| 80.3|**93.70**|51.20|15.10|
> ||Co-rewarding-II|80.8|**80.7**|92.42|**53.46**|14.48|
> |**Qwen3-4B-Base**|Before RL|71.2|70.4|26.15|21.08|4.58|
> ||Self-Certainty|71.6|72.0|71.79|38.86|11.67|
> ||Entropy|77.0|77.3|88.10|**47.44**|10.94|
> ||Majority-Voting|77.4|76.3|90.07|45.33|10.10|
> ||Co-rewarding-I|**78.8**|**77.8**|**91.28**|46.08|**13.85**|
> ||Co-rewarding-II|78.0|77.5|88.86|45.93|12.17|
> |**Llama-3.2-3B-Instruct**| Before RL|39.2|39.6|65.73|10.54|3.75|
> ||Self-Certainty|43.4|44.3|74.91|18.83|6.88|
> ||Entropy|43.4|44.4|66.19|20.18|6.56|
> ||Majority-Voting|46.8|47.0|78.77|20.48|9.27|
> ||Co-rewarding-I|**50.2**|**49.6**|**79.45**|**23.80**|10.00|
> ||Co-rewarding-II|49.8|47.8|79.30|22.59|**10.73**|
>
> |Training Set: DAPO-14k|Method|MATH500 Pass@1|MATH500 Avg@4|GSM8K Pass@1|AMC Avg@8|AIME24 Avg@32|
> |-|-|-|-|-|-|-|
> | **Qwen3-8B-Base**     | Before RL                  | 72.4           | 70.8          | 27.82        | 20.93      | 3.75           |
> |                       | Self-Certainty             |**82.0**|**81.1**| 77.63        | 49.85      | 15.00          |
> |                       | Entropy                    | 79.4           | 80.7          | 80.82        | 45.48      | 15.00          |
> |                       | Majority-Voting            | 78.6           | 77.8          | 91.66        | 50.00      | 11.25          |
> |                       | Co-rewarding-I             | 78.4           | 79.0          | 88.02        | 51.20      | 11.88          |
> |                       | Co-rewarding-II            | 80.6           | 80.7|**94.01**|**54.37**|**16.35**|
> | **Qwen3-4B-Base**     | Before RL                  | 71.2 | 70.4          | 26.15        | 21.08      | 4.58           |
> |                       | Self-Certainty             | 68.4           | 68.3          | 44.81        | 35.39      | 8.85           |
> |                       | Entropy                    | 76.6           | 76.0          | 82.79        | 43.37      | 12.81          |
> |                       | Majority-Voting            | 73.4           | 73.8          | 64.06        | 40.81      | 9.17           |
> |                       | Co-rewarding-I             | 73.8           | 74.7          | 75.89        | 43.83      | 10.63          |
> |                       | Co-rewarding-II            |**77.8**|**77.5**|**91.89**|**48.49**|**14.27**|
> | **Llama-3.2-3B-Instruct**      | Before RL                  | 39.2           | 39.6          | 65.73        | 10.54      | 3.75           |
> |                       | Self-Certainty             | 42.4           | 42.7          | 74.71        | 17.32      | 4.79           |
> |                       | Entropy                    | 44.0           | 42.2          | 65.85        | 17.32      | 6.56           |
> |                       | Majority-Voting            | 42.8           | 43.2          | 70.96        | 17.62      |**8.74**           |
> |                       | Co-rewarding-I             | 46.0           | 45.5          | 70.58        |**20.93**| 7.08           |
> |                       | Co-rewarding-II            |**49.8**|**47.8**|**78.62**|19.73|8.02|
>
> We observe that **the behaviors of Avg@k (k>1) are similar to the trends of Pass@1: Co-rewarding outperforms existing self-rewarding baselines (Self-Certainty, Entropy and Majority-Voting) on most cases**. Specifically, Co-rewarding-I and Co-rewarding-II achieve average improvements of 4.82% and 6.31% over the best self-rewarding baselines on MATH and DAPO-14k, respectively. This demonstrates that the models trained by Co-rewarding can generate more correct reasoning answers under multi-round sampling compared to self-rewarding baselines, further reflecting the reliability of our Co-rewarding framework across multiple rollouts. We will include these results in the revised manuscript.

---

> ### Author Response · Authors · 2025-11-22
> **Response to reviewer odXG (Part 3/4)**
>
> > **Q1:** Why not experiment with a combination of Co-rewarding-I and Co-rewarding-II?
>
> **A:** Thanks for your contrustive question. We agree with the reviewer that, given the orthogonality of Co-rewarding-I and Co-rewarding-II, it is natural to explore a combined variant. Following this insight, we introduce **Co-rewarding-III, which integrates the data-side cross-supervision from rephrased questions (Co-rewarding-I) with the model-side self-distillation pseudo labels from the EMA-updated reference teacher (Co-rewarding-II)**. Specifically, Co-rewarding-III generates pseudo labels for both the original and rephrased questions from the slowly-updated teacher and then self-supervises across the two views. We conduct experiments of Co-rewarding-III across three models and two training datasets. The results are shown below:
> |Training Set: MATH| Method           | MATH500 | GSM8K | AMC   | AIME24 | LiveCode | CRUX  | IFEval | MMLU-Pro |
> |-----------------------|------------------|---------|-------|-------|--------|----------|-------|--------|----------|
> | **Qwen3-8B-Base**     | Co-rewarding-I   | 81.2    | 93.70 | 51.20 | 15.10  | 30.81    | 66.00 | 55.79  | 59.95    |
> |                       | Co-rewarding-II  | 80.8    | 92.42 | 53.46 | 14.48  | 30.23    | 62.83 | 60.70  | 57.50    |
> |                       | Co-rewarding-III | 81.4    | 90.98 | 54.07 | 13.33  | 30.71    | 63.75 | 53.69  | 59.10    |
> | **Qwen3-4B-Base**     | Co-rewarding-I   | 78.8    | 91.28 | 46.08 | 13.85  | 26.64    | 56.50 | 50.35  | 53.26    |
> |                       | Co-rewarding-II  | 78.0    | 88.86 | 45.93 | 12.17  | 26.25    | 55.00 | 51.30  | 53.88    |
> |                       | Co-rewarding-III | 78.6    | 90.75 | 48.80 | 12.71  | 26.16    | 56.00 | 49.23  | 53.08    |
> | **Llama-3.2-3B-Instruct**      | Co-rewarding-I   | 50.2    | 79.45 | 23.80 | 10.00  | 11.28    | 29.88 | 48.89  | 33.77    |
> |          | Co-rewarding-II  | 49.8    | 79.30 | 22.59 | 10.73  | 10.80    | 30.63 | 49.90  | 33.61    |
> |                       | Co-rewarding-III | 51.6    | 79.91 | 25.45 | 10.42  | 10.43    | 32.50 | 46.37  | 34.50    |
>
> | Training Set: DAPO-14k| Method           | MATH500 | GSM8K | AMC   | AIME24 | LiveCode | CRUX  | IFEval | MMLU-Pro |
> |-----------------------|------------------|---------|-------|-------|--------|----------|-------|--------|----------|
> | **Qwen3-8B-Base**     | Co-rewarding-I   | 78.4    | 88.02 | 51.20 | 11.88  | 29.38    | 62.50 | 50.17  | 55.39    |
> |                       | Co-rewarding-II  | 80.6    | 94.01 | 54.37 | 16.35  | 31.66    | 67.12 | 53.31  | 59.83    |
> |                       | Co-rewarding-III | 81.6    | 92.27 | 53.77 | 17.71  | 32.70    | 66.75 | 55.85  | 60.02    |
> | **Qwen3-4B-Base**     | Co-rewarding-I   | 73.8    | 75.89 | 45.83 | 10.63  | 26.25    | 50.12 | 46.84  | 51.51    |
> |                       | Co-rewarding-II  | 77.8    | 91.89 | 48.49 | 14.27  | 26.64    | 54.87 | 48.90  | 52.83    |
> |                       | Co-rewarding-III | 79.2    | 90.45 | 48.95 | 15.10  | 27.58    | 54.87 | 50.30  | 54.79    |
> | **Llama-3.2-3B-Instruct**      | Co-rewarding-I   | 46.0    | 70.58 | 20.93 | 7.08   | 9.57     | 27.25 | 53.04  | 32.61    |
> || Co-rewarding-II  | 49.8    | 78.62 | 19.73 | 8.02   | 10.43    | 32.25 | 51.92  | 34.46    |
> |                       | Co-rewarding-III | 48.6    | 76.95 | 21.84 | 8.13   | 9.86     | 30.50 | 49.92  | 34.01    |
>
> From the above results, it can be observed that combining the data-side and model-side views further boost the performance. Across two training sets and four math reasoning benchmarks (MATH500, GSM8K, AMC, and AIME24), despite some occanional slight drop in performance, **Co-rewarding-III achieves overall performance improvements on average of 7.11% over Co-rewarding-I and 1.72% over Co-rewarding-II**. On the code generation benchmarks (LiveCode and CRUX), Co-rewarding-III achieves average gains of 5.51% and 0.21% over Co-rewarding-I and Co-rewarding-II, respectively. These results suggest that the combined variant Co-rewarding-III benefits from both sides: the data-side view in Co-rewarding-I enhances data diversity, while the model-side view in Co-rewarding-II stabilizes pseudo-label generation by decoupling it from the online policy. Bringing these two views together makes reward hacking more difficult and yields more stable self-supervised RL training. We will revise the manuscript accordingly to include the description of Co-rewarding-III and its experimental results.

---

> ### Author Response · Authors · 2025-11-22
> **Response to reviewer odXG (Part 4/4)**
>
> > Q2: Why do you think Co-rewarding sometimes outperforms using GT reward? This seems unintuitive and is not currently adequately discussed, only stated.
>
> **A:** Thanks for your valuable question. We would like to **further discuss the observation that our Co-rewarding without need of GT labels seems to outperform GT-Reward GRPO in a few cases**. As shown in Table 1 and Table 2, GT-Reward generally remains stronger across most benchmarks. Our observation is simply that, in a few cases, such as 94.01% Pass@1 on GSM8K, Co-rewarding reaches or occasionally exceeds the performance of GT-Reward. This observation is indeed somewhat surprising, and we view it as an encouraging by-product to further reflect performance superiority of Co-rewarding over existing self-rewarding baselines. Given the evaluation benchmarks do not perfectly match the distribution of the training sets, even though both are math-related, this mismatch may create opportunities for self-supervised methods to perform better in certain cases.
>
> Additionally, **we find that annotated datasets such as MATH are not perfectly clean**. Consequently, GT-Reward can be influenced by label noise present in these datasets. During our manual inspection of the MATH training set, we identified several examples where the annotated labels are incorrectly formatted:
>
> | Question                                 | Annotated label | Standard annotation |
> |-------------------------------------------|------------------|----------------------|
> | Evaluate $\log_3\frac{1}{\sqrt3}$.        | -\frac12         | -\frac{1}{2}         |
> | Evaluate $\log_8 2$.                      | \frac13          | \frac{1}{3}          |
>
> Such noisy or non-standard labels may misguide training at answer-matching stage. In these cases, a correct solution predicted by the model may fail to match the annotated label and receive a penalty, while an incorrect prediction may be matched as correct. This mismatch can distort the reward signal in GT-Reward and negatively influence its training performance. We will include the above discussion in the revised manuscript.

---

> ### Author Response · Authors · 2025-11-23
> **Kind reminder: updated manuscript**
>
> Thanks again for the reviewer's valuable time and efforts in reviewing our manuscript.
>
> We would like to let the reviewer know that **the manuscript has been updated**, and all modified content is marked with **purple** highlights for clarity. For the modified tables and figures, their captions have also been highlighted in **purple**.
>
> This is just a kind reminder. If you have any concerns or further suggestions, we would be glad to engage in further discussion.
>
> Authors of 11193

---

> ### Author Response · Authors · 2025-11-26
> **Looking forward to your reply**
>
> Dear Reviewer odXG,
>
> We sincerely thank you for your efforts in reviewing our work and for your great support! Please let us know if you need any further information or if there are additional points you would like to discuss with us. We would be glad to engage further discussion and answer any questions of your interest.
>
> Thank you very much!
>
> Best regards,
>
> Authors of #11193

---

### Official Review · Reviewer_CbNe · 2025-10-30

**Soundness:** 3
**Presentation:** 3
**Contribution:** 3
**Rating:** 6
**Confidence:** 4

**Summary:**

This paper tackles the training collapse issue in self-rewarding RL for LLM reasoning. The authors propose Co-rewarding, a self-supervised RL framework that seeks complementary supervision for stability. It is a two-stage process where Co-rewarding-I uses "contrastive agreement" via cross-reference between original and rephrased questions and Co-rewarding-II employs self-distillation from a slowly-updated EMA "teacher model" to provide stable pseudo-labels, decoupling the reward signal from the online policy.

Experiments show Co-rewarding achieves stable training, avoiding the collapse seen in baselines like Entropy and Majority-Voting. The method outperforms these baselines and, notably, even surpasses training with ground-truth labels in several cases, achieving a 94.01% Pass@1 on GSM8K.

**Strengths:**

1.The paper's claims are supported by comprehensive experiments. The authors validate their method's effectiveness across a diverse range of models (Qwen series, Llama-3.2-3B-Instruct)and multiple training datasets (MATH, DAPO-14k, OpenRS). The evaluation is similarly thorough, spanning not only in-domain mathematical reasoning but also out-of-domain tasks like code generation and general abilities (MMLU-Pro, IFEval) . This extensive validation strongly supports the paper's conclusions.
2. A key strength of this work is the principled and general nature of the proposed Co-rewarding framework. The two instantiations are not ad-hoc fixes but are well-grounded in established, sound concepts: Co-rewarding-I draws its intuition from contrastive learning (analogy-invariance) , while Co-rewarding-II effectively implements a form of self-distillation (temporal invariance).

**Weaknesses:**

1. Although I like the idea of Co-rewarding-I, it seems not that effective. It relies on additional stronger model to revise the training data but offers marginal improvement in across both training datasets. Sometimes may harm the performance. The author tries to show the effectiveness of rewarding-I with a ablation study but the ablation is not fair in my view point, the fair comparasion should be a model trained on the union of orignal and rephrased instead of training on them seperately since the rewarding-I use both of the datasets.
2. The ablation study in Table 3 is confusing as it appears to conflate results from different training sets. Specifically, the Co-rewarding-I ablation data (e.g., "Majority-Voting w/ Original") matches the MATH-trained results from Table 1 , while the Co-rewarding-II ablation data ("w/o Updating Reference") matches the DAPO-14k-trained results from Table 2. This is not explicitly stated in the table's caption or the main text, making a direct comparison difficult.

**Questions:**

1. The evaluation omits high-difficulty, competition-level benchmarks (e.g., AIME 24/25). Given that the training set includes data from DAPO-Math-17k, why wasn't a benchmark like AIME used to more rigorously validate the model's advanced reasoning capabilities?
2. The design of Co-rewarding-II, which uses a dynamic, EMA-updated 'teacher' as the reference for the KL constraint , seems to violate the fundamental purpose of this term. Traditionally, the KL divergence penalizes deviation from a fixed reference policy, which acts as a stable anchor to prevent catastrophic forgetting . By using a 'moving anchor' that slowly follows the student policy, doesn't this design remove the primary safeguard against significant policy drift?
3. The paper presents Co-rewarding-I (data-side) and Co-rewarding-II (model-side) as two 'orthogonal' instantiations of the core framework. Given their complementary nature, was a combined 'Co-rewarding-I+II' method ever explored?

---

> ### Author Response · Authors · 2025-11-22
> **Response to reviewer CbNe (Part 1/4)**
>
> We sincerely thank the reviewer for the time and effort spent reviewing our paper and for acknowledging the novelty of our method, the principled design of our framework, and solid experimental results. Below, we address the raised concerns and questions in detail:
>
> > **W1:** Although I like the idea of Co-rewarding-I, it seems not that effective. It relies on additional stronger model to revise the training data but offers marginal improvement in across both training datasets. Sometimes may harm the performance. The author tries to show the effectiveness of rewarding-I with a ablation study but the ablation is not fair in my view point, the fair comparasion should be a model trained on the union of orignal and rephrased instead of training on them seperately since the rewarding-I use both of the datasets.
>
> **A:** Thanks for your insightful comments about the ablation study. **Following the reviewer's suggestions, we supplement an ablation study to train "Majority-Voting" on Qwen3-8B-Base and Llama-3.2-3B-Instruct using the union of original and rephrased questions of the MATH training set**. The experimental results are reported as follows:
> |Training Set: MATH|Method|MATH500|GSM8K|AMC|AIME24|LiveCode|CRUX|IFEval|MMLU-Pro|
> |-|-|-|-|-|-|-|-|-|-|
> |**Qwen3-8B-Base**|Co-rewarding-I|**81.2**|**93.70**|**51.20**|15.10|30.81|**66.00**|**55.79**|**59.95**|
> ||Majority-Voting w/ Union|80.2|93.48|49.70|15.63|**31.94**|64.88|54.25|59.51|
> ||Majority-Voting w/ Original|79.8|89.76|49.09|**15.83**|30.52|63.38|51.80|56.93|
> ||Majority-Voting w/ Rephrased|79.2|91.51|50.75|15.00|31.66|60.38|52.24|57.26|
> |**Llama-3.2-3B-Instruct**|Co-rewarding-I|**50.2**|79.45|**23.80**|**10.00**|**11.28**|29.88|**48.89**|33.77|
> ||Majority-Voting w/ Union|48.0|**80.52**|21.84|9.69|10.14|30.00|43.35|**34.05**|
> ||Majority-Voting w/ Original|46.8|78.77|20.48|9.27|11.00|**31.25**|47.96|33.18|
> ||Majority-Voting w/ Rephrased|44.0|78.85|21.23|8.65|10.04|17.25|47.84|33.72|
>
> It can be observed that Co-rewarding-I generally outperforms "Majority-Voting w/ Union" on most benchmarks, while and in some cases “Majority-Voting w/ Union” even performs worse than “Majority-Voting w/ Original” using less data. Although the union of original and rephrased questions enlarges the training data, **solely relying on more data does not mitigate the inherent instability of single-view self-rewarding methods** such as Majority-Voting, which are prone to training collapse. **Consequently, adding more data cannot reliably translate into performance gains, and may even degrade results on certain benchmarks**. By contrast, the more stable training dynamics of Co-rewarding enable the model to benefit from both original and rephrased questions, leading to the observed performance gains. These results will be included in the revised manuscript.

---

> ### Author Response · Authors · 2025-11-22
> **Response to reviewer CbNe (Part 2/4)**
>
> > **W2:** The ablation study in Table 3 is confusing as it appears to conflate results from different training sets. Specifically, the Co-rewarding-I ablation data (e.g., "Majority-Voting w/ Original") matches the MATH-trained results from Table 1 , while the Co-rewarding-II ablation data ("w/o Updating Reference") matches the DAPO-14k-trained results from Table 2. This is not explicitly stated in the table's caption or the main text, making a direct comparison difficult.
>
> **A:** Thanks for your valuable comments regarding the ablation study. The reviewer's understanding is correct: the ablation for Co-rewarding-I was conducted on the MATH training set, whereas the ablation for Co-rewarding-II was conducted on the DAPO-14k training set. We agree with the reviewer that this may cause confusion when interpreting Table 3. To make the ablation analysis more complete and easier to follow, **we additionally include the Co-rewarding-II ablation on the MATH training set using Qwen3-8B-Base and Llama-3.2-3B-Instruct**. The supplemental results are summarized as follows:
> |Training Set: MATH|Method|MATH500|GSM8K|AMC|AIME24|LiveCode|CRUX|IFEval|MMLU-Pro|
> |-|-|-|-|-|-|-|-|-|-|
> |**Qwen3-8B-Base**|Co-rewarding-II|**80.8**|**92.42**|**53.46**|**14.48**| 30.23|**62.83**|**60.70**|**57.50**|
> ||w/o Updating Reference|79.2|89.46|51.51|13.96|**30.62**|61.75|56.93|51.85|
> |**Llama-3.2-3B-Instruct**|Co-rewarding-II|**49.8**|**79.30**|**22.59**|**10.73**|**10.80**|30.63|**49.90**|**33.61**|
> ||w/o Updating Reference|47.0|78.92|22.29|9.06|5.50|**31.25**|47.88|33.32|
>
> |Training Set: DAPO-14k|Method|MATH500|GSM8K|AMC|AIME24|LiveCode|CRUX|IFEval|MMLU-Pro|
> |-----------------------|-------------------------|---------|-------|-------|--------|----------|-------|--------|----------|
> | **Qwen3-8B-Base**     | Co-rewarding-II|**80.6**|**94.01**|**54.37**|**16.35**|**31.66**|**67.12**|**53.31**|**59.83**|
> |                       | w/o Updating Reference  | 78.0    | 88.40 | 51.66 | 15.94  | 30.62    | 63.75 | 52.48  | 58.01    |
> | **Llama-3.2-3B-Instruct**      | Co-rewarding-II         |**49.8**|**78.62**|**19.73**|**8.02**|**10.43**|**32.25**|**51.92**|**34.46**|
> |           | w/o Updating Reference  | 45.0    | 76.72 | 17.92 |**8.02**| 10.05    | 30.63 | 51.33  | 33.94    |
>
> From these results, we observe that the ablation results on the MATH exhibit the same trend as those on DAPO-14k: removing the EMA update of the reference teacher leads to performance decline. This suggests that maintaining a slowly updated teacher is important for improving pseudo-label quality. **To make the presentation clear, we will update Table 3 in the revised manuscript to explicitly specify the corresponding training sets for each ablation setting**, ensuring that readers can clearly identify the experimental configuration.

---

> ### Author Response · Authors · 2025-11-22
> **Response to reviewer CbNe (Part 3/4)**
>
> > **Q1:** The evaluation omits high-difficulty, competition-level benchmarks (e.g., AIME 24/25). Given that the training set includes data from DAPO-Math-17k, why wasn't a benchmark like AIME used to more rigorously validate the model's advanced reasoning capabilities?
>
> **A:** Thanks for your constructive comments. AIME is indeed a widely used benchmark for LLM reasoning. We agree that including this benchmark would provide a more rigorous assessment of our method. **Following the reviewer's suggestion, we supplement the evaluation on AIME24 for all models trained in our experiments.** Consistent with prior work [1,2], we adopt Avg@32 as the evaluation metric. The results are summarized as follows:
>
> |Training Set: DAPO-14k (AIME24)|Qwen3-8B-Base| Qwen3-4B-Base | Llama-3.2-3B-Instruct |
> |-|-|-|-|
> |Before RL|3.75|4.58|3.75|
> |GT-Reward|24.58|20.63|9.17|
> |Self-Certainty|15.00|8.85|4.79|
> |Entropy|15.00|12.81|6.56|
> |Majority-Voting|11.25|9.17|8.74|
> |Co-rewarding-I|11.88|10.63|7.08|
> |Co-rewarding-II|16.35|14.27|8.02|
>
> |Training Set: DAPO-14k (AIME24)| Qwen3-8B-Base | Qwen3-4B-Base | Llama-3.2-3B-Instruct | Qwen3-1.7B-Base | Qwen2.5-3B | Qwen2.5-7B |
> |-----|---------------|---------------|------------------------|------------------|------------|-------------|
> | Before RL         | 3.75          | 4.58          | 3.75                   | 1.15             | 0.52       | 2.81        |
> | GT-Reward         | 17.15         | 15.00         | 11.67                  | 8.23             | 6.77       | 14.06       |
> | Self-Certainty    | 15.73         | 11.67         | 6.88                   | 3.02             | 5.00       | 8.75        |
> | Entropy           | 15.63         | 10.94         | 6.56                   | 6.88             | 5.94       | 10.73       |
> | Majority-Voting   | 15.83         | 10.10         | 9.27                   | 7.50             | 5.10       | 11.04       |
> | Co-rewarding-I    | 15.10         | 13.85         | 10.00                  | 8.65             | 5.31       | 10.73       |
> | Co-rewarding-II   | 14.48         | 12.17         | 10.73                  | 7.50             | 4.47       | 11.98       |
>
> From these results, we observe that GT-Reward, which benefits from reliable label-supervised rewards, achieves the strongest performance on AIME24. Both Co-rewarding-I and Co-rewarding-II outperform the existing self-rewarding baselines (Self-Certainty, Entropy, and Majority-Voting). **Specifically, Co-rewarding-I achieves an average improvement of 3.98% on the MATH training set, and Co-rewarding-II achieves an average improvement of 4.05% on the DAPO-14k training set over the best self-rewarding baselines.** These results indicate that effective self-supervised RL can still yield meaningful gains on high-difficulty reasoning benchmark AIME24, and that mitigating training collapse further boost performance improvements. These results will be updated into the corresponding tables in our revised manuscript.
>
> [1] Yu Q, Zhang Z, Zhu R, et al. Dapo: An open-source llm reinforcement learning system at scale[J]. arXiv preprint arXiv:2503.14476, 2025.
> [2] Zeng W, Huang Y, Liu Q, et al. Simplerl-zoo: Investigating and taming zero reinforcement learning for open base models in the wild[J]. arXiv preprint arXiv:2503.18892, 2025.
>
> > **Q2:** The design of Co-rewarding-II, which uses a dynamic, EMA-updated 'teacher' as the reference for the KL constraint , seems to violate the fundamental purpose of this term. Traditionally, the KL divergence penalizes deviation from a fixed reference policy, which acts as a stable anchor to prevent catastrophic forgetting . By using a 'moving anchor' that slowly follows the student policy, doesn't this design remove the primary safeguard against significant policy drift?
>
> **A:** Thanks for your valuable comments. We understand the reviewer's concern that an EMA-updated reference teacher may appear to weaken the regularization effect of KL penalty in GRPO. In practice, **the EMA update rule ensures that the teacher evolves at a much slower pace than the online policy, which allows it to serve as an approximately stable anchor for KL-based regularization**.
>
> Empirically, **we further evaluate Co-rewarding-II on MMLU-Pro and IFEval to examine whether such a design causes harmful policy drift or degrades the model’s general-domain instruction-following and multi-task capabilities**. As shown in Table 1 and Table 2, Co-rewarding-II does not exhibit any degradation on these benchmarks, indicating that the EMA teacher does not result in undesirable drift. Moreover, recent studies such as DAPO [1] and Skywork [3] have reported strong performance even without KL regularization, that the role of KL penalty and policy drift in LLM RL remains an open problem for further discussion in future research.
>
> [3] He J, Liu J, Liu C Y, et al. Skywork open reasoner 1 technical report[J]. arXiv preprint arXiv:2505.22312, 2025.

---

> ### Author Response · Authors · 2025-11-22
> **Response to reviewer CbNe (Part 4/4)**
>
> > **Q3:** The paper presents Co-rewarding-I (data-side) and Co-rewarding-II (model-side) as two 'orthogonal' instantiations of the core framework. Given their complementary nature, was a combined 'Co-rewarding-I+II' method ever explored?
>
> **A:** Thanks for your insightful comments. Following your valuable suggestions, **we supplement Co-rewarding-III, a new instantiation that combines Co-rewarding-I and Co-rewarding-II to both consider data-side cross-supervision and model-side self-distillation**. Specifically, Co-rewarding-III produces pseudo labels for both the original and the rephrased questions using the slowly-updated reference teacher, and then adopts cross-supervision between the two views based on these teacher-generated pseudo labels. The experiments of Co-rewarding-III are conducted on two training sets, i.e., MATH and DAPO-14k, across three models, i.e., Qwen3-8B/4B-Base and Llama-3.2-3B-Instruct:
> |Training Set: MATH| Method           | MATH500 | GSM8K | AMC   | AIME24 | LiveCode | CRUX  | IFEval | MMLU-Pro |
> |-----------------------|------------------|---------|-------|-------|--------|----------|-------|--------|----------|
> | **Qwen3-8B-Base**     | Co-rewarding-I   | 81.2    | 93.70 | 51.20 | 15.10  | 30.81    | 66.00 | 55.79  | 59.95    |
> |                       | Co-rewarding-II  | 80.8    | 92.42 | 53.46 | 14.48  | 30.23    | 62.83 | 60.70  | 57.50    |
> |                       | Co-rewarding-III | 81.4    | 90.98 | 54.07 | 13.33  | 30.71    | 63.75 | 53.69  | 59.10    |
> | **Qwen3-4B-Base**     | Co-rewarding-I   | 78.8    | 91.28 | 46.08 | 13.85  | 26.64    | 56.50 | 50.35  | 53.26    |
> |                       | Co-rewarding-II  | 78.0    | 88.86 | 45.93 | 12.17  | 26.25    | 55.00 | 51.30  | 53.88    |
> |                       | Co-rewarding-III | 78.6    | 90.75 | 48.80 | 12.71  | 26.16    | 56.00 | 49.23  | 53.08    |
> | **Llama-3.2-3B-Instruct**      | Co-rewarding-I   | 50.2    | 79.45 | 23.80 | 10.00  | 11.28    | 29.88 | 48.89  | 33.77    |
> |          | Co-rewarding-II  | 49.8    | 79.30 | 22.59 | 10.73  | 10.80    | 30.63 | 49.90  | 33.61    |
> |                       | Co-rewarding-III | 51.6    | 79.91 | 25.45 | 10.42  | 10.43    | 32.50 | 46.37  | 34.50    |
>
> | Training Set: DAPO-14k| Method           | MATH500 | GSM8K | AMC   | AIME24 | LiveCode | CRUX  | IFEval | MMLU-Pro |
> |-----------------------|------------------|---------|-------|-------|--------|----------|-------|--------|----------|
> | **Qwen3-8B-Base**     | Co-rewarding-I   | 78.4    | 88.02 | 51.20 | 11.88  | 29.38    | 62.50 | 50.17  | 55.39    |
> |                       | Co-rewarding-II  | 80.6    | 94.01 | 54.37 | 16.35  | 31.66    | 67.12 | 53.31  | 59.83    |
> |                       | Co-rewarding-III | 81.6    | 92.27 | 53.77 | 17.71  | 32.70    | 66.75 | 55.85  | 60.02    |
> | **Qwen3-4B-Base**     | Co-rewarding-I   | 73.8    | 75.89 | 45.83 | 10.63  | 26.25    | 50.12 | 46.84  | 51.51    |
> |                       | Co-rewarding-II  | 77.8    | 91.89 | 48.49 | 14.27  | 26.64    | 54.87 | 48.90  | 52.83    |
> |                       | Co-rewarding-III | 79.2    | 90.45 | 48.95 | 15.10  | 27.58    | 54.87 | 50.30  | 54.79    |
> | **Llama-3.2-3B-Instruct**      | Co-rewarding-I   | 46.0    | 70.58 | 20.93 | 7.08   | 9.57     | 27.25 | 53.04  | 32.61    |
> || Co-rewarding-II  | 49.8    | 78.62 | 19.73 | 8.02   | 10.43    | 32.25 | 51.92  | 34.46    |
> |                       | Co-rewarding-III | 48.6    | 76.95 | 21.84 | 8.13   | 9.86     | 30.50 | 49.92  | 34.01    |
>
> From the above results, we observe that Co-rewarding-III that combines two views exhibits performance gain compared to Co-rewarding-I and Co-rewarding-II. Specifically, on four math reasoning benchmarks (MATH500, GSM8K, AMC, and AIME24), despite slight drop of performance in some cases, **Co-rewarding-III achieves overall improvements on average of 7.11% and 1.72% over Co-rewarding-I and Co-rewarding-II, respectively**. On code generation benchmarks (LiveCode and CRUX), Co-rewarding-III brings average gains of 5.51% over Co-rewarding-I and 0.21% over Co-rewarding-II. These results indicate that integrating data-side cross-supervision and model-side self-distillation makes reward hacking more difficult, leading to more stable self-supervised RL training. We will revise the manuscript to incorporate the Co-rewarding-III description and its experimental results.

---

> ### Author Response · Authors · 2025-11-23
> **Kind reminder: updated manuscript**
>
> Thanks again for the reviewer's valuable time and efforts in reviewing our manuscript.
>
> We would like to let the reviewer know that **the manuscript has been updated**, and all modified content is marked with **purple** highlights for clarity. For the modified tables and figures, their captions have also been highlighted in **purple**.
>
> This is just a kind reminder. If you have any concerns or further suggestions, we would be glad to engage in further discussion.
>
> Authors of 11193

---

> > ### Comment · Reviewer_CbNe · 2025-11-26
> >
> > Thanks to the authors for the quick response and the additional experiments they significantly strengthen the manuscript. I am wondering whether the authors could add a column reporting the mean values to the table in Part 4. Wishing you a pleasant Thanksgiving.

---

> > > ### Author Response · Authors · 2025-11-26
> > > **Response to reviewer CbNe**
> > >
> > > Thanks for your kind suggestion. Of course, including the column of average score makes the presentation more intuitive. Following your suggestion, **we have added not only the overall average in the last column, but also the averages across the four math benchmarks (MATH500, GSM8K, AMC, and AIME24) and the two code generation benchmarks (LiveCode and CRUX), respectively**. The updated table is presented below:
> > > |Training Set: MATH|Method|Mathematics|||||Code|||Instruction|Multi-Task||
> > > |-|-|-|-|-|-|-|-|-|-|-|-|-|
> > > |||**MATH500**|**GSM8K**|**AMC**|**AIME24**|**Avg.**|**LiveCode**|**CRUX**|**Avg.**|**IFEval**|**MMLU-Pro**|**Overall Avg.**|
> > > |**Qwen3-8B-Base**|Co-rewarding-I|81.2|93.70|51.20|15.10|60.30|30.81|66.00|48.41|55.79|59.95|56.72|
> > > ||Co-rewarding-II|80.8|92.42|53.46|14.48|60.29|30.23|62.83|46.53|60.70|57.50|56.55|
> > > ||Co-rewarding-III|81.4|90.98|54.07|13.33|60.04|30.71|63.75|47.23|53.69|59.10|55.88|
> > > |**Qwen3-4B-Base**|Co-rewarding-I|78.8|91.28|46.08|13.85|57.50|26.64|56.50|41.57|50.35|53.26|52.10|
> > > ||Co-rewarding-II|78.0|88.86|45.93|12.17|56.24|26.25|55.00|40.63|51.30|53.88|51.42|
> > > ||Co-rewarding-III|78.6|90.75|48.80|12.71|**57.71**|26.16|56.00|**41.08**|49.23|53.08|51.92|
> > > |**Llama-3.2-3B-Instruct**|Co-rewarding-I|50.2|79.45|23.80|10.00|40.86|11.28|29.88|20.58|48.89|33.77|35.91|
> > > ||Co-rewarding-II|49.8|79.30|22.59|10.73|40.61|10.80|30.63|20.72|49.90|33.61|35.92|
> > > ||Co-rewarding-III|51.6|79.91|25.45|10.42|**41.84**|10.43|32.50|**21.47**|46.37|34.50|**36.40**|
> > >
> > > |Training Set: DAPO-14k|Method|Mathematics|||||Code|||Instruction|Multi-Task||
> > > |-|-|-|-|-|-|-|-|-|-|-|-|-|
> > > |||**MATH500**|**GSM8K**|**AMC**|**AIME24**|**Avg.**|**LiveCode**|**CRUX**|**Avg.**|**IFEval**|**MMLU-Pro**|**Overall Avg.**|
> > > | **Qwen3-8B-Base**|Co-rewarding-I   | 78.4    | 88.02 | 51.20 | 11.88|57.38  | 29.38    | 62.50|45.94 | 50.17  | 55.39|53.37|
> > > |  | Co-rewarding-II  | 80.6 | 94.01 | 54.37 | 16.35|61.33 | 31.66  | 67.12|49.39 | 53.31  | 59.83|57.16    |
> > > |  | Co-rewarding-III | 81.6 | 92.27 | 53.77 | 17.71 |**61.34** | 32.70 | 66.75|**49.73** | 55.85  | 60.02 |**57.58**   |
> > > | **Qwen3-4B-Base** | Co-rewarding-I   | 73.8    | 75.89 | 45.83 | 10.63|51.54  | 26.25    | 50.12|38.19 | 46.84  | 51.51 |47.61|
> > > |  | Co-rewarding-II  | 77.8  | 91.89 | 48.49 | 14.27|58.11  | 26.64    | 54.87|40.75 | 48.90  | 52.83|51.96    |
> > > | | Co-rewarding-III | 79.2 | 90.45 | 48.95 | 15.10 |**58.43** | 27.58  | 54.87 |**41.22** | 50.30  | 54.79 |**52.66** |
> > > | **Llama-3.2-3B-Instruct**| Co-rewarding-I | 46.0 | 70.58 | 20.93 | 7.08 | 36.15 | 9.57  | 27.25 | 18.41 | 53.04  | 32.61 | 33.38  |
> > > || Co-rewarding-II  | 49.8  | 78.62 | 19.73 | 8.02 |39.04  | 10.43    | 32.25 |21.34| 51.92  | 34.46 |35.65 |
> > > |  | Co-rewarding-III | 48.6 | 76.95 | 21.84 | 8.13 |38.88  | 9.86  | 30.50 | 20.18 | 49.92  | 34.01  |34.98  |
> > >
> > > where we additionally highlight in bold the best average score achieved by Co-rewarding-III. It can be observed that **Co-rewarding-III achieves more best average scores than Co-rewarding-I and Co-rewarding-II**, suggesting that III effectively integrates the strengths of I and II to boost performance.
> > >
> > > Wish you a Happy Thanksgiving! Thanks again for your valuable advice and support to help us improve the quality of our manuscript. If you feel that our responses have adequately addressed your concerns, we would sincerely appreciate your reconsideration of the score.

---

> > > > ### Comment · Reviewer_CbNe · 2025-11-26
> > > >
> > > > I believe the authors have adequately addressed my concerns, and the proposed co-rewarding approach is intuitive and compelling. I support accepting this paper and will update my score to 8. Wishing the authors all the best.

---

> > > > > ### Author Response · Authors · 2025-11-26
> > > > > **Thank you**
> > > > >
> > > > > Dear Reviewer CbNe,
> > > > >
> > > > > Thank you very much for your continued support and insightful comments, which means a lot to us, especially during Thanksgiving. **We sincerely appreciate your reconsideration for raising the score, and we are grateful for your constructive feedback throughout the review process**.
> > > > >
> > > > > Thank you once again for your valuable time and efforts! Wish you a Happly Thanksgiving!
> > > > >
> > > > > Best regards,
> > > > >
> > > > > Authors of # 11193

---

> ### Author Response · Authors · 2025-11-26
> **Looking forward to your reply**
>
> Dear Reviewer CbNe,
>
> We sincerely thank you for your efforts in reviewing our work and for your great support! Please let us know if you need any further information or if there are additional points you would like to discuss with us. We would be glad to engage further discussion and answer any questions of your interest.
>
> Thank you very much!
>
> Best regards,
>
> Authors of #11193

---

### Official Review · Reviewer_qDgd · 2025-11-01

**Soundness:** 2
**Presentation:** 3
**Contribution:** 2
**Rating:** 4
**Confidence:** 2

**Summary:**

The paper proposes Co-rewarding, a self-supervised reinforcement learning framework designed to improve the reasoning ability of large language models without relying on ground-truth labels. The method aims to prevent the collapse and reward-hacking issues common in existing self-rewarding approaches by introducing complementary supervision from two perspectives: (1) Co-rewarding-I, which derives reward signals through contrastive agreement across semantically rephrased questions, and (2) Co-rewarding-II, which employs a slowly updated teacher model to provide temporally decoupled pseudo-labels. Experiments across several reasoning benchmarks (GSM8K, MATH, AMC, etc.) show that Co-rewarding achieves more stable training and in some cases matches or surpasses RL with ground-truth rewards, especially when using Qwen-3 and Llama-3 models.

**Strengths:**

+ Introduces a clear and well-motivated framework that targets a real weakness in current self-rewarding RL methods: instability and collapse caused by single-view reward signals.

+ Extensive empirical results across multiple models, datasets, and baselines, including ablations that isolate the contribution of each component.

+ The paper is overall well-written, with clear mathematical formulation and structured presentation.

**Weaknesses:**

+ The reliance on high-quality paraphrasing for Co-rewarding-I is insufficiently examined. The framework may degrade when rephrasing quality is low or domain-specific, yet no robustness analysis is provided. Although the paper claims that rephrased questions should yield similar reasoning outcomes, in practice the reasoning trace can vary a lot depending on how the question is phrased. Thus, how do the authors ensure the meta-transferability of both reasoning and final answers during rephrasing? Moreover, when paraphrasing leads to divergent results, how do the authors obtain the updated groundtruth?

+ Although positioned as scalable and label-free, the method still requires additional compute from rephrasing models and dual rollouts, but the paper does not report cost or latency analysis w.r.t. the baslines.

**Questions:**

1. The paper states that the two co-rewarding techniques stem from the same core insight but offer complementary perspectives for cross-supervision. However, it remains unclear whether combining them actually yields better performance than using either one alone. If yes, what is the underlying mechanism that makes the combination more effective than a single method?

2. The paper would benefit from additional case studies and quantitative evaluation of the rephrasing models, particularly under low-quality paraphrasing which may change the reasoning results. Since the method claims to stabilize RL, one would expect robustness to varying data-augmentation conditions. Can the authors analyze training stability when rephrasing degrades, and report how performance varies with different levels of paraphrasing noise?

---

> ### Author Response · Authors · 2025-11-22
> **Response to reviewer qDgd (Part 1/3)**
>
> We sincerely thank the reviewer for the time and effort spent reviewing our paper and for acknowledging the novelty of our method, paper presentation, and extensive experiments. Below, we address the raised concerns and questions in detail:
>
> > **W1:** The reliance on high-quality paraphrasing for Co-rewarding-I is insufficiently examined. The framework may degrade when rephrasing quality is low or domain-specific, yet no robustness analysis is provided. Although the paper claims that rephrased questions should yield similar reasoning outcomes, in practice the reasoning trace can vary a lot depending on how the question is phrased. Thus, how do the authors ensure the meta-transferability of both reasoning and final answers during rephrasing? Moreover, when paraphrasing leads to divergent results, how do the authors obtain the updated groundtruth?
> **Q2:** The paper would benefit from additional case studies and quantitative evaluation of the rephrasing models, particularly under low-quality paraphrasing which may change the reasoning results. Since the method claims to stabilize RL, one would expect robustness to varying data-augmentation conditions. Can the authors analyze training stability when rephrasing degrades, and report how performance varies with different levels of paraphrasing noise?
>
> **A:** Thanks for your insightful comments. The valuable comments mainly concern two aspects: **(1) the robustness analysis of rephrased questions**; and **(2) the potential divergent risks introduced by rephrasing**. We note that both concerns primarily pertain to the data-side instantiation Co-rewarding-I, whereas Co-rewarding-II does not rely on rephrased questions.
>
> To encourage high-quality paraphrasing, we employ the strong open-source Qwen3-32B model. As cases illustrated in Table 19, Qwen3-32B effectively preserves the analogically mathematical essence while providing different surface forms. Additionally, the ablation in Table 3 shows that "Majority-Voting w/ Rephrased Questions" performs similarly to "Majority-Voting w/ Original Questions", which suggests that the rephrased questions produced by Qwen3-32B have quality similar to the originals.
>
> We agree with the reviewer that a robustness analysis of rephrased questions is beneficial for understanding how Co-rewarding-I behaves when rephrasing quality decreases. **Therefore, we supplement additional experiments using smaller LLMs instead of Qwen3-32B for rephrasing.** To control architectural variability in the rephraser LLMs, we employ two smaller LLMs from the same family, i.e., Qwen3-8B and Qwen3-1.7B, for rephrasing the MATH training set. The rephrasing success rates are summarized below:
> |Model|Original Questions|Successfully Rephrased|Success Rate (%)|
> |-|-|-|-|
> |Qwen3-32B|7,500|7,498| 99.97%|
> |Qwen3-8B|7,500|7,477|99.69%|
> |Qwen3-1.7B|7,500|2,060|27.47%|
>
> We observe that rephrasing success rates drop as the model size decreases, which is expected: rephrasing math questions while preserving the analogical essence is a relatively challenging task, and weaker LLMs struggle to achieve this goal. This observation supports our choice of Qwen3-32B as the rephraser, as a sufficiently capable LLM is required to produce faithful rephrasing. **We then train Co-rewarding-I on Qwen3-8B-Base using rephrased data generated by Qwen3-32B, Qwen3-8B, and Qwen3-1.7B, respectively.** The performance is reported as follows:
> |Trained Model|Rephraser LLM|MATH500|GSM8K|AMC|AIME24|LiveCode|CRUX|IFEval|MMLU-Pro|
> |-|-|-|-|-|-|-|-|-|-|
> |Qwen3-8B-Base|Qwen3-32B|**81.2**|**93.70**|51.20|**15.10**|30.81|**66.00**|**55.79**|**59.95**|
> |Qwen3-8B-Base|Qwen3-8B| 79.2|92.72|**51.51**|14.58|**30.90**|63.12|54.73|59.30|
> |Qwen3-8B-Base|Qwen3-1.7B|78.2|87.41|49.25|12.81|29.57|61.00|53.44|55.85|
>
> From the results, **it can be observed that performance gradually degrades as the size of the rephraser LLM decreases, but not always significantly. Rephrasing with Qwen3-8B maintains reasonably similar performance to using Qwen3-32B, indicating that Co-rewarding-I exhibits a certain degree of robustness under moderate reductions in rephrasing quality.** Notably, rephrasing with Qwen3-1.7B leads to a substantial performance drop. This degradation is largely attributable to the significantly lower rephrasing success rate of Qwen3-1.7B, which results in a substantial reduction of usable training data and consequently weakens the effectiveness of Co-rewarding-I. We will update these experimental results in the revised manuscript.

---

> ### Author Response · Authors · 2025-11-22
> **Response to reviewer qDgd (Part 2/3)**
>
> [continued response to **W1** and **Q2**]
> Furthermore, we agree with the reviewer that considering potential divergent risks introduced by rephrasing is important for a comprehensive analysis of Co-rewarding-I. **Our rephrasing template (Appendix C.2)** explicitly instructs the rephraser LLM to maintain the underlying mathematical essence of each question with the same ground-truth answer between original and rephrased questions while only differing on surface descriptions. **From the perspective of human mathematical reasoning, when two questions share the same mathematical essence, despite different descriptions, the reasoning steps required to solve them should remain similar.** This motivates our cross-supervision design in Co-rewarding-I: the pseudo labels of original and rephrased questions supervise each other, increasing the difficulty of reward hacking and reducing the chance of training collapse. These results will be included in the revised manuscript.
>
> > **W2:** Although positioned as scalable and label-free, the method still requires additional compute from rephrasing models and dual rollouts, but the paper does not report cost or latency analysis w.r.t. the baslines.
>
> **A:** Thanks for your valuable comments. We clarify that **the rephrasing step is performed entirely as a one-time data preprocessing procedure, and thus does not affect subsequent RL training**. Moreover, **rephrasing is only required for Co-rewarding-I, whereas Co-rewarding-II does not require data rephrasing**. To clearly present the computational cost of rephrasing, we report the concrete rephrasing time and throughput rate by Qwen3-32B model using `vllm` with one H100-80G GPU for the three training sets:
> |Dataset|Size|Rephrasing Time|Throughput Rate (questions/sec)|
> |-|-|-|-|
> |Math|7,500|18 min 25 s|6.79|
> |DAPO-14k|14,100|29 min 38 s|7.93|
> |OpenRS|7,000|7 min 16 s|16.06|
>
> It can be observed that even using a 32B-level model to rephrase thousands of questions, the total rephrasing time remains computationally feasible. Importantly, **rephrasing occurs only once before training begins**.
>
> Regarding rollout cost, the understanding of the reviewer is correct that Co-rewarding performs dual rollouts compared to vanilla GRPO. **We view this as a reasonable and widely accepted cost in self-supervised learning settings where ground-truth (GT) labels are unavailable.** Similar patterns are widely observed in contrastive learning frameworks such as SimCLR [1] and InfoNCE [2], which also rely on augmentation data and extra computation to encode the augmentation data. Importantly, **our Co-rewarding framework is intentionally designed to keep computational overhead low:** no additional models are introduced beyond vanilla GRPO, and Co-rewarding-II reuses the existing reference model as a slowly updated teacher. To provide a clear analysis of training costs, we report the training time and throughput rate for three training epochs on MATH and DAPO-14k using 4×H100-80G GPUs, with validation performed every 10 training steps:
> |Training Set|Model|Steps|Training Time (Vanilla GRPO)|Throughput (steps/hour)|Training Time (Co-rewarding-I)|Throughput |Training Time (Co-rewarding-II)|Throughput|
> |-|-|-|-|-|-|-|-|-|
> |MATH|Qwen3-8B-Base|174|6 h 56 min 45 s|25.06|9 h 48 min 50 s|17.73|7 h 42 min 57 s|22.55|
> |MATH|Qwen3-4B-Base|174|4 h 7 min 43 s|42.13|5 h 49 min 32 s|29.86|5 h 55 min 12 s|29.39|
> |MATH|Llama-3.2-3B-Instruct|174|3 h 12 min 32 s|54.21|4 h 19 min 12 s|40.28|4 h 6 min 41 s|42.32|
> |MATH|Qwen3-1.7B-Base|174|2 h 27 min 43 s|70.68|3 h 52 min 43 s|44.87|3 h 35 min 31 s|48.44|
> |MATH|Qwen2.5-7B|174|4 h 32 min 25 s|38.35|6 h 45 min 17 s|25.77|6 h 19 min 25 s|27.52|
> | MATH         | Qwen2.5-3B              | 174   | 2 h 54 min 55 s      | 59.67              | 3 h 54 min 39 s      | 44.50            | 4 h 23 min 50 s      | 39.57             |
> | DAPO-14k     | Qwen3-8B-Base           | 327   | 13 h 49 min 53 s     | 23.64              | 16 h 41 min 16 s     | 19.59            | 17 h 38 min 22 s     | 18.54             |
> | DAPO-14k     | Qwen3-4B-Base           | 327   | 9 h 35 min 28 s      | 34.08              | 14 h 29 min 51 s     | 22.55            | 13 h 51 min 39 s     | 23.59             |
> | DAPO-14k     | Llama-3.2-3B-Instruct   | 327   | 6 h 3 min 58 s       | 53.92              | 8 h 46 min 07 s      | 37.31            | 9 h 32 min 52 s      | 34.25             |
>
> From these results, the extra cost introduced by Co-rewarding remains moderate and acceptable. Although dual rollouts reduce throughput rate compared to vanilla GRPO, the increasing cost remains well within practical training budgets.
>
> [1] Chen T, Kornblith S, Norouzi M, et al. A simple framework for contrastive learning of visual representations[C]//International conference on machine learning. PmLR, 2020: 1597-1607.
> [2] Oord A, Li Y, Vinyals O. Representation learning with contrastive predictive coding[J]. arXiv preprint arXiv:1807.03748, 2018.

---

> ### Author Response · Authors · 2025-11-22
> **Response to reviewer qDgd (Part 3/3)**
>
> > **Q1:** The paper states that the two co-rewarding techniques stem from the same core insight but offer complementary perspectives for cross-supervision. However, it remains unclear whether combining them actually yields better performance than using either one alone. If yes, what is the underlying mechanism that makes the combination more effective than a single method?
>
> **A:** Thanks for your insightful suggestions. Since Co-rewarding-I and Co-rewarding-II provide complementary perspectives from the data side and the model side respectively, we agree with the reviewer that integrating a combined variant would give a clearer understanding of our overall Co-rewarding framework. Following the reviewer's suggestions, **we introduce Co-rewarding-III, which integrates the data-side cross-supervision from Co-rewarding-I with the model-side teacher-based pseudo labels from Co-rewarding-II**, providing a dual-view constraint that further increases the difficulty of reward hacking. We conduct experiments on Co-rewarding-III using two training sets (MATH and DAPO-14k) and three LLMs (Qwen3-8B-Base, Qwen3-4B-Base, and Llama-3.2-3B-Instruct). The results evaluated on multiple benchmarks are summarized as follows:
> |Training Set: MATH| Method           | MATH500 | GSM8K | AMC   | AIME24 | LiveCode | CRUX  | IFEval | MMLU-Pro |
> |-----------------------|------------------|---------|-------|-------|--------|----------|-------|--------|----------|
> | **Qwen3-8B-Base**     | Co-rewarding-I   | 81.2    | 93.70 | 51.20 | 15.10  | 30.81    | 66.00 | 55.79  | 59.95    |
> |                       | Co-rewarding-II  | 80.8    | 92.42 | 53.46 | 14.48  | 30.23    | 62.83 | 60.70  | 57.50    |
> |                       | Co-rewarding-III | 81.4    | 90.98 | 54.07 | 13.33  | 30.71    | 63.75 | 53.69  | 59.10    |
> | **Qwen3-4B-Base**     | Co-rewarding-I   | 78.8    | 91.28 | 46.08 | 13.85  | 26.64    | 56.50 | 50.35  | 53.26    |
> |                       | Co-rewarding-II  | 78.0    | 88.86 | 45.93 | 12.17  | 26.25    | 55.00 | 51.30  | 53.88    |
> |                       | Co-rewarding-III | 78.6    | 90.75 | 48.80 | 12.71  | 26.16    | 56.00 | 49.23  | 53.08    |
> | **Llama-3.2-3B-Instruct**      | Co-rewarding-I   | 50.2    | 79.45 | 23.80 | 10.00  | 11.28    | 29.88 | 48.89  | 33.77    |
> |          | Co-rewarding-II  | 49.8    | 79.30 | 22.59 | 10.73  | 10.80    | 30.63 | 49.90  | 33.61    |
> |                       | Co-rewarding-III | 51.6    | 79.91 | 25.45 | 10.42  | 10.43    | 32.50 | 46.37  | 34.50    |
>
> | Training Set: DAPO-14k| Method           | MATH500 | GSM8K | AMC   | AIME24 | LiveCode | CRUX  | IFEval | MMLU-Pro |
> |-----------------------|------------------|---------|-------|-------|--------|----------|-------|--------|----------|
> | **Qwen3-8B-Base**     | Co-rewarding-I   | 78.4    | 88.02 | 51.20 | 11.88  | 29.38    | 62.50 | 50.17  | 55.39    |
> |                       | Co-rewarding-II  | 80.6    | 94.01 | 54.37 | 16.35  | 31.66    | 67.12 | 53.31  | 59.83    |
> |                       | Co-rewarding-III | 81.6    | 92.27 | 53.77 | 17.71  | 32.70    | 66.75 | 55.85  | 60.02    |
> | **Qwen3-4B-Base**     | Co-rewarding-I   | 73.8    | 75.89 | 45.83 | 10.63  | 26.25    | 50.12 | 46.84  | 51.51    |
> |                       | Co-rewarding-II  | 77.8    | 91.89 | 48.49 | 14.27  | 26.64    | 54.87 | 48.90  | 52.83    |
> |                       | Co-rewarding-III | 79.2    | 90.45 | 48.95 | 15.10  | 27.58    | 54.87 | 50.30  | 54.79    |
> | **Llama-3.2-3B-Instruct**      | Co-rewarding-I   | 46.0    | 70.58 | 20.93 | 7.08   | 9.57     | 27.25 | 53.04  | 32.61    |
> || Co-rewarding-II  | 49.8    | 78.62 | 19.73 | 8.02   | 10.43    | 32.25 | 51.92  | 34.46    |
> |                       | Co-rewarding-III | 48.6    | 76.95 | 21.84 | 8.13   | 9.86     | 30.50 | 49.92  | 34.01    |
>
>
> From the above results, across two training sets and four math reasoning benchmarks (MATH500, GSM8K, AMC, and AIME24), despite some occanional slight drop in performance, **Co-rewarding-III achieves improvements on average of 7.11% and 1.72% over Co-rewarding-I and Co-rewarding-II, respectively**. On the code-generation benchmarks (LiveCode and CRUX), Co-rewarding-III further yields average gains of 5.51% over Co-rewarding-I and 0.21% over Co-rewarding-II. These results suggest that combining data-side and model-side signals provides complementary advantages: the dual-view supervision further increases the difficulty of reward hacking and promotes more stable and effective self-supervised RL training. We will update the manuscript to integrate the content and experiments of Co-rewarding-III.

---

> ### Author Response · Authors · 2025-11-23
> **King reminder: updated manuscript**
>
> Thanks again for the reviewer's valuable time and efforts in reviewing our manuscript.
>
> We would like to let the reviewer know that **the manuscript has been updated**, and all modified content is marked with **purple** highlights for clarity. For the modified tables and figures, their captions have also been highlighted in **purple**.
>
> This is just a kind reminder. If you have any concerns or further suggestions, we would be glad to engage in further discussion.
>
> Authors of 11193

---

> ### Author Response · Authors · 2025-11-26
> **Looking forward to your reply**
>
> Dear Reviewer qDgd,
>
> We sincerely thank you for your efforts in reviewing our work and for your great support! Please let us know if you need any further information or if there are additional points you would like to discuss with us. We would be glad to engage further discussion with you.
>
> Thank you very much!
>
> Best regards,
>
> Authors of #11193

---

> ### Author Response · Authors · 2025-11-27
> **Looking forward to your reply**
>
> Dear Reviewer qDgd,
>
> We sincerely thank you for your insightful review and support of our manuscirpt. Please let us know if you need any further information or if there are additional points you would like to discuss with us. We would be glad to engage further discussion with you.
>
> We also wish you a very Happy Thanksgiving!
>
> Thank you once again for your valuable time and efforts.
>
> Best regards,
>
> Authors of #11193

---

### Author Response · Authors · 2025-11-26
**General response**

Dear All Reviewers and AC,

We would like to express our deep gratitude to all reviewers for their insightful suggestions and constructive comments, which are immensely helpful for us. During this period, we are pleased to receive numerous positive remarks, including:
- **Clear motivation for the training collapse issue:** qDgd, CbNe, pcnh.
- **Novel, principled, and well-grounded framework design:** qDgd, CbNe, pcnh.
- **Comprehensive and solid experiments:** qDgd, CbNe, odXG, pcnh.
- **Well-written quality of our manuscript:** qDgd.
- **Strong empirical performance of the proposed methods:** qDgd, CbNe, odXG, pcnh.
- **Practical and simple to implement:** CbNe, pcnh.
- **Broad generalization beyond training domain:** CbNe, odXG.

More importantly, inspired by the reviewers' valuable comments and suggestions, our manuscript has been continually improved regarding the unclear parts or experiments about some specific points. We carefully followed the reviewers' suggestions to include additional experiments in our revised manuscript. For your reference, we supplement the main additions in the following:
- **Addtional description of the combined instantiation Co-rewarding-III:** supplemented in Section 3.2 and Appendix B.1 and Appendix B.2.
- **Additional experiments of the combined instantiation Co-rewarding-III:** supplemented in Table 1 and Table 2.
- **New evaluation benchmark AIME24:** supplemented in Table 1, Table 2, and Table 7.
- **Robustness analysis of different rephraser LLMs:** supplemented in Table 13, Table 14, and Appendix D.8.
- **Additional ablation study of "Majority-Voting w/ Union":** supplemented in Table 3 and Section 4.2.2.
- **Additional ablation study on MATH training set:** supplemented in Table 3 and Section 4.2.2.
- **Impact of math training collapse on other tasks:** supplemented in Table 11 and Appendix D.6.
- **Quantitative features of MATH and DAPO-14k training sets:** supplemented in Table 12 and Appendix D.7.
- **Additional case study of code generation task:** supplemented in Appendix D.14.

Thanks once again for all reviewers' valuable time and efforts. Your insigtful comments, no matter about the pros or the cons, all have been instrumental in improving the quality of our manuscript and have inspired us to continue advancing this research.

Best regards,

Authors of #11193

---

### Meta-Review · Area_Chair_1D8q · 2026-01-02

**Summary:**

This paper proposes Co-rewarding, a self-supervised reinforcement learning framework for eliciting reasoning abilities in large language models. The framework is instantiated in two ways: Co-rewarding-I, which leverages contrastive agreement across semantically equivalent questions on the data side; Co-rewarding-II, which introduces a slowly updated EMA teacher to provide temporally decoupled pseudo-labels on the model side.  Experimental results across several reasoning benchmarks (GSM8K, MATH, AMC, etc.) show that Co-rewarding achieves more stable training and in some cases matches or surpasses RL with ground-truth rewards.

Overall, the paper addresses a timely and practically relevant problem, presents a well-motivated and reasonably novel framework. Moreover, the authors provided detailed responses to each reviewer during the rebuttal phase.

**Reviewer Concerns:**

All reviewers’ concerns have been addressed.

**Reviewer Scores:**

The authors have provided a detailed response to Reviewer qDgd’s comments, and I believe that Reviewer qDgd will consider raising their score based on the rebuttal.

---

### Decision · Program_Chairs · 2026-01-26

Accept (Poster)